# A Convergence Analysis of Adaptive Optimizers under Floating-point Quantization

**Xuan Tang**[*]
School of Computing and Data Science
The University of Hong Kong
xuantang8@connect.hku.hk

**Jichu Li**[*]
School of Statistics
Renmin University of China
lijichu52@gmail.com

**Difan Zou**
School of Computing and Data Science & Institute of Data Science
The University of Hong Kong
dzou@hku.hk

## Abstract

The rapid scaling of large language models (LLMs) has made low-precision training essential for reducing memory, improving efficiency, and enabling larger models and datasets. Existing convergence theories for adaptive optimizers, however, assume all components are exact and neglect hardware-aware quantization, leaving open the question of why low-precision training remains effective. We introduce the first theoretical framework for analyzing the convergence of adaptive optimizers, including Adam and Muon, under floating-point quantization of gradients, weights, and optimizer states (e.g., moment estimates). Within this framework, we derive convergence rates on smooth non-convex objectives under standard stochastic gradient assumptions, explicitly characterizing how quantization errors from different components affect convergence. We show that both algorithms retain rates close to their full-precision counterparts provided mantissa length scales only logarithmically with the number of iterations. Our analysis further reveals that Adam is highly sensitive to weights and second-moment quantization due to its reliance on $\beta_2 \to 1$, while Muon requires weaker error control and is thus potentially more robust. These results narrow the gap between empirical success and theoretical understanding of low-precision training methods. Numerical experiments on synthetic and real-world data corroborate our theory.

## 1 Introduction

The rapid scaling of large language models (LLMs) has made low-precision training indispensable for modern deep learning. By reducing memory usage and improving computational efficiency, low-precision formats such as bfloat16 (BF16) and FP8 enable training with larger models and datasets on contemporary hardware accelerators (Peng et al., 2023; Fishman et al., 2025). The introduction of FP8 in Nvidia's Hopper GPU architecture (NVIDIA, 2022; Micikevicius et al., 2022) further cements its role as a practical datatype for the next generation of LLM training. In practice, numerous frameworks now leverage mixed- or low-precision formats to quantize gradients, weights, and optimizer states (Liu et al., 2024; 2025), showing that aggressively quantized training can scale to trillion-token workloads without loss of accuracy.

Despite its empirical success, a rigorous theoretical understanding of quantization, particularly for adaptive optimizers like Adam (Kingma, 2014) with decoupled weight decay (Loshchilov & Hutter, 2019) and Muon (Jordan et al., 2024), which are widely used in practice, remain largely underdeveloped. Existing theoretical work on the non-convex optimization analysis under quantization has primarily focused on Stochastic Gradient Descent with quantized gradients (QSGD) (Alistarh et al., 2017). For example, Jiang & Agrawal (2018) established $\mathcal{O}(1/T^{1/4})$ convergence under un-

---

[*]Equal contribution.

biased quantization, while error-feedback mechanisms (Karimireddy et al., 2019) were later introduced to handle biased quantization with the same guarantees. Extensions to QSGD and Quantized SGDM with error feedback have been analyzed in various settings (Tang et al., 2019; Zheng et al., 2019; Koloskova et al., 2020), again achieving $\mathcal{O}(1/T^{1/4})$ rates. More recent efforts target Quantized Adam (Chen et al., 2021; Modoranu et al., 2024; Ozkara et al., 2025). Chen et al. (2021) proved convergence of Adam with quantized gradients and weights under error feedback achieves $\mathcal{O}(1/T^{1/4})$, but the method requires storing error terms for every parameter, which is memory-intensive and impractical for modern low-precision LLM training. Modoranu et al. (2024) reduced this cost by compressing error feedback with unbiased compression, proving $\mathcal{O}(1/T^{1/4})$ convergence for Adam with quantized gradients. Ozkara et al. (2025) further explored stochastic rounding (SR) as a mechanism for mitigating numerical errors in low-precision training, providing analyses of implicit regularization and convergence of Adam under SR; however, their analysis omits optimizer state quantization or practical floating-point formats, which are increasingly central to low-bit LLM optimization (Dettmers et al., 2021; Xi et al., 2025; Fishman et al., 2025). This leaves a critical gap: practical low-bit training crucially involves the quantization of optimizer states (e.g., momentum and second-moment estimates), a component these analyses omit. Furthermore, these studies often rely on assumptions like unbiased quantization or error-feedback mechanisms that are not consistent with modern large-scale LLM training. Consequently, the community lacks a theoretical framework to explain the robust convergence observed when adaptive optimizers are quantized in all components during LLM training.

**This paper.** The objective of this work is to develop the convergence analysis of adaptive optimization algorithms with a more practical quantization configuration. In particular, we develop the first analytical framework for quantized adaptive optimizers under floating-point quantization. More importantly, following the practical configuration (Liu et al., 2024), our framework explicitly models the quantization of all key components: gradients, weights, momentum, and second moments. We then establish convergence guarantees for both Adam and Muon optimizers, expressing the results as a function of the quantization errors in these components. This clearly reveals how each type of error individually affects convergence. Crucially, rather than relying on unbiased quantization assumptions or storing per-parameter error feedback, we require only relative error control, which aligns with the behavior of standard floating-point formats (FP32 → BF16 or FP8; Section 3, Figure 5, 6, 10, 11; see also (Kuzmin et al., 2022)).

We then summarize the main contributions of this work as follows:

- We introduce a rigorous analytical framework for adaptive optimizers under hardware-aware low-precision training, explicitly modeling the quantization of weights, gradients, and optimizer states (Section 3). Unlike prior works that rely on unbiased quantization assumptions or error-feedback mechanisms, which are impractical in large-scale LLM training, we adopt a relative error model (Assumption 3.1) that faithfully captures the behavior of floating-point quantization. This facilitates a formal and rigorous convergence analysis for quantized adaptive optimization algorithms that closely align with real-world implementations.

- We provide the first convergence guarantees for quantized Adam (Theorem 4.5) and Muon (Theorem 4.6) on smooth non-convex objectives under the relative error quantization model (Assumption 3.1), which closely reflects the behavior of floating-point quantization. Our analysis shows that both methods attain the same convergence rates as their full-precision counterparts (Défossez et al., 2022; Shen et al., 2025), provided the mantissa length increases only logarithmically with the number of iterations, which is consistent with practical hardware precision.

- Our analysis in Theorems 4.5 and 4.6 precisely characterizes how quantization errors in different components impact convergence. Notably, we show that Adam is particularly sensitive to quantization of weights and second moments due to their dependence on $\beta_2$, which is typically set close to 1 for convergence in practice and theory (Figure 7) (Zhang et al., 2022). This aligns with empirical observations from Peng et al. (2023); Yu et al. (2024), where weights and second moments require slightly higher precision than gradients or the momentum. Our experiments (Figures 3, 4, 8, 9, 12) corroborate this, demonstrating graceful degradation with reduced precision and near full-precision performance at moderate mantissa lengths. In contrast, Theorem 4.6 reveals that Muon is more tolerant to quantization, requiring weaker relative error conditions (e.g., $1/T^{1/2}$ versus $1/T^2$ for Adam). This robustness stems from the SVD-based sign operator in

Muon, which avoids the amplification of quantization errors by the inverse square root of historical gradient variances. This theoretical insight also explains empirical findings in Liu et al. (2025) that Muon exhibits superior robustness to low-precision training compared to Adam (Figure 13).

Overall, our results narrow the gap between the empirical success of quantized adaptive training and its theoretical understanding, providing a foundation for analyzing and designing future low-precision optimization algorithms.

**Notations.** Scalars are denoted by lowercase letters $(x, \ldots)$, vectors by bold lowercase $(\mathbf{x}, \ldots)$, and matrices by bold uppercase $(\mathbf{X}, \ldots)$. The $i$-th entry of $\mathbf{x}$ is $x_i$, and the $(i, j)$-th entry of $\mathbf{X}$ is $X_{ij}$. The $\ell_2$ norm of $\mathbf{x}$ is $\|\mathbf{x}\|_2 = \sqrt{\sum_i x_i^2}$, the Frobenius norm of $\mathbf{X}$ is $\|\mathbf{X}\|_F = \sqrt{\sum_{i,j} X_{ij}^2}$, and the nuclear norm of $\mathbf{X}$ is $\|\mathbf{X}\|_* = \sum_i \sigma_i(\mathbf{X})$, where $\sigma_i(\mathbf{X})$ denotes the $i$-th singular value. For $d \in \mathbb{N}^+$, let $[d] = \{1, 2, \ldots, d\}$. For real sequences $\{a_t\}$ and $\{b_t\}$, we write $a_t = O(b_t)$ if there exist constants $C, N > 0$ such that $a_t \leq Cb_t$ for all $t \geq N$; $a_t = \Omega(b_t)$ if $b_t = O(a_t)$; $a_t = \Theta(b_t)$ if both $a_t = O(b_t)$ and $a_t = \Omega(b_t)$; and we use $\widetilde{O}(\cdot)$ and $\widetilde{\Omega}(\cdot)$ to suppress logarithmic factors. The quantization operator is $\mathcal{Q}(\cdot)$, with $x^Q$ denoting the quantized version of $x$.

## 2 RELATED WORK

**Adaptive Optimization.** Adaptive optimizers are a key part of deep learning because they can automatically respond to changes in the data. The progression of modern adaptive optimizers began with Adagrad (Duchi et al., 2011), which scales learning rates based on the accumulated sum of past squared gradients. Despite extensive convergence analysis (Zou et al., 2019; Chen et al., 2018; Shi et al., 2020; Li & Orabona, 2019; Faw et al., 2022), its aggressive learning rate decay often leads to premature stalling. RMSProp (Hinton et al., 2012) addressed this issue by using an exponentially decaying average of squared gradients instead, a method whose convergence has also been well-studied (Zaheer et al., 2018; De et al., 2018; Shi et al., 2020; Li et al., 2025). Adam (Kingma, 2014) then synthesized these ideas by incorporating momentum, effectively combining the adaptive learning rates of RMSProp with first-moment estimates. Its widespread success has motivated a vast body of theoretical work analyzing its convergence and implicit bias generalization under various settings (Reddi et al., 2018; Défossez et al., 2022; Zou et al., 2019; Chen et al., 2018; Zhang et al., 2022; Wang et al., 2022; Guo et al., 2021; Hong & Lin, 2023; Li et al., 2023; Wang et al., 2023; Zhang et al., 2025; 2024; Zou et al., 2023; Cattaneo et al., 2024). More recently, the Muon optimizer (Jordan et al., 2024) was proposed, which leverages a matrix-based perspective for optimization, with its convergence guarantees established by concurrent works (Shen et al., 2025; Sato et al., 2025). While convergence guarantees for these methods have been established in high-precision settings, their behavior under the low-precision quantization common in modern large model training is not well understood, a gap that this paper aims to address.

**Low-bit Training.** As the field of deep learning continues to advance rapidly, the scale of models, particularly Large Language Models (LLMs), has grown exponentially. Low precision training (Wang et al., 2018; Wortsman et al., 2023; Liu et al., 2023; Xi et al., 2024; Liu et al., 2024) has become a prominent technique in modern deep learning, offering reductions in both computational costs and memory requirements. Mixed-precision training typically performs forward and backward passes in low-precision formats like FP16 (Micikevicius et al., 2017) or the more stable, wider-range BF16 (Kalamkar et al., 2019), while maintaining master weights and optimizer states in FP32. The advent of hardware like NVIDIA's Hopper GPU architecture (NVIDIA, 2022) has made 8-bit floating-point (FP8) training a practical reality for further efficiency gains (Micikevicius et al., 2022; Peng et al., 2023; Xi et al., 2025; Fishman et al., 2025). Even more aggressive approaches now extend to 4-bit (FP4) training (Wang et al., 2025; Zhou et al., 2025). Especially in adaptive optimization, the optimizer states can consume as much memory as the model parameters themselves. This has motivated a class of methods that specifically compresses these states, decompressing them to a higher precision just-in-time for the weight update to save memory (Dettmers et al., 2021; Peng et al., 2023; Li et al., 2024; Fishman et al., 2025; Xi et al., 2025). Despite the empirical success of these techniques, a comprehensive theory explaining their convergence behavior remains absent. Our work addresses this gap by establishing an analytical framework that formally incorporates

quantization errors from all parts of a realistic low-bit training pipeline, from gradients and weights to the crucial optimizer states themselves.

**Quantization Convergence.** Most convergence guarantees for optimizers assume ideal, high-precision arithmetic, failing to account for the quantization effects inherent in modern large-scale training. Much of the existing theoretical work in this area has therefore focused on the convergence of Quantized Stochastic Gradient Descent (SGD). Early analyses established convergence rates for SGD with quantized gradients, often relying on the strong assumption of an unbiased quantizer (Alistarh et al., 2017; Jiang & Agrawal, 2018; Wen et al., 2017). To handle more practical, biased quantization schemes, subsequent work introduced error-feedback mechanisms to compensate for the quantization bias and still guarantee convergence (Karimireddy et al., 2019; Zheng et al., 2019; Tang et al., 2019; Koloskova et al., 2020). Complementing these efforts on gradient compression, another line of research has analyzed the convergence of SGD when the model weights themselves are also quantized (Markov et al., 2023). Beyond SGD, analyzing quantized adaptive optimizers is a more recent challenge. Early work in this direction includes Hou et al. (2019), which studies Adam with $\beta_1 = 0$, analyzing joint quantization of both gradients and weights in convex settings. Other studies have applied error-feedback to ensure the convergence of Adam under quantized weights and gradients in non-convex settings (Chen et al., 2021; Modoranu et al., 2024; Robert et al., 2025). However, these existing analyses for adaptive methods rely heavily on error-feedback mechanisms, which are often impractical in state-of-the-art LLM training pipelines (Xi et al., 2025; Fishman et al., 2025). Complementary work on stochastic rounding (SR) (Ozkara et al., 2025) studies a different quantization regime: SR is approximately unbiased but introduces variance, and their Adam analysis quantizes only the final weight update while assuming full-precision gradients and optimizer states (with $\beta_1 = 0$). Such an additive-noise formulation and simplification avoid the recursive and interaction-heavy quantization error propagation that arises in adaptive optimization under realistic floating-point rounding. In contrast, our work addresses this critical gap by providing the first convergence framework for adaptive optimizers under a realistic floating-point error model that covers all components of the training process, without resorting to error-feedback or unbiasedness assumptions.

## 3 PRELIMINARIES AND PROBLEM SETUP

### 3.1 PRELIMINARIES

We begin by formalizing the quantization operator and its error properties. Our focus is on floating-point quantization, which is widely adopted in practice. Compared to integer quantization, floating-point formats achieve strictly smaller reconstruction errors due to their exponent scaling (Kuzmin et al., 2022). This explains why most large-scale low-precision training frameworks rely on floating-point representations, including recent FP8 and mixed-precision systems (Peng et al., 2023; Liu et al., 2024; Fishman et al., 2025).

**Floating-point quantization.** Let $\mathcal{Q} : \mathbb{R} \rightarrow \mathbb{R}$ be a scalar quantization operator applied elementwise to vectors and matrices. We illustrate $\mathcal{Q}$ through the common case of quantizing from single precision (fp32) to brain floating-point (bf16). The fp32 format uses 1 sign bit, 8 exponent bits, and 23 mantissa bits (total 32 bits) (IEEE, 2019), while bf16 keeps the same sign and exponent layout but truncates the mantissa to 7 bits (Wang & Kanwar, 2019). Thus, FP32 can be written as

$$x_{\text{fp32}} = (-1)^S \times 2^{E-127} \times (1.M_{0:22}),$$

where $S$ is the sign bit, $E$ the exponent, and $M_{0:22}$ the mantissa bits. Quantization discards the low-order 16 mantissa bits $M_{7:22}$, possibly with rounding or truncation. The BF16 number becomes

$$x_{\text{bf16}} = (-1)^S \times 2^{E-127} \times \left(1.M_{0:6} + C \cdot 2^{-7}\right),$$

where $C \in \{0, 1\}$ is a carry bit from rounding. Dequantization pads the truncated mantissa with zeros to recover an fp32 value. Figure 1 visualizes this process.

**Relative error.** The above construction implies that the quantization error satisfies

$$\left|x_{\text{bf16}} - x_{\text{fp32}}\right| = \left|C \cdot 2^{-7} - 0.M_{7:22} \cdot 2^{-7}\right| \cdot 2^{E-127} \leq 2^{-7} \cdot 2^{E-127} \leq q|x_{\text{fp32}}|,$$

Figure 1: Floating-point quantization from fp32 to bf16. Only the mantissa is truncated, while sign and exponent remain unchanged.

where $q = \Theta(2^{-M})$ and $M$ is the mantissa length of the target format (here $M = 7$ for bf16). More generally, we assume the absence of underflow and overflow, so that the sign and exponent remain unchanged after quantization. This prevents large quantization errors and guarantees convergence of the quantization process. In practice, this assumption is well justified: low-precision LLM training commonly employs engineering techniques such as per-tensor or per-channel scaling, which ensure that post-quantization values remain within the representable range (Peng et al., 2023; Fishman et al., 2025). Under this condition, the relative quantization error decays exponentially with the number of mantissa bits—a property that is intrinsic to floating-point representations. This observation motivates the following assumption.

**Assumption 3.1** (Quantization Error). Let $\mathcal{Q} : \mathbb{R} \to \mathbb{R}$ be a scalar quantization operator applied elementwise. Then, for any $x \in \mathbb{R}$, the quantization error is relatively bounded:

$$|x^Q - x| \leq q|x|,$$

where $q = \Theta(2^{-M})$, and $M$ is the mantissa length of the target floating-point format.

## 3.2 PROBLEM SETUP

We study stochastic optimization (3.1) with low-precision training under an analytical quantization framework shown in Figure 2. Formally, the goal is to minimize the loss:

$$\min_{\mathbf{W} \in \mathbb{R}^{m \times n}} F(\mathbf{W}) = \mathbb{E}_{\boldsymbol{\xi}}[f(\mathbf{W}; \boldsymbol{\xi})], \tag{3.1}$$

where $\mathbf{W}$ denotes the model parameters, $\boldsymbol{\xi}$ is a random variable representing the data, and $f(\mathbf{W}; \boldsymbol{\xi})$ is the sample loss. We denote $F^* = \inf_{\mathbf{W}} F(\mathbf{W}) > -\infty$ as the optimal objective value.

**Low-precision training framework.** During training, both computation and communication are constrained by memory and bandwidth. Modern practice therefore quantizes weights, gradients, and optimizer states into lower-precision formats (e.g., BF16, FP8) to accelerate training (Peng et al., 2023; Liu et al., 2024; Fishman et al., 2025). We model this process with the analytical framework shown in Figure 2. The key steps are:

1. The master maintains full-precision weights $\mathbf{W}_t$ but transmits their quantized version $\mathbf{W}_t^Q$ to workers.

2. Workers perform forward and backward passes with $\mathbf{W}_t^Q$, compute gradients $\nabla f(\mathbf{W}_t^Q; \boldsymbol{\xi})$, quantize them, and send quantized gradients back.

3. The master dequantizes gradients, updates quantized optimizer states (e.g., momentum, second moment), and applies the optimizer update. Updated states are re-quantized for storage.

The first two steps can be illustrated by Algorithm 1, while the third step depends on the choice of optimizer (e.g., Adam in Algorithm 2 or Muon in Algorithm 3). The dashed arrows in Figure 2 highlight the quantization operations applied to weights, gradients, and optimizer states within the proposed framework.

**Relative errors.** We denote the relative errors $q$ of different components after applying $\mathcal{Q}$ as

$$q_W \quad \text{(weights)}, \qquad q_G \quad \text{(gradients)}, \qquad q_M \quad \text{(first moment)}, \qquad q_V \quad \text{(second moment)}.$$

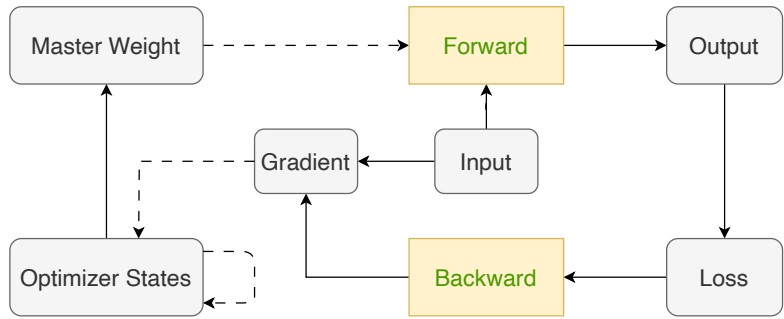

Figure 2: An analytical low-precision training framework

---

**Algorithm 1** Analytical Adaptive Method Quantization Training Framework

---

1: **Input 1**: Algorithm $\mathcal{A} \in \{\text{Adam}, \text{Muon}\}$ and its parameters set $\boldsymbol{\Theta} \in \{\{\beta_1, \beta_2, \epsilon\}, \{\beta\}\}$, initial weights $\mathbf{W}_0$, learning rate schedule $\{\eta_t\}$, batch size $B$, quantization operator $\mathcal{Q}$
2: **for** $t = 0, \dots, T-1$ **do**
3:     Sample batch $\{\boldsymbol{\xi}_{t,i}\}_{i=1}^B$ uniformly                  ▷ $B$ workers
4:     $\mathbf{G}_t = \frac{1}{B}\sum_{i=1}^B \nabla^Q f(\mathbf{W}_t^Q; \boldsymbol{\xi}_{t,i})$        ▷ Master receives $B$ quantized gradients
5:     $\mathbf{W}_{t+1} = \mathcal{A}(\mathbf{W}_t, \mathbf{G}_t, \boldsymbol{\Theta}, t)$                  ▷ Update by Adam or Muon
6: **end for**

---

| **Algorithm 2** Adam($\mathbf{W}_t, \mathbf{G}_t, \boldsymbol{\Theta}, t$) | **Algorithm 3** Muon($\mathbf{W}_t, \mathbf{G}_t, \boldsymbol{\Theta}, t$) |
|---|---|
| 1: $\{\beta_1, \beta_2, \epsilon\} \leftarrow \boldsymbol{\Theta}$ | 1: $\{\beta\} \leftarrow \boldsymbol{\Theta}$ |
| 2: $\mathbf{M}_t \leftarrow \beta_1 \mathbf{M}_{t-1}^Q + \mathbf{G}_t$ if $t > 0$ else $\mathbf{M}_0 = \mathbf{G}_0$ | 2: $\mathbf{M}_t = \beta \mathbf{M}_{t-1}^Q + (1-\beta)\mathbf{G}_t$ if $t > 0$ else $\mathbf{M}_0 = \mathbf{G}_0$ |
| 3: $\mathbf{V}_t \leftarrow \beta_2 \mathbf{V}_{t-1}^Q + \mathbf{G}_t^2$ if $t > 0$ else $\mathbf{V}_0 = \mathbf{G}_0^2$ | 3: $(\mathbf{U}_t, \mathbf{S}_t, \mathbf{V}_t) = \text{SVD}(\mathbf{M}_t)$ |
| 4: **return** $\mathbf{W}_t - \eta_t \mathbf{M}_t / \sqrt{\mathbf{V}_t + \epsilon \mathbf{1}}$ | 4: **return** $\mathbf{W}_t - \eta_t \mathbf{U}_t \mathbf{V}_t^\top$ |

Each error term arises from applying a floating-point quantization operator $\mathcal{Q}$ that satisfies Assumption 3.1. Formally, for any quantized quantity $\mathbf{X}_t$ (e.g., $\mathbf{W}_t, \mathbf{G}_t, \mathbf{M}_t, \mathbf{V}_t$) at iteration $t$, its relative quantization error is defined as the smallest constant $q \geq 0$ such that

$$|[\mathbf{X}_t^Q]_{ij} - [\mathbf{X}_t]_{ij}| \leq q_X |[\mathbf{X}_t]_{ij}|, \qquad \forall t \in 0, \dots, T-1.$$

In particular, we have

$$q_W := \inf\left\{q \geq 0 : |[\mathbf{W}_t^Q]_{ij} - [\mathbf{W}_t]_{ij}| \leq q|[\mathbf{W}_t]_{ij}|, \forall t\right\},$$
$$q_G := \inf\left\{q \geq 0 : |[\mathbf{G}_t^Q]_{ij} - [\mathbf{G}_t]_{ij}| \leq q|[\mathbf{G}_t]_{ij}|, \forall t\right\},$$
$$q_M := \inf\left\{q \geq 0 : |[\mathbf{M}_t^Q]_{ij} - [\mathbf{M}_t]_{ij}| \leq q|[\mathbf{M}_t]_{ij}|, \forall t\right\},$$
$$q_V := \inf\left\{q \geq 0 : |[\mathbf{V}_t^Q]_{ij} - [\mathbf{V}_t]_{ij}| \leq q|[\mathbf{V}_t]]_{ij}|, \forall t\right\}.$$

This framework is more general than most prior theoretical analyses, which typically consider quantization of only a subset of components (e.g., gradients).

**Optimizers.** We focus on two adaptive optimizers: Adam (Kingma, 2014) and Muon (Jordan et al., 2024). Algorithm 1 outlines the general quantized training loop, while Algorithms 2 and 3 detail the specific update rules for Adam[1] and Muon, respectively. Note that the quantization operator $\mathcal{Q}$ can represent any floating-point quantization (e.g., fp32 → bf16 or fp8) satisfying Assumption 3.1.

In the following sections, we will analyze the convergence of quantized Adam and Muon under this framework with relative quantization errors $(q_W, q_G, q_M, q_V)$.

---

[1]Our Algorithm slightly differs from the standard Adam, but will not affect the proof. We provide a detailed discussion in Appendix A.1.

## 4 MAIN RESULTS

We now present our main theoretical results on the convergence of quantized Adam and Muon under the analytical framework in Section 3. We begin by stating the assumptions required for our analysis, followed by the convergence theorems for each optimizer.

**Assumption 4.1** (Unbiased Stochastic Gradient). The stochastic gradient $\nabla f(\mathbf{W}; \boldsymbol{\xi})$ is an unbiased estimator of the true gradient $\nabla F(\mathbf{W})$, i.e., $\mathbb{E}[\nabla f(\mathbf{W}; \boldsymbol{\xi})] = \nabla F(\mathbf{W})$.

**Assumption 4.2** (Stochastic Gradient Bounds). The stochastic gradient $\nabla f(\mathbf{W}; \boldsymbol{\xi})$ satisfies the following bounds depending on the algorithm:

- **Adam**: The stochastic gradient is $\ell_\infty$ uniformly almost surely bounded, i.e., there exists a constant $R > \sqrt{\epsilon}$ (where $\epsilon > 0$ is the stability constant used to simplify the final bounds) such that

$$\|\nabla f(\mathbf{W}; \boldsymbol{\xi})\|_\infty = \max_{i,j} |[\nabla f(\mathbf{W}; \boldsymbol{\xi})]_{ij}| \leq R - \sqrt{\epsilon}, \quad \text{a.s..}$$

- **Muon**: The stochastic gradient has bounded variance, i.e., there exists a constant $\sigma > 0$ such that

$$\mathbb{E}[\|\nabla f(\mathbf{W}; \boldsymbol{\xi}) - \nabla F(\mathbf{W})\|_F^2] \leq \sigma^2.$$

**Assumption 4.3** (Smoothness). The objective function $F : \mathbb{R}^{m \times n} \to \mathbb{R}$ is $L$-smooth, i.e., for any $\mathbf{X}, \mathbf{Y} \in \mathbb{R}^{m \times n}$, we have

$$\|\nabla F(\mathbf{X}) - \nabla F(\mathbf{Y})\|_F \leq L\|\mathbf{X} - \mathbf{Y}\|_F.$$

Assumptions 4.1, 4.2 and 4.3 are standard in the analysis of smooth non-convex stochastic optimization (Zaheer et al., 2018; Chen et al., 2019; Zou et al., 2019; Défossez et al., 2022; Chen et al., 2022; Zhang et al., 2022; Wang et al., 2023). They are usually employed to control the stochastic gradient noise and the local geometry of the objective function.

We remark that a more general $(L_0, L_1)$-smoothness condition (Zhang et al., 2020), which has been adopted in recent analyses of Adam (Li et al., 2023; Wang et al., 2024; Hong & Lin, 2024), allows the smoothness constant to depend on the gradient norm: $\|\nabla F(\mathbf{X}) - \nabla F(\mathbf{Y})\|_F \leq (L_0 + L_1 \|\nabla F(\mathbf{Y})\|_F)\|\mathbf{X} - \mathbf{Y}\|_F$. While this condition can better capture practical deep learning scenarios, since our focus is on characterizing how quantization errors influence the convergence behavior of Adam and Muon, we adopt the standard $L$-smoothness assumption for simplicity. Extending our results to $(L_0, L_1)$-smoothness remains an interesting direction for future work.

Finally, we assume the optimization begins from a controlled initialization:

**Assumption 4.4** (Bounded Initialization). The initial parameter matrix $\mathbf{W}_0$ and its gradient are bounded in Frobenius norm, i.e., $\|\mathbf{W}_0\|_F \leq D$, $\|\nabla F(\mathbf{W}_0)\|_F \leq G$, for some constants $D, G > 0$.

Bounding the initialization ensures that quantization errors remain controlled and their propagation through the optimization iterations can be rigorously analyzed, which is crucial for establishing convergence guarantees under low-precision training.

### 4.1 THEORETICAL RESULTS OF ADAM

We first present the convergence result of Adam under FP quantization in the following theorem.

**Theorem 4.5** (Convergence of Quantized Adam). Suppose Assumptions 3.1, 4.1–4.4 hold. Let $d = mn$ be the number of trainable parameters, consider the Quantized Adam algorithm defined in 1 run for $T$ iterations with $\eta_t = (1 - \beta_1)\Omega_t \eta$, where $\Omega_t = \sqrt{\sum_{j=0}^{t-1} \beta_2^j}$. Suppose $\beta_1^2(1 + q_M)^2 < \beta_2(1 - q_V)$, $\beta_1(1 + q_M) < \beta_2(1 - q_V)$, and $2\beta_1/(1 - \beta_1) \leq T$, then for an iteration index $\tau$ chosen randomly from $\{0, \ldots, T-1\}$ with $P(\tau = j) \propto (1 - \beta_1^{T-j})$, we have:

$$\mathbb{E}\left[\|\nabla F(\mathbf{W}_\tau)\|_F^2\right] \leq 4(1 + q_G)R \frac{F_0 - F_*}{\eta T} + \frac{\widetilde{Q}(T)}{T} + \frac{2q_W T \eta \cdot (1 - \beta_1)d^{\frac{3}{2}}\eta L(1 + q_G)R^2}{\sqrt{\epsilon}(1 - \beta_2)\sqrt{1 - \frac{\beta_1^2(1+q_M)^2}{\beta_2(1-q_V)}}}$$

$$+ \frac{4(1 + q_G)d}{\sqrt{\epsilon}(1 - \beta_2)}\left(q_G R^3 + L q_W R^2 D\right) + \frac{C}{T}\left(\ln\left(1 + \frac{((1 + q_G)R)^2}{\epsilon(1 - \beta_2(1 - q_V))}\right) - T \ln(\beta_2(1 - q_V))\right),$$

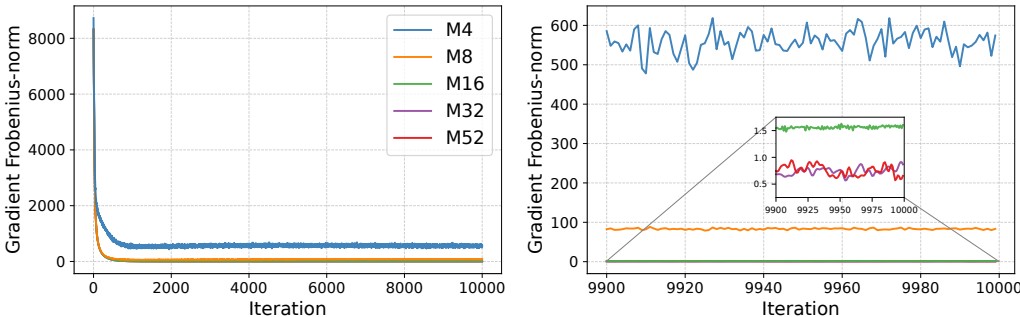

Figure 3: Rosenbrock: Adam gradient norms under different mantissa precisions $M$ (left: full 10,000 iterations; right: last 100 iterations). Larger mantissa bit-lengths yield smaller converged gradient norms. Together with Figure 5, this shows that higher precision reduces quantization error and improves convergence, consistent with Theorem 4.5.

where $C$ is a constant depending on the problem hyperparameters, and $\widetilde{Q}(T)$ is a function with respect to $T$, $q_V$, $q_G$, and $q_M$, which approaches zero when $q_V, q_M \to 0$ (please refer to Eq A.43 for their detailed formula).

Moreover, by setting $\eta = \Theta(1/\sqrt{T})$, $1 - \beta_2 = \Theta(1/T)$, $q_G = \mathcal{O}(1/T)$, $q_M = \mathcal{O}(1/T)$, $q_V = \mathcal{O}(1/T^2)$, $q_W = \mathcal{O}(1/T^2)$, then $\widetilde{Q}(T)/T = \mathcal{O}(T^{-1/2})$ (refer to Eq. A.44 for calculation details) and

$$\mathbb{E}[\|\nabla F(\mathbf{W}_\tau)\|_F] = \widetilde{\mathcal{O}}\left(T^{-1/4}\right).$$

Theorem 4.5 provides the first convergence guarantee for Adam under a practical floating-point quantization model (Peng et al., 2023; Liu et al., 2024; Fishman et al., 2025), in contrast to prior works that assume unbiased quantization or error-feedback mechanisms (Jiang & Agrawal, 2018; Chen et al., 2021; Modoranu et al., 2024). The most similar prior theoretical work is Ozkara et al. (2025), whose analysis also builds on Défossez et al. (2022); however, they consider only quantization of the final weight update, assuming full-precision gradients and optimizer states with $\beta_1 = 0$, and thus do not capture the recursive error propagation arising from fully quantized Adam under realistic floating-point rounding. Our analysis demonstrates that by setting the hyperparameters as $\eta = \Theta(1/\sqrt{T})$ and $1 - \beta_2 = \Theta(1/T)$, and ensuring the relative quantization errors satisfy $q_G, q_M = \mathcal{O}(1/T)$ and $q_W, q_V = \mathcal{O}(1/T^2)$, Quantized Adam achieves a convergence rate of $\widetilde{\mathcal{O}}(T^{-1/4})$, which successfully matches the established one for its full-precision counterpart in smooth non-convex optimization (Guo et al., 2021; Défossez et al., 2022; Wang et al., 2023; Hong & Lin, 2024).

Our theorem further reveals a nuanced sensitivity to different types of quantization error. The required precision for the second moment ($q_V$) is stricter than for the first moment ($q_M$). This sensitivity arises because accumulated errors in the second-moment estimate $\mathbf{V}_t$ are non-linearly amplified by the update step's inverse square root. This theoretical finding provides a rigorous explanation for the empirical observation that the second moment often require higher precision than the first moment in low-bit training setups (Peng et al., 2023; Yu et al., 2024; Fishman et al., 2025). Similarly, the stricter precision requirement for weights ($q_W = \mathcal{O}(1/T^2)$) is necessary to control error accumulation over the entire training trajectory. Our analysis must account for the potential growth of weight magnitudes throughout training, which acts as an amplification factor for the relative quantization error. To guarantee convergence under this worst-case scenario of unbounded weight growth, the proof requires $q_W$ to decay rapidly to counteract this amplification. However, this strict condition is a consequence of the proof's generality. In practice, where weight norms often remain bounded, this error amplification is less severe, and the precision requirement for $q_W$ could be relaxed to $\mathcal{O}(1/T)$.

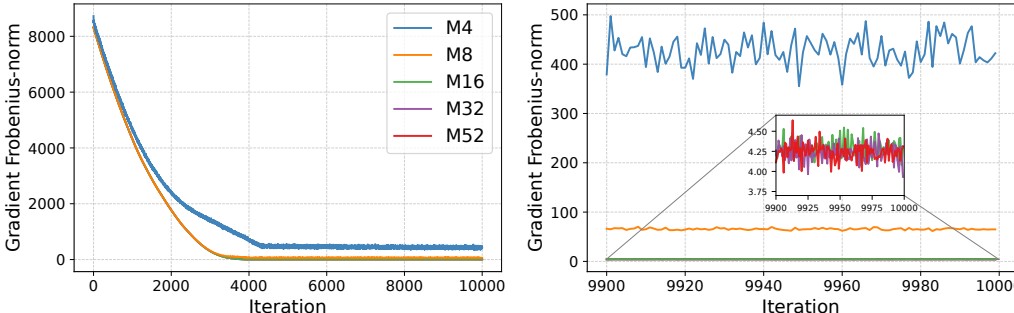

Figure 4: Rosenbrock: Muon gradient norms under different mantissa precisions $M$ (left: full 10,000 iterations; right: last 100 iterations). Larger mantissa bit-lengths yield smaller converged gradient norms. Together with Figure 6, this shows that higher precision reduces quantization error and improves convergence, consistent with Theorem 4.6.

## 4.2 THEORETICAL RESULTS OF MUON

Then, we present the convergence result of quantized Muon in the following theorem.

**Theorem 4.6** (Convergence of Quantized Muon). Suppose Assumptions 3.1, 4.1–4.4 hold. Consider the Quantized Muon algorithm in 1 and 3 run for $T$ iterations with $\eta_t = \eta, \beta(1 + q_M) < 1$, then

$$\frac{1}{T} \sum_{t=0}^{T-1} \mathbb{E}[\|\nabla F(\mathbf{W}_t)\|_F] \leq \frac{\mathbb{E}[F(\mathbf{W}_0) - F(\mathbf{W}_T)]}{\eta T} + \frac{2\beta L \eta r}{1 - \beta} + \frac{6\sigma\sqrt{r}}{T(1-\beta)\sqrt{B}} + \sqrt{\frac{1-\beta}{1+\beta}} \frac{6\sigma\sqrt{r}}{\sqrt{B}} +$$

$$\frac{L\eta r}{2} + C_2 \cdot \left( q_G + q_W + q_G T\eta + q_W T\eta + \frac{q_M \beta}{1 - \beta(1 + q_M)}(1 + T\eta) \right),$$

where $C_2$ is absolute constant, $r = \min\{m, n\}$. Moreover, suppose $F(\mathbf{W}_0) - F^* \leq \Delta$ for constant $\Delta > 0$, set $1 - \beta = \Theta(T^{-1/2})$, $\eta = \Theta(T^{-3/4})$, and $B = 1$, if $q_G = q_W = q_M = \mathcal{O}(T^{-1/2})$, then

$$\frac{1}{T} \sum_{t=0}^{T-1} \mathbb{E}[\|\nabla F(\mathbf{W}_t)\|_F] = \mathcal{O}(T^{-1/4}).$$

Theorem 4.6 establishes the convergence of Quantized Muon under relative quantization errors $(q_W, q_G, q_M)$ for weights, gradients, and momentum, respectively—a practical setting for low-precision training (Peng et al., 2023; Liu et al., 2024; Fishman et al., 2025), in contrast to prior works that assume unbiased quantization or error-feedback mechanisms (Jiang & Agrawal, 2018; Chen et al., 2021; Modoranu et al., 2024). As a sanity check, when $q_W = q_G = q_M = 0$, our result recovers the exact convergence rate $\mathcal{O}(1/T^{1/4})$ of Shen et al. (2025) up to constant factors. More importantly, as long as the mantissa length of the floating-point format scales logarithmically with $T$, i.e., $M = \Omega(\log T)$, the quantization errors decay as $q_W = q_G = q_M = \mathcal{O}(T^{-1/2})$. With appropriate choices of $\eta$ and $\beta$ (as in Theorem 4.6), the full-precision convergence rate $\mathcal{O}(T^{-1/4})$ is preserved.

Finally, we highlight a sharp contrast with quantized Adam. Theorem 4.6 requires only relative errors on the order of $q = \mathcal{O}(T^{-1/2})$, whereas Theorem 4.5 demands stricter conditions, at least $q = \mathcal{O}(T^{-1})$ and in some cases $q = \mathcal{O}(T^{-2})$. This theoretical distinction explains why Muon adapts more efficiently to low-precision settings than Adam, corroborating the empirical observations of Liu et al. (2025).

**Experiments.** We evaluate our theory on synthetic, image, and LLM benchmarks.

**Synthetic setup.** For the synthetic benchmark, we adopt the classical Rosenbrock function (Rosenbrock, 1960). Let $\mathbf{W} \in \mathbb{R}^{m \times n}$, and define $F(\mathbf{W}) = \sum_{j=1}^{n-1} \left( 100\|\mathbf{W}_{j+1} - \mathbf{W}_j^2\|_F^2 + \|\mathbf{1}_m - \mathbf{W}_j\|_F^2 \right)$, where $\mathbf{W}_j$ denotes the $j$-th column of $\mathbf{W}$ and $\mathbf{1}_m$ is the $m$-dimensional all-ones vector. We set $m = 50$, $n = 100$, and run $T = 10,000$ iterations with learning rate $\eta = 5 \times 10^{-4}$. Mantissa

bit-lengths are selected from $M = 4, 8, 16, 24, 32, 52$ to quantize gradients, weights, and optimizer states. For Adam, we use $\beta_1 = 0.9$, $\beta_2 = 0.999$, and $\epsilon = 10^{-8}$; for Muon, we set $\beta = 0.9$ and employ $n_s = 10$ power iterations, following Jordan et al. (2024).

**CIFAR-10 setup.** We train a 4-layer fully connected network $[\mathrm{FC}(512) - \mathrm{ReLU} - \mathrm{FC}(256) - \mathrm{ReLU} - \mathrm{FC}(64) - \mathrm{ReLU} - \mathrm{FC}(10)]$ on CIFAR-10 using Adam and Muon. Additional implementation details are provided in Appendix C.

**nanoGPT setup.** We train `nanoGPT` on OpenWebText ($\sim$ 26M parameters, 4 layers, 4 heads, embedding 384, batch size 128, block size 512). Both AdamW and Muon are tested under varying mantissa lengths $M$, with Muon applied to 2D parameters in transformer blocks and AdamW applied to all 1D parameters (embedding, lm_head, layernorm).

**Empirical Validation of Theory.** Across all benchmarks, our results empirically validate Theorems 4.5 and 4.6. We observe a direct link between quantization error and convergence. As shown across the Rosenbrock, CIFAR-10, and nanoGPT experiments (Figures 3, 4, 8, 9, 12, 13), very low mantissa lengths ($M$) lead to significant convergence degradation. This degradation correlates directly with high relative quantization errors (detailed in Appendix Figures 5, 6, 10, 11), which stall the optimization. Conversely, moderate $M$ values yield sufficiently small errors, enabling convergence nearly identical to the full-precision baseline. Furthermore, our experiment in Figure 7 explicitly confirms our analysis of Adam, showing that optimizer sensitivity to quantization error increases significantly as $\beta_2 \to 1$. The language modeling results in Figure 13 suggest that Muon is more robust than AdamW under low-precision training, consistent with Theorems 4.5 and 4.6. We provide full experimental details and results in Appendix C.

## 5 CONCLUSION AND LIMITATIONS

We introduced the first theoretical framework for analyzing adaptive optimizers under realistic floating-point quantization, jointly modeling the quantization of gradients, parameters, and optimizer states. Unlike prior work, our analysis does not rely on unbiased quantization or error feedback—assumptions that are impractical in modern large-scale low-precision training. Within this framework, we derived the first convergence guarantees for Adam and Muon, with rates expressed explicitly in terms of component-wise quantization errors. Our results highlight that Adam is highly sensitive to parameter and second-moment quantization due to its reliance on $\beta_2 \to 1$, whereas Muon requires weaker error control and is therefore more robust. These findings explain empirical observations in large-scale LLM training and narrow the gap between practice and theory.

**Limitations and Future Directions.** Several challenges remain. First, our analysis focuses on smooth unconstrained non-convex objectives, leaving open extensions to broader settings, including $(L_0, L_1)$-smooth functions (Zhang et al., 2020), non-smooth convex objectives (Mishchenko & Defazio, 2023; Defazio et al., 2024), constrained or composite problems (Kovalev, 2025; Pethick et al., 2025), and structured scenarios studied in recent works (Shen et al., 2025). Second, our theoretical guarantees assume an increasing-bit regime, $M = \Omega(\log T)$, to control cumulative quantization error. In practice, bit-width is typically fixed (e.g., FP8 or BF16), which means convergence is guaranteed only to a neighborhood of a stationary point; understanding why moderate fixed precision suffices empirically remains an open question. Third, we focus primarily on fully quantized Adam/Muon and have not yet extended the framework to other popular optimizers benchmarked in LLM training (Vlassis et al., 2025; Semenov et al., 2025; Wen et al., 2025). Finally, our analysis models quantized states under exact arithmetic and does not account for practical considerations such as low-precision operations (e.g., FP8 matrix multiplications) or communication-efficient distributed training, which are critical for large-scale training. Incorporating these aspects would provide a more complete theoretical account of large-scale low-precision optimization.

## ACKNOWLEDGMENTS

We would like to thank the anonymous reviewers and area chairs for their helpful comments. We acknowledge the support from NSFC 62306252, Hong Kong ECS award 27309624, Guangdong NSF 2024A1515012444, and the central fund from HKU IDS.

## ETHICS STATEMENT

We have carefully reviewed the ICLR Code of Ethics and affirm that our work does not raise any significant ethical concerns. Our research is purely theoretical and experimental within the scope of optimization and quantization. It does not involve human subjects, personally identifiable or sensitive data, or applications that may pose harm. All experiments are conducted on synthetically generated datasets and standard benchmark datasets (e.g., CIFAR-10) and are intended solely to validate the theoretical analysis. We believe our methodology and contributions adhere to principles of fairness, transparency, and research integrity.

## REPRODUCIBILITY STATEMENT

We have made significant efforts to ensure the reproducibility of our work. All theoretical results are fully detailed, with complete proofs provided in Appendix A and Appendix B. The experimental setup, including training protocols, hyperparameters, and evaluation details, is comprehensively documented in Appendix C. Experiments are conducted on both synthetically generated datasets and the CIFAR-10 benchmark dataset; for synthetic datasets, precise generation procedures are included to eliminate ambiguity. Together, these details allow independent researchers to reproduce both the theoretical and experimental results that support the main conclusions of the paper.

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

USE OF LARGE LANGUAGE MODELS

**Polishing writing.** We used multiple large language models (LLMs) to polish the presentation of the paper, focusing on grammar, fluency, and readability while preserving the technical meaning of the content. For each passage, we generated outputs from several LLMs and selected the best version based on clarity and accuracy. The LLMs served only as editorial assistants, and all suggested outputs were carefully checked and revised by the authors. The prompt used for polishing is as follows:

```
I am preparing a paper for ICLR in Optimization.
Please help me polish the following
[sentence/paragraph/section] to make it more logical,
precise, clear, and accurate, while preserving the
technical meaning and mathematical correctness.  Focus
on improving sentence structure, clarity, flow, and
readability, and enhance logical coherence between
statements.  Highlight any ambiguities or imprecise
statements and suggest more rigorous alternatives.
[sentence/paragraph/section]
```

**Assisting LATEX code.** We also used github copilot/cursor as a typing assistant to conveniently type LATEX code for mathematical formulas and derivations. All generated code was manually checked, corrected, and integrated by the authors.

# A  PROOF OF THEOREM 4.5

## A.1  PRELIMINARIES

We consider an optimization problem in a $d$-dimensional space (let $d = mn$ be the number of trainable parameters), where coordinates are indexed by $i \in [d] = \{1, 2, \ldots, d\}$. Our algorithm generates a sequence of vectors $(\mathbf{u}_t)_{t \in \mathbb{N}}$, with the $i$-th component of $\mathbf{u}_t$ denoted by $u_{t,i}$. The objective is to find a critical point of a global function $F : \mathbb{R}^d \to \mathbb{R}$ within a stochastic framework, where we have access to a sequence of i.i.d. sample functions $(f_t)_{t \in \mathbb{N}^*}$ (e.g., the loss on a data minibatch). For any differentiable function $h : \mathbb{R}^d \to \mathbb{R}$, we denote its gradient by $\nabla h$ and its $i$-th component by $\nabla_i h$. Finally, we use a small constant $\epsilon > 0$ for numerical stability and let $\mathbb{E}_t[\cdot]$ denote the conditional expectation given the history of samples $f_1, \ldots, f_{t-1}$. We use $\text{vec}(\cdot)$ to vectorize a matrix and $\text{mat}(\cdot)$ for the inverse operation.

Recall the dynamic system of our theoretical Quantized Adam. In the proof, we denote $\mathbf{w}_t = \text{vec}(\mathbf{W}_t)$, $\widehat{\mathbf{g}}_t = \text{vec}(\mathbf{G}_t)$ and the dimension $d = m \cdot n$. For an iteration $t \in \mathbb{N}^*$, we define:

$$\begin{cases} m_{t,i} & = \beta_1 m_{t-1,i}^Q + \widehat{g}_{t,i} = \beta_1(m_{t-1,i} + \xi_{t-1,i}) + (\nabla_i f_t(\mathbf{w}_{t-1}) + \delta_{t,i}), \\ v_{t,i} & = \beta_2 v_{t-1,i}^Q + \widehat{g}_{t,i}^2 = \beta_2(v_{t-1,i} + \theta_{t-1,i}) + (\nabla_i f_t(\mathbf{w}_{t-1}) + \delta_{t,i})^2, \\ w_{t,i} & = w_{t-1,i} - \eta_t \frac{m_{t,i}}{\sqrt{\epsilon + v_{t,i}}}. \end{cases} \quad \text{(A.1)}$$

with the step size given by

$$\eta_t = \eta(1 - \beta_1)\sqrt{\frac{1 - \beta_2^t}{1 - \beta_2}}. \quad \text{(A.2)}$$

Here, $\delta_{t,i}$, $\xi_{t-1,i}$, and $\theta_{t-1,i}$ represent the quantization errors for the gradient, first moment, and second moment, respectively. Especially, $\nabla_i f_t(\mathbf{w}_{t-1}) = \frac{1}{B}\sum_{j=1}^{B} \nabla_i f(\mathbf{w}_{t-1}; \gamma_{t,j})$, $\delta_{t,i} = \frac{1}{B}\sum_{j=1}^{B} \nabla_i^Q f(\mathbf{w}_{t-1}^Q; \gamma_{t,j}) - \frac{1}{B}\sum_{j=1}^{B} \nabla_i f_t(\mathbf{w}_{t-1}; \gamma_{t,j})$. And we have $\mathbb{E}\left[\nabla_i f_t(\mathbf{w}_{t-1})\right] = \nabla_i F(\mathbf{w}_{t-1})$

Our convergence analysis for Quantized Adam is predicated on a specific, analytically convenient formulation of the algorithm. This section serves to rigorously justify our theoretical framework

by establishing two foundational equivalences. First, we demonstrate that our representation of the Adam update is equivalent to the standard formulation. Following the methodology of Défossez et al. (2022), we absorb the scaling factor into the learning rate, which simplifies the recursive structure of the momentum term. Second, and more critically for our work, we prove that the theoretical analysis of quantizing these weighted-sum states is directly applicable to the practical scenario of quantizing the standard weighted-average states.

**Equivalence with Standard Adam** Our formulation in (A.1) utilizes a weighted sum for the moments, which differs slightly from the standard weighted-average approach in the original Adam algorithm (Kingma, 2014). The standard first moment, often expressed as $\widetilde{m}_{t,i} = (1 - \beta_1) \sum_{k=1}^{t} \beta_1^{t-k} \widehat{g}_{k,i}$, is simply a scaled version of our definition, i.e., $\widetilde{m}_{t,i} = (1 - \beta_1)m_{t,i}$. This constant scaling factor can be directly absorbed into the learning rate.

Furthermore, the standard Adam algorithm includes bias correction terms to counteract the zero-initialization of moments. These corrections are equivalent to using a time-dependent step size of the form:

$$\eta_{t,\text{Adam}} = \eta \cdot \frac{1 - \beta_1}{\sqrt{1 - \beta_2}} \cdot \frac{\sqrt{1 - \beta_2^t}}{1 - \beta_1^t}. \tag{A.3}$$

For analytical tractability, our analysis adopts the simplified step size $\eta_t$ from (A.2), which omits the bias correction for the first moment ($m_{t,i}$). This simplification is motivated by several practical and theoretical considerations. First, it ensures that our step size $\eta_t$ is monotonic with respect to $t$, which is advantageous for the convergence proof. Second, for typical hyperparameter values (e.g., $\beta_1 = 0.9$, $\beta_2 = 0.999$), the omitted term $1/(1 - \beta_1^t)$ converges to its limit of 1 much more rapidly than the retained term $\sqrt{1 - \beta_2^t}$. Finally, removing this term effectively implements a learning rate warm-up, a common and beneficial practice, while retaining the correction for $v_{t,i}$ prevents an undesirably large initial step size that could lead to training instability.

**Equivalence of Quantization Schemes.** A subtle but crucial aspect of our setup is the object of quantization. Our theoretical framework analyzes the quantization of weighted-sum moments $(\mathbf{m}_t, \mathbf{v}_t)$, while a practical implementation would quantize the standard weighted-average moments $(\widetilde{\mathbf{m}}_t, \widetilde{\mathbf{v}}_t)$. We now prove that these two approaches are, in fact, analytically equivalent.

To establish this rigorously, we first abstract the core dynamic behavior into a general mathematical lemma. We will show that two discrete-time systems, representing the weighted-sum and weighted-average accumulation methods under relative error perturbations, are analytically indistinguishable. We will then apply this result to the specific case of Quantized Adam.

**Lemma A.1** (Equivalence of Perturbed Dynamical Systems)**.** Consider two scalar sequences $\{a_k\}_{k \geq 0}$ and $\{c_k\}_{k \geq 0}$ evolving according to the following dynamics for $k \geq 1$, with initial conditions $a_0 = c_0 = 0$:

$$a_k = \beta(a_{k-1} + d_{k-1}) + b_k \tag{A.4}$$
$$c_k = \beta(c_{k-1} + e_{k-1}) + (1 - \beta)b_k \tag{A.5}$$

where $\beta \in (0, 1)$ is a decay factor, $\{b_k\}$ is an external input sequence, and $\{d_k\}$, $\{e_k\}$ are perturbation sequences. These perturbations are bounded by a relative error model with factor $q \in [0, 1)$:

$$|d_{k-1}| \leq q|a_{k-1}| \quad \text{and} \quad |e_{k-1}| \leq q|c_{k-1}|. \tag{A.6}$$

Then, the sequence $\{c_k\}$ and the scaled sequence $\{a'_k\} \triangleq (1 - \beta)a_k$ are analytically equivalent. Specifically, they follow identical recurrence relations where their respective perturbation terms satisfy identical relative error bounds with respect to their own states.

*Proof.* To prove the equivalence, we derive the recurrence relation for the scaled sequence $\{a'_k\}$ and compare its structure and error properties to that of $\{c_k\}$.

**Step 1: Derive the recurrence for $\{a'_k\}$.** Starting from the definition $a'_k = (1 - \beta)a_k$ and substituting the dynamics from (A.4):

$$a'_k = (1 - \beta)\left[\beta(a_{k-1} + d_{k-1}) + b_k\right]$$
$$= \beta(1 - \beta)a_{k-1} + \beta(1 - \beta)d_{k-1} + (1 - \beta)b_k.$$

Now, we replace $a_{k-1}$ with $a'_{k-1}/(1-\beta)$:

$$a'_k = \beta(1-\beta)\left(\frac{a'_{k-1}}{1-\beta}\right) + \beta(1-\beta)d_{k-1} + (1-\beta)b_k$$
$$= \beta a'_{k-1} + \beta(1-\beta)d_{k-1} + (1-\beta)b_k.$$

**Step 2: Compare recurrence structures.** Let us place the recurrence relations for $\{c_k\}$ and $\{a'_k\}$ side-by-side:

$$c_k = \beta c_{k-1} + \beta e_{k-1} + (1-\beta)b_k$$
$$a'_k = \beta a'_{k-1} + \beta(1-\beta)d_{k-1} + (1-\beta)b_k.$$

Both sequences share the identical structure: $X_k = \beta X_{k-1} + \text{Perturbation}_k + (1-\beta)b_k$. The only difference lies in the form of their respective perturbation terms.

**Step 3: Compare relative bounds of the perturbation terms.** The equivalence hinges on whether these different perturbation terms satisfy the same relative error property with respect to their own system's state.

For system $\{c_k\}$, the perturbation term is $\beta e_{k-1}$. Using the bound from (A.6):

$$|\text{Perturbation}_c| = |\beta e_{k-1}| = \beta|e_{k-1}| \le \beta q|c_{k-1}|.$$

For system $\{a'_k\}$, the effective perturbation term is $\beta(1-\beta)d_{k-1}$. Using the bound from (A.6) and the scaling relationship $a_{k-1} = a'_{k-1}/(1-\beta)$:

$$|\text{Perturbation}_a| = |\beta(1-\beta)d_{k-1}| = \beta(1-\beta)|d_{k-1}| \le \beta(1-\beta)q|a_{k-1}| = \beta(1-\beta)q\frac{|a'_{k-1}|}{1-\beta} = \beta q|a'_{k-1}|.$$

**Conclusion of Proof.** Both systems, $\{c_k\}$ and the scaled $\{a'_k\}$, adhere to the same mathematical dynamics. Their evolution is governed by an identical recurrence structure, and their respective perturbation terms are bounded by the exact same relative factor $\beta q$ with respect to their own previous state. Therefore, from an analytical standpoint, the two systems are indistinguishable. Any conclusion regarding the long-term behavior (e.g., convergence, stability) of $\{c_k\}$ under its perturbation model will apply directly to $\{a'_k\}$ (and thus proportionally to $\{a_k\}$) under its own perturbation model. $\qquad\square$

With Lemma A.1 established, we can now apply this general result to our specific case of Quantized Adam. The weighted-sum moment $\mathbf{m}_t$ from our analysis corresponds to the abstract sequence $\{a_k\}$, while the standard weighted-average moment $\widetilde{\mathbf{m}}_t$ corresponds to $\{c_k\}$. The gradient term $\mathbf{g}_t$ corresponds to $\{b_k\}$, and the relative quantization errors in both schemes are modeled by the perturbations $\{d_k\}$ and $\{e_k\}$ with the bound factor $q$.

Lemma A.1 thus formally proves that analyzing the quantization of our weighted-sum moment $\mathbf{m}_t$ is analytically equivalent to analyzing the quantization of the standard weighted-average moment $\widetilde{\mathbf{m}}_t$. This rigorously justifies our proof strategy and ensures that our theoretical findings are directly relevant to practical implementations of Quantized Adam. A parallel argument holds for the second moments $\mathbf{v}_t$ and $\widetilde{\mathbf{v}}_t$ with decay factor $\beta_2$.

## A.2 DETAILED PROOF

To systematically analyze the effects of quantization, we begin by isolating the different sources of error. We introduce auxiliary moment estimates, $m'_{t,i}$ and $v'_{t,i}$, which are defined to incorporate the quantization error from the stochastic gradient, $\delta_{t,i}$, but are themselves assumed to be stored with perfect precision. Their dynamics are given by:

$$m'_{t,i} = \beta_1 m'_{t-1,i} + (\nabla_i f_t(\mathbf{w}_{t-1}) + \delta_{t,i})$$
$$v'_{t,i} = \beta_2 v'_{t-1,i} + (\nabla_i f_t(\mathbf{w}_{t-1}) + \delta_{t,i})^2$$

(A.7)

Throughout the following proof we note $\mathbb{E}_{t-1}[\cdot]$ the conditional expectation with respect to $f_1, \ldots, f_{t-1}$. In particular, $\mathbf{w}_{t-1}, \mathbf{v}_{t-1}$ is deterministic knowing $f_1, \ldots, f_{t-1}$. With slightly abuse of notation in the detailed proof, We introduce

$$\mathcal{G}_t = \nabla F(\mathbf{w}_{t-1}) \quad \text{and} \quad \mathbf{g}_t = \nabla f_t(\mathbf{w}_{t-1}).$$

(A.8)

We introduce the update $\mathbf{u}_t \in \mathbb{R}^d$, as well as the update without momentum $\mathbf{U}_t \in \mathbb{R}^d$:

$$u_{t,i} = \frac{m_{t,i}}{\sqrt{\epsilon + v_{t,i}}} \quad \text{and} \quad U_{t,i} = \frac{g_{t,i} + \delta_{t,i}}{\sqrt{\epsilon + v'_{t,i}}}. \tag{A.9}$$

For any $k \in \mathbb{N}$ with $k < t$, we define $\widetilde{v}_{t,k} \in \mathbb{R}^d$ by

$$\widetilde{v}_{t,k,i} = \beta_2^k v'_{t-k,i} + \mathbb{E}_{t-k}\left[\sum_{j=t-k+1}^{t} \beta_2^{t-j}(g_{j,i} + \delta_{j,i})^2\right], \tag{A.10}$$

i.e. the contribution from the $k$ last gradients are replaced by their expected value for know values of $f_1, \ldots, f_{t-k-1}$.

Using the smoothness of $F$, we have

$$F(\mathbf{w}_t) \leq F(\mathbf{w}_{t-1}) - \eta_t \mathcal{G}_t^T \mathbf{u}_t + \frac{\eta_t^2 L}{2}\|\mathbf{u}_t\|_2^2.$$

The overall motivation of our proof is to find a lower bound for $\eta_t \mathcal{G}_t^T \mathbf{u}_t$.

$$\mathcal{G}_t^T \mathbf{u}_t = \sum_{i \in [d]} \mathcal{G}_{t,i} \frac{m_{t,i}}{\sqrt{\epsilon + v_{t,i}}} \tag{A.11}$$

We can rewrite $\mathcal{G}_t^T \mathbf{u}_t$ as:

$$\sum_{i \in [d]} \mathcal{G}_{t,i} \frac{m_{t,i}}{\sqrt{\epsilon + v_{t,i}}} = \underbrace{\left(\sum_{i \in [d]} \mathcal{G}_{t,i} \frac{m_{t,i}}{\sqrt{\epsilon + v_{t,i}}} - \sum_{i \in [d]} \mathcal{G}_{t,i} \frac{m'_{t,i}}{\sqrt{\epsilon + v'_{t,i}}}\right)}_{A} + \underbrace{\sum_{i \in [d]} \mathcal{G}_{t,i} \frac{m'_{t,i}}{\sqrt{\epsilon + v'_{t,i}}}}_{B} \tag{A.12}$$

Here, Term A represents the error component arising from the quantization of the momentum accumulators $(\mathbf{m}_t, \mathbf{v}_t)$, while Term B represents the behavior of the update driven by an ideal accumulator.

Now we can bound term A. We split A into two parts as before:

$$|A| \leq \underbrace{\left|\sum_{i \in [d]} \mathcal{G}_{t,i} \frac{m_{t,i} - m'_{t,i}}{\sqrt{\epsilon + v_{t,i}}}\right|}_{A_1} + \underbrace{\left|\sum_{i \in [d]} \mathcal{G}_{t,i}\left(\frac{m'_{t,i}}{\sqrt{\epsilon + v_{t,i}}} - \frac{m'_{t,i}}{\sqrt{\epsilon + v'_{t,i}}}\right)\right|}_{A_2} \tag{A.13}$$

The first term, $A_1$, which arises from the quantization noise on the first moment $\mathbf{m}$, is bounded using Lemma A.3 (with $q = q_M$) and Lemma A.5:

$$|A_1| \leq \sum_{i \in [d]} R \frac{|m_{t,i} - m'_{t,i}|}{\sqrt{v_{t,i}}}$$

$$\leq R \sum_{i \in [d]} \frac{\sum_{k=0}^{t}(\beta_1^{t-k}((1+q_M)^{t-k} - 1))|\nabla_i f_k(\mathbf{w}_{k-1}) + \delta_{k,i}|}{\sqrt{\sum_{k=0}^{t} \beta_2^{t-k}(1 - q_V)^{t-k}(\nabla_i f_k(\mathbf{w}_{k-1}) + \delta_{k,i})^2}} \leq q_M \cdot dR \cdot C_q \tag{A.14}$$

where $C_q = \frac{\sqrt{r'(1+r')}}{(1+q_M)(1-r')^{3/2}}$ and $r' = \frac{\beta_1^2(1+q_M)^2}{\beta_2(1-q_V)}$.

For the second term, $A_2$, we use the bound on the gradient $\|\mathcal{G}_t\|_\infty \leq R$ and apply Lemma A.2, which requires a case analysis.

$$|A_2| \leq \sum_{i \in [d]} R|m'_{t,i}|\left|\frac{1}{\sqrt{\epsilon + v_{t,i}}} - \frac{1}{\sqrt{\epsilon + v'_{t,i}}}\right|$$

$$\leq \sum_{i \in [d]} R |m'_{t,i}| \max \left\{ \frac{1}{\sqrt{\epsilon + LB_{t,i}}} - \frac{1}{\sqrt{\epsilon + v'_{t,i}}}, \frac{1}{\sqrt{\epsilon + v'_{t,i}}} - \frac{1}{\sqrt{\epsilon + UB_{t,i}}} \right\} \tag{A.15}$$

Let $g_{k,i} = \nabla_i f_k(\mathbf{w}_{k-1}) + \delta_{k,i}$. We analyze the two terms inside the $\max$ for a single coordinate $i$.

**Case I (Deviation from Lower Bound):** Following the approximation in the provided sketch, we have

$$|m'_{t,i}| \left( \frac{1}{\sqrt{\epsilon + LB_{t,i}}} - \frac{1}{\sqrt{\epsilon + v'_{t,i}}} \right) = |m'_{t,i}| \left( \frac{v'_{t,i} - LB_{t,i}}{\sqrt{\epsilon + LB_{t,i}}\sqrt{\epsilon + v'_{t,i}}(\sqrt{\epsilon + LB_{t,i}} + \sqrt{\epsilon + v'_{t,i}})} \right)$$

$$= \frac{|m'_{t,i}|}{\sqrt{\epsilon + LB_{t,i}}} \frac{v'_{t,i} - LB_{t,i}}{\sqrt{\epsilon + v'_{t,i}}(\sqrt{\epsilon + LB_{t,i}} + \sqrt{\epsilon + v'_{t,i}})}$$

$$\leq \frac{|m'_{t,i}|}{\sqrt{\epsilon + LB_{t,i}}} \frac{v'_{t,i} - LB_{t,i}}{\epsilon + v'_{t,i}}$$

$$= \frac{\sum_{k=0}^t \beta_1^{t-k} |g_{k,i} + \delta_{k,i}|}{\sqrt{\epsilon + \sum_{k=0}^t (\beta_2(1-q_V))^{t-k}(g_{k,i} + \delta_{k,i})^2}} \cdot \frac{\sum_{k=0}^t (\beta_2^{t-k} - (\beta_2(1-q_V))^{t-k})(g_{k,i} + \delta_{k,i})^2}{\epsilon + \sum_{k=0}^t \beta_2^{t-k}(g_{k,i} + \delta_{k,i})^2} \tag{A.16}$$

The first fraction is bounded by Lemma A.4: $\frac{\sum_{k=0}^t \beta_1^k |a_{t-k,i}|}{\sqrt{\epsilon + \sum_{k=0}^t (\beta_2(1-q_V))^k a_{t-k,i}}} \leq \frac{1}{\sqrt{1 - \beta_1^2/(\beta_2(1-q_V))}}$. The second fraction is a ratio of weighted sums $\frac{\sum_{k=0}^t (\beta_2^k - (\beta_2(1-q_V))^k) a_{t-k,i}}{\epsilon + \sum_{k=0}^t \beta_2^k a_{t-k,i}}$. This ratio is bounded by the maximum ratio of its coefficients:

$$\max_{k \in \{0,\ldots,t\}} \frac{\beta_2^k - (\beta_2(1-q_V))^k}{\beta_2^k} = \max_{k \in \{0,\ldots,t\}} (1 - (1-q_V)^k) = 1 - (1-q_V)^t.$$

Combining these bounds, the term for Case I is bounded by $\frac{1}{\sqrt{1 - \beta_1^2/(\beta_2(1-q_V))}}(1 - (1-q_V)^t)$.

**Case II (Deviation from Upper Bound):** Similarly, we bound the second term:

$$|m'_{t,i}| \left( \frac{1}{\sqrt{\epsilon + v'_{t,i}}} - \frac{1}{\sqrt{\epsilon + UB_{t,i}}} \right) \leq \frac{|m'_{t,i}|}{\sqrt{\epsilon + v'_{t,i}}} \frac{UB_{t,i} - v'_{t,i}}{\epsilon + UB_{t,i}}$$

Applying Lemma A.4 and the maximum ratio:

$$\leq \frac{1}{\sqrt{1 - \beta_1^2/\beta_2}} \cdot \left(1 - (1 + q_V)^{-t}\right)$$

Comparing the bounds from the two cases, the bound from Case I is larger since $1 - (1-q_V)^t > 1 - (1+q_V)^{-t}$ and the denominator term $\sqrt{1 - \beta_1^2/(\beta_2(1-q_V))}$ is smaller than $\sqrt{1 - \beta_1^2/\beta_2}$. Therefore, taking the maximum and summing over $d$ dimensions:

$$|A_2| \leq \sum_{i \in [d]} R \frac{1 - (1-q_V)^t}{\sqrt{1 - \frac{\beta_1^2}{\beta_2(1-q_V)}}} = \frac{dR(1 - (1-q_V)^t)}{\sqrt{1 - \frac{\beta_1^2}{\beta_2(1-q_V)}}} \tag{A.17}$$

Combining the bounds for $|A_1|$ and $|A_2|$, we can get the bound for term A:

$$|A| \leq |A_1| + |A_2| \leq q_M \cdot dR \cdot C_q + \frac{dR(1 - (1-q_V)^t)}{\sqrt{1 - \frac{\beta_1^2}{\beta_2(1-q_V)}}} \triangleq Q(t) \tag{A.18}$$

Now we move on to bound Term B. Let us now focus on bounding Term B from (A.12). By expanding the definition of the first moment estimate $m'_{t,i}$, we can decompose Term B into two parts, which we will call Term C and Term D:

$$\sum_{i \in [d]} \mathcal{G}_{t,i} \frac{m'_{t,i}}{\sqrt{\epsilon + v'_{t,i}}} = \sum_{i \in [d]} \mathcal{G}_{t,i} \sum_{k=0}^{t-1} \beta_1^k \frac{g_{t-k,i} + \delta_{t-k,i}}{\sqrt{\epsilon + v'_{t,i}}} \tag{A.19}$$

$$= \underbrace{\sum_{i \in [d]} \sum_{k=0}^{t-1} \beta_1^k \mathcal{G}_{t-k,i} \frac{g_{t-k,i} + \delta_{t-k,i}}{\sqrt{\epsilon + v'_{t,i}}}}_{C} + \underbrace{\sum_{i \in [d]} \sum_{k=0}^{t-1} \beta_1^k \left(\mathcal{G}_{t,i} - \mathcal{G}_{t-k,i}\right) \frac{g_{t-k,i} + \delta_{t-k,i}}{\sqrt{\epsilon + v'_{t,i}}}}_{D}.$$

(A.20)

The magnitude of Term D, which captures the error from gradient drift, is bounded by Lemma A.8:

$$|D| \leq \frac{\eta_t^2 L^2 \sqrt{1-\beta_1}}{4(1+q_G)R} \left(\sum_{l=1}^{t-1} ||\mathbf{u}_{t-l}||_2^2 \sum_{k=l}^{t-1} \beta_1^k \sqrt{k}\right) + \frac{(1+q_G)R}{\sqrt{1-\beta_1}} \sum_{k=0}^{t-1} \left(\frac{\beta_1}{\beta_2}\right)^k \sqrt{k+1} ||\mathbf{U}_{t-k}||_2^2.$$

(A.21)

For Term C, we establish a lower bound on its expectation in Lemma A.9:

$$\mathbb{E}\left[C\right] \geq \frac{1}{2} \left(\sum_{i \in [d]} \sum_{k=0}^{t-1} \beta_1^k \mathbb{E}\left[\frac{\mathcal{G}_{t-k,i}^2}{\sqrt{\epsilon + \widetilde{v}_{t,k+1,i}}}\right]\right) - \frac{2(1+q_G)R}{\sqrt{1-\beta_1}} \left(\sum_{i \in [d]} \sum_{k=0}^{t-1} \left(\frac{\beta_1}{\beta_2}\right)^k \sqrt{k+1} \mathbb{E}\left[||\mathbf{U}_{t-k}||_2^2\right]\right)$$
$$- d \sum_{k=0}^{t-1} \beta_1^k M_{t-k}.$$

(A.22)

where $M_{t-k} = \frac{q_G R^2 + Lq_W R ||\mathbf{w}_{t-k-1}||_2}{\sqrt{\epsilon}}$. Injecting (A.22), (A.21) and (A.20) into (A.12). We get the final lower bound for $\mathcal{G}_t^T \mathbf{u}_t$:

$$\mathbb{E}\left[\sum_{i \in [d]} \mathcal{G}_{t,i} \frac{m_{t,i}}{\sqrt{\epsilon + v_{t,i}}}\right] \geq \frac{1}{2} \left(\sum_{i \in [d]} \sum_{k=0}^{t-1} \beta_1^k \mathbb{E}\left[\frac{\mathcal{G}_{t-k,i}^2}{\sqrt{\epsilon + \widetilde{v}_{t,k+1,i}}}\right]\right) - Q(t) - d \sum_{k=0}^{t-1} \beta_1^k M_{t-k}$$
$$- \frac{\eta_t^2 L^2}{4(1+q_G)R} \sqrt{1-\beta_1} \left(\sum_{l=1}^{t-1} ||\mathbf{u}_{t-l}||_2^2 \sum_{k=l}^{t-1} \beta_1^k \sqrt{k}\right) - \frac{3(1+q_G)R}{\sqrt{1-\beta_1}} \left(\sum_{k=0}^{t-1} \left(\frac{\beta_1}{\beta_2}\right)^k \sqrt{k+1} ||\mathbf{U}_{t-k}||_2^2\right).$$

(A.23)

Now lets look back at:

$$F(\mathbf{w}_t) \leq F(\mathbf{w}_{t-1}) - \eta_t \mathcal{G}_t^T \mathbf{u}_t + \frac{\eta_t^2 L}{2} ||\mathbf{u}_t||_2^2.$$

inject (A.23) into it:

$$\mathbb{E}\left[F(\mathbf{w}_t)\right] \leq \mathbb{E}\left[F(\mathbf{w}_{t-1})\right] - \frac{\eta_t}{2} \left(\sum_{i \in [d]} \sum_{k=0}^{t-1} \beta_1^k \mathbb{E}\left[\frac{\mathcal{G}_{t-k,i}^2}{\sqrt{\epsilon + \widetilde{v}_{t,k+1,i}}}\right]\right) + \frac{\eta_t^2 L}{2} \mathbb{E}\left[||\mathbf{u}_t||_2^2\right] + \eta_t Q(t) + \eta_t d \sum_{k=0}^{t-1} \beta_1^k M_{t-k}$$
$$+ \frac{\eta_t^3 L^2}{4(1+q_G)R} \sqrt{1-\beta_1} \left(\sum_{l=1}^{t-1} ||\mathbf{u}_{t-l}||_2^2 \sum_{k=l}^{t-1} \beta_1^k \sqrt{k}\right) + \frac{3\eta_t(1+q_G)R}{\sqrt{1-\beta_1}} \left(\sum_{k=0}^{t-1} \left(\frac{\beta_1}{\beta_2}\right)^k \sqrt{k+1} ||\mathbf{U}_{t-k}||_2^2\right).$$

(A.24)

We have for any $k \in \mathbb{N}$, $k < t$, and any coordinate $i \in [d]$, $\sqrt{\epsilon + \widetilde{v}_{t,k+1,i}} \leq (1+q_G)R\sqrt{\sum_{j=0}^{t-1} \beta_2^j}$.
Introducing $\Omega_t = \sqrt{\sum_{j=0}^{t-1} \beta_2^j}$, we have

$$\mathbb{E}\left[F(\mathbf{w}_t)\right] \leq \mathbb{E}\left[F(\mathbf{w}_{t-1})\right] - \frac{\eta_t}{2(1+q_G)R\Omega_t} \sum_{k=0}^{t-1} \beta_1^k \mathbb{E}\left[||\mathcal{G}_{t-k}||_2^2\right] + \frac{\eta_t^2 L}{2} \mathbb{E}\left[||\mathbf{u}_t||_2^2\right] + \eta_t Q(t) + \eta_t d\mathbb{E}\left[\sum_{k=0}^{t-1} \beta_1^k M_{t-k}\right]$$
$$+ \frac{\eta_t^3 L^2}{4(1+q_G)R} \sqrt{1-\beta_1} \left(\sum_{l=1}^{t-1} ||\mathbf{u}_{t-l}||_2^2 \sum_{k=l}^{t-1} \beta_1^k \sqrt{k}\right) + \frac{3\eta_t(1+q_G)R}{\sqrt{1-\beta_1}} \left(\sum_{k=0}^{t-1} \left(\frac{\beta_1}{\beta_2}\right)^k \sqrt{k+1} ||\mathbf{U}_{t-k}||_2^2\right).$$

(A.25)

Now summing over all iterations $t \in [T]$ for $T \in \mathbb{N}^*$, and $\eta_t$ is non-decreasing, as well the fact that $F$ is bounded below by $F_*$, we get

$$\underbrace{\frac{1}{2(1+q_G)R} \sum_{t=1}^{T} \frac{\eta_t}{\Omega_t} \sum_{k=0}^{t-1} \beta_1^k \mathbb{E}\left[||\mathcal{G}_{t-k}||_2^2\right]}_{\widetilde{A}} \leq F(\mathbf{w}_0) - F_* + \underbrace{\frac{\eta_T^2 L}{2} \sum_{t=1}^{T} \mathbb{E}\left[||\mathbf{u}_t||_2^2\right]}_{\widetilde{B}} + \underbrace{\eta_T \sum_{t=1}^{T} Q(t)}_{E_Q} + \underbrace{\eta_T d \sum_{t=1}^{T} \mathbb{E}\left[\sum_{k=0}^{t-1} \beta_1^k M_{t-k}\right]}_{M}$$

$$+ \underbrace{\frac{\eta_T^3 L^2}{4(1+q_G)R} \sqrt{1-\beta_1} \sum_{t=1}^{T} \sum_{l=1}^{t-1} \mathbb{E}\left[||\mathbf{u}_{t-l}||_2^2\right] \sum_{k=l}^{t-1} \beta_1^k \sqrt{k} + \frac{3\eta_T(1+q_G)R}{\sqrt{1-\beta_1}} \sum_{t=1}^{T} \sum_{k=0}^{t-1} \left(\frac{\beta_1}{\beta_2}\right)^k \sqrt{k+1} \mathbb{E}\left[||\mathbf{U}_{t-k}||_2^2\right]}_{\widetilde{D}}.$$

$$\tag{A.26}$$

First lets bound Term $\widetilde{A}$. We have $\eta_t = (1-\beta_1)\Omega_t \eta$. Thus, we can simplify the $\widetilde{A}$ term from (A.26), also using the usual change of index $j = t - k$, to get

$$\widetilde{A} = \frac{1}{2(1+q_G)R} \sum_{t=1}^{T} \frac{\eta_t}{\Omega_t} \sum_{j=1}^{t} \beta_1^{t-j} \mathbb{E}\left[||\mathcal{G}_j||_2^2\right]$$

$$= \frac{\eta(1-\beta_1)}{2(1+q_G)R} \sum_{j=1}^{T} \mathbb{E}\left[||\mathcal{G}_j||_2^2\right] \sum_{t=j}^{T} \beta_1^{t-j}$$

$$= \frac{\eta}{2(1+q_G)R} \sum_{j=1}^{T} (1-\beta_1^{T-j+1}) \mathbb{E}\left[||\mathcal{G}_j||_2^2\right]$$

$$= \frac{\eta}{2(1+q_G)R} \sum_{j=1}^{T} (1-\beta_1^{T-j+1}) \mathbb{E}\left[||\nabla F(\mathbf{w}_{j-1})||_2^2\right]$$

$$= \frac{\eta}{2(1+q_G)R} \sum_{j=0}^{T-1} (1-\beta_1^{T-j}) \mathbb{E}\left[||\nabla F(\mathbf{w}_j)||_2^2\right]. \tag{A.27}$$

To establish our convergence guarantee, we analyze the expected gradient norm at a randomly selected iteration $\tau$, drawn from the set $\{0, \dots, T-1\}$. The selection is not uniform but is instead weighted to properly account for the influence of the momentum term over the iterations. The probability of selecting a specific iteration $t$ is defined as:

$$\forall t \in \{0, \dots, T-1\}, \quad \mathbb{P}(\tau = t) \propto 1 - \beta_1^{T-t}. \tag{A.28}$$

We can notice that

$$\sum_{j=0}^{T-1} (1-\beta_1^{T-j}) = T - \beta_1 \frac{1-\beta_1^T}{1-\beta_1} \geq T - \frac{\beta_1}{1-\beta_1}. \tag{A.29}$$

Introducing

$$\widetilde{T} = T - \frac{\beta_1}{1-\beta_1}, \tag{A.30}$$

we then have

$$\widetilde{A} \geq \frac{\eta \widetilde{T}}{2(1+q_G)R} \mathbb{E}\left[||\nabla F(\mathbf{w}_\tau)||_2^2\right]. \tag{A.31}$$

Next looking at $\widetilde{B}$, we apply Lemma A.13,

$$\widetilde{B} \leq B'\left(\ln\left(1 + \frac{((1+q_G)R)^2}{\epsilon(1-\beta_2(1-q_V))}\right) - T \ln(\beta_2(1-q_V))\right) \tag{A.32}$$

with

$$B' = \frac{d\eta_T^2 L}{2(1 - \beta_1(1 + q_M))(1 - \frac{\beta_1(1+q_M)}{\beta_2(1-q_V)})}. \tag{A.33}$$

Then looking at $\widetilde{C}$ and introducing the change of index $j = t - l$,

$$\begin{aligned}
\widetilde{C} &= \frac{\eta_T^3 L^2}{4(1 + q_G)R} \sqrt{1 - \beta_1} \sum_{t=1}^{T} \sum_{j=1}^{t} \mathbb{E}\left[\|\mathbf{u}_j\|_2^2\right] \sum_{k=t-j}^{t-1} \beta_1^k \sqrt{k} \\
&= \frac{\eta_T^3 L^2}{4(1 + q_G)R} \sqrt{1 - \beta_1} \sum_{j=1}^{T} \mathbb{E}\left[\|\mathbf{u}_j\|_2^2\right] \sum_{t=j}^{T} \sum_{k=t-j}^{t-1} \beta_1^k \sqrt{k} \\
&= \frac{\eta_T^3 L^2}{4(1 + q_G)R} \sqrt{1 - \beta_1} \sum_{j=1}^{T} \mathbb{E}\left[\|\mathbf{u}_j\|_2^2\right] \sum_{k=0}^{T-1} \beta_1^k \sqrt{k} \sum_{t=j}^{j+k} 1 \\
&= \frac{\eta_T^3 L^2}{4(1 + q_G)R} \sqrt{1 - \beta_1} \sum_{j=1}^{T} \mathbb{E}\left[\|\mathbf{u}_j\|_2^2\right] \sum_{k=0}^{T-1} \beta_1^k \sqrt{k}(k + 1) \\
&\leq \frac{\eta_T^3 L^2}{(1 + q_G)R} \sum_{j=1}^{T} \mathbb{E}\left[\|\mathbf{u}_j\|_2^2\right] \frac{\beta_1}{(1 - \beta_1)^2}. \tag{A.34}
\end{aligned}$$

using Lemma A.12. Finally, using Lemma A.13, we get

$$\widetilde{C} \leq C'\left(\ln\left(1 + \frac{((1 + q_G)R)^2}{\epsilon(1 - \beta_2(1 - q_V))}\right) - T\ln(\beta_2(1 - q_V))\right). \tag{A.35}$$

with

$$C' = \frac{d\eta_T^3 L^2 \beta_1}{(1 + q_G)R(1 - \beta_1)^2(1 - \beta_1(1 + q_M))(1 - \frac{\beta_1(1+q_M)}{\beta_2(1-q_V)})}. \tag{A.36}$$

introducing the same change of index $j = t - k$ for $\widetilde{D}$, we get

$$\begin{aligned}
\widetilde{D} &= \frac{3\eta_T(1 + q_G)R}{\sqrt{1 - \beta_1}} \sum_{t=1}^{T} \sum_{j=1}^{t} \left(\frac{\beta_1}{\beta_2}\right)^{t-j} \sqrt{1 + t - j}\,\mathbb{E}\left[\|\mathbf{U}_j\|_2^2\right] \\
&= \frac{3\eta_T(1 + q_G)R}{\sqrt{1 - \beta_1}} \sum_{j=1}^{T} \mathbb{E}\left[\|\mathbf{U}_j\|_2^2\right] \sum_{t=j}^{T} \left(\frac{\beta_1}{\beta_2}\right)^{t-j} \sqrt{1 + t - j} \\
&\leq \frac{6\eta_T(1 + q_G)R}{\sqrt{1 - \beta_1}} \sum_{j=1}^{T} \mathbb{E}\left[\|\mathbf{U}_j\|_2^2\right] \frac{1}{(1 - \beta_1/\beta_2)^{3/2}}, \tag{A.37}
\end{aligned}$$

using Lemma A.11. Finally, using Lemma A.10, we get

$$\begin{aligned}
\widetilde{D} &\leq \frac{6\eta_T(1 + q_G)R}{\sqrt{1 - \beta_1}(1 - \beta_1/\beta_2)^{3/2}} \sum_{i\in[d]} \left(\ln\left(1 + \frac{v'_{T,i}}{\epsilon}\right) - T\ln(\beta_2)\right) \\
&\leq \frac{6d\eta_T(1 + q_G)R}{\sqrt{1 - \beta_1}(1 - \beta_1/\beta_2)^{3/2}} \left(\ln\left(1 + \frac{((1 + q_G)R)^2}{\epsilon(1 - \beta_2)}\right) - T\ln(\beta_2)\right) \tag{A.38}
\end{aligned}$$

Then we rewrite the quantization error term $E_Q$:

$$E_Q = \eta_T \sum_{t=1}^{T} \left(q_M \cdot dR \cdot C_q + \frac{dR(1 - (1 - q_V)^t)}{\sqrt{1 - \frac{\beta_1^2}{\beta_2(1-q_V)}}}\right)$$

$$= \eta_T \left( \sum_{t=1}^{T} \left( q_M \cdot dR \cdot C_q + \frac{dR}{\sqrt{1 - \frac{\beta_1^2}{\beta_2(1-q_V)}}} \right) - \sum_{t=1}^{T} \frac{dR(1-q_V)^t}{\sqrt{1 - \frac{\beta_1^2}{\beta_2(1-q_V)}}} \right)$$

$$= \eta_T \left( T \left( q_M \cdot dR \cdot C_q + \frac{dR}{\sqrt{1 - \frac{\beta_1^2}{\beta_2(1-q_V)}}} \right) - \frac{dR}{\sqrt{1 - \frac{\beta_1^2}{\beta_2(1-q_V)}}} \sum_{t=1}^{T} (1-q_V)^t \right)$$

$$= \eta_T \left( T \cdot q_M \cdot dR \cdot C_q + \frac{TdR}{\sqrt{1 - \frac{\beta_1^2}{\beta_2(1-q_V)}}} - \frac{dR}{\sqrt{1 - \frac{\beta_1^2}{\beta_2(1-q_V)}}} \left( \frac{1-q_V}{q_V} (1 - (1-q_V)^T) \right) \right)$$

$$\tag{A.39}$$

Finally, we bound Term M using Lemma A.14:

$$M \le \frac{\eta_T dT}{\sqrt{\epsilon}(1-\beta_1)} \left( q_G R^2 + L q_W R ||\mathbf{w}_0||_2 \right) + \frac{\eta_T^2 d^{\frac{3}{2}} L q_W R T^2}{2\sqrt{\epsilon}(1-\beta_1)\sqrt{1 - \frac{\beta_1^2(1+q_M)^2}{\beta_2(1-q_V)}}}, \tag{A.40}$$

Now putting (A.31), (A.32), (A.35), (A.38), (A.39) and (A.40) together into (A.26) and noting that $\eta_T \le \eta \frac{1-\beta_1}{\sqrt{1-\beta_2}}$, we get:

$$\mathbb{E}\left[ ||\nabla F(\mathbf{w}_\tau)||_2^2 \right] \le 2(1+q_G)R \frac{F_0 - F_*}{\eta \widetilde{T}} + \frac{E}{\widetilde{T}} \left( \ln \left( 1 + \frac{((1+q_G)R)^2}{\epsilon(1-\beta_2)} \right) - T \ln(\beta_2) \right)$$

$$+ \frac{H}{\widetilde{T}} \left( \ln \left( 1 + \frac{((1+q_G)R)^2}{\epsilon(1-\beta_2(1-q_V))} \right) - T \ln(\beta_2(1-q_V)) \right) + \frac{Q(T)}{\widetilde{T}}$$

$$+ \frac{2(1+q_G)dT}{\widetilde{T}\sqrt{\epsilon}(1-\beta_2)} \left( q_G R^3 + L q_W R^2 ||\mathbf{w}_0||_2 \right) + \frac{(1-\beta_1)d^{\frac{3}{2}}\eta L q_W (1+q_G)R^2 T^2}{\widetilde{T}\sqrt{\epsilon}(1-\beta_2)\sqrt{1 - \frac{\beta_1^2(1+q_M)^2}{\beta_2(1-q_V)}}}.$$

$$\tag{A.41}$$

with

$$E = \frac{12d((1+q_G)R)^2\sqrt{1-\beta_1}}{(1-\beta_1/\beta_2)^{3/2}\sqrt{1-\beta_2}}$$

$$H = \frac{d\eta L(1+q_G)R(1-\beta_1)^2}{(1-\beta_1(1+q_M))(1 - \frac{\beta_1(1+q_M)}{\beta_2(1-q_V)})(1-\beta_2)} + \frac{2d\eta^2 L^2 \beta_1(1-\beta_1)}{(1-\beta_1(1+q_M))(1 - \frac{\beta_1(1+q_M)}{\beta_2(1-q_V)})(1-\beta_2)^{\frac{3}{2}}}$$

$$Q(T) = \frac{2(1+q_G)q_M dR^2(1-\beta_1)T}{\sqrt{1-\beta_2}} \cdot \frac{\sqrt{r'(1+r')}}{(1+q_M)(1-r')^{3/2}} + \frac{2(1+q_G)dR^2(1-\beta_1)T}{\sqrt{(1 - \frac{\beta_1^2}{\beta_2(1-q_V)})(1-\beta_2)}}$$

$$- \frac{2(1+q_G)dR^2(1-\beta_1)}{\sqrt{(1 - \frac{\beta_1^2}{\beta_2(1-q_V)})(1-\beta_2)}} \left( \frac{1-q_V}{q_V} (1 - (1-q_V)^T) \right)$$

$$\text{where} \quad r' = \frac{\beta_1^2(1+q_M)^2}{\beta_2(1-q_V)}$$

$$\tag{A.42}$$

For clarity in the main theorem statement, we can present a slightly looser but more accessible version of this bound. By noting that for a sufficiently large $T$, we have $\widetilde{T} \ge T/2$, and

$$\left( \ln \left( 1 + \frac{((1+q_G)R)^2}{\epsilon(1-\beta_2)} \right) - T \ln(\beta_2) \right) < \left( \ln \left( 1 + \frac{((1+q_G)R)^2}{\epsilon(1-\beta_2(1-q_V))} \right) - T \ln(\beta_2(1-q_V)) \right),$$

we can state the simplified bound presented in Theorem 4.5:

$$\mathbb{E}\left[ ||\nabla F(\mathbf{w}_\tau)||_2^2 \right] \le 4(1+q_G)R \frac{F_0 - F_*}{\eta T} + \frac{C}{T} \left( \ln \left( 1 + \frac{((1+q_G)R)^2}{\epsilon(1-\beta_2(1-q_V))} \right) - T \ln(\beta_2(1-q_V)) \right)$$

$$+ \frac{\widetilde{Q}(T)}{T} + \frac{4(1+q_G)d}{\sqrt{\epsilon(1-\beta_2)}}\left(q_G R^3 + Lq_W R^2 D\right) + \frac{2(1-\beta_1)d^{\frac{3}{2}}\eta Lq_W(1+q_G)R^2 T}{\sqrt{\epsilon(1-\beta_2)}\sqrt{1-\frac{\beta_1^2(1+q_M)^2}{\beta_2(1-q_V)}}},$$

with

$$C = \frac{24d((1+q_G)R)^2\sqrt{1-\beta_1}}{(1-\beta_1/\beta_2)^{3/2}\sqrt{1-\beta_2}} + \frac{2d\eta L(1+q_G)R(1-\beta_1)^2}{(1-\beta_1(1+q_M))(1-\frac{\beta_1(1+q_M)}{\beta_2(1-q_V)})(1-\beta_2)}$$

$$+ \frac{4d\eta^2 L^2\beta_1(1-\beta_1)}{(1-\beta_1(1+q_M))(1-\frac{\beta_1(1+q_M)}{\beta_2(1-q_V)})(1-\beta_2)^{\frac{3}{2}}},$$

$$\widetilde{Q}(T) = \frac{4(1+q_G)q_M dR^2(1-\beta_1)T}{\sqrt{1-\beta_2}} \cdot \frac{\sqrt{r'(1+r')}}{(1+q_M)(1-r')^{3/2}} + \frac{4(1+q_G)dR^2(1-\beta_1)T}{\sqrt{(1-\frac{\beta_1^2}{\beta_2(1-q_V)})(1-\beta_2)}}$$

$$- \frac{4(1+q_G)dR^2(1-\beta_1)}{\sqrt{(1-\frac{\beta_1^2}{\beta_2(1-q_V)})(1-\beta_2)}}\left(\frac{1-q_V}{q_V}(1-(1-q_V)^T)\right)$$

$$\text{where} \quad r' = \frac{\beta_1^2(1+q_M)^2}{\beta_2(1-q_V)} \tag{A.43}$$

Theory 4.5 states that under a specific schedule for the hyperparameters and a gradual reduction in quantization error, Quantized Adam achieves the same convergence rate as its full-precision counterpart. We prove this by performing a detailed asymptotic analysis of each term in the main bound from Theorem 4.5 as the total number of iterations $T \to \infty$.

However, to perform a precise asymptotic analysis and derive the tightest possible convergence rate from our framework, we will now analyze the order of each component from the more detailed bound in A.41:

$$\mathbb{E}\left[||\nabla F(\text{vec}(\mathbf{W})_\tau)||_2^2\right] \leq \underbrace{2(1+q_G)R\frac{F_0 - F_*}{\eta\widetilde{T}}}_{\text{Term 1}}$$

$$+ \underbrace{\frac{12d((1+q_G)R)^2\sqrt{1-\beta_1}}{\widetilde{T}(1-\beta_1/\beta_2)^{3/2}\sqrt{1-\beta_2}}\left(\ln\left(1+\frac{((1+q_G)R)^2}{\epsilon(1-\beta_2)}\right) - T\ln(\beta_2)\right)}_{\text{Term 2}}$$

$$+ \underbrace{\frac{E}{\widetilde{T}}\left(\ln\left(1+\frac{((1+q_G)R)^2}{\epsilon(1-\beta_2(1-q_V))}\right) - T\ln(\beta_2(1-q_V))\right)}_{\text{Term 3}}$$

$$+ \underbrace{\frac{Q(T)}{\widetilde{T}}}_{\text{Term 4}} + \underbrace{\frac{2(1+q_G)dT}{\widetilde{T}\sqrt{\epsilon(1-\beta_2)}}\left(q_G R^3 + Lq_W R^2||\text{vec}(\mathbf{W}_0)||_2\right)}_{\text{Term 5}}$$

$$+ \underbrace{\frac{(1-\beta_1)d^{\frac{3}{2}}\eta Lq_W(1+q_G)R^2 T^2}{\widetilde{T}\sqrt{\epsilon(1-\beta_2)}\sqrt{1-\frac{\beta_1^2(1+q_M)^2}{\beta_2(1-q_V)}}}}_{\text{Term 6}}.$$

Our proof strategy is to analyze the asymptotic order of each term under the following scaling assumptions.

**Scaling Assumptions.** We adopt the scaling assumptions provided in the Theorem:

- **Quantization Error Schedules:** The quantization errors are annealed over time such that $q_G = \mathcal{O}(T^{-1})$, $q_M = \mathcal{O}(T^{-1})$, $q_W = \mathcal{O}(T^{-2})$, and $q_V = \mathcal{O}(T^{-2})$.

- **Adam Hyperparameters:** The learning rate and second-moment decay are set as $\eta = \Theta(T^{-1/2})$ and $1 - \beta_2 = \Theta(1/T)$, while $\beta_1$ is treated as a constant.

**Asymptotic Analysis of Bound Terms.** We now analyze the order of magnitude for each of the six terms.

**Term 1 (Initial Condition Term):** This term is given by $T_1 = 2(1 + q_G)R\frac{F_0 - F_*}{\eta\widetilde{T}}$. We analyze the components of its denominator. The effective number of iterations is $\widetilde{T} = T - \frac{\beta_1}{1 - \beta_1} = \Theta(T)$. The learning rate scales as $\eta = \Theta(T^{-1/2})$. The denominator thus scales as $\eta\widetilde{T} = \Theta(T^{-1/2})\Theta(T) = \Theta(T^{1/2})$. Since all other quantities are constants and $q_G \to 0$, the entire term scales as:

$$T_1 = \Theta\left(\frac{1}{\eta\widetilde{T}}\right) = \Theta\left(\frac{1}{T^{1/2}}\right) = \Theta(T^{-1/2}).$$

**Term 2 (First Logarithmic Term):** This term is:

$$T_2 = \frac{12d((1 + q_G)R)^2\sqrt{1 - \beta_1}}{\widetilde{T}(1 - \beta_1/\beta_2)^{3/2}\sqrt{1 - \beta_2}}\left(\ln\left(1 + \frac{((1 + q_G)R)^2}{\epsilon(1 - \beta_2)}\right) - T\ln(\beta_2)\right)$$

The leading fraction's order is determined by its denominator, $\widetilde{T}\sqrt{1 - \beta_2}$. With $\widetilde{T} = \Theta(T)$ and $1 - \beta_2 = \Theta(1/T)$, we have $\sqrt{1 - \beta_2} = \Theta(T^{-1/2})$. Thus, the fraction scales as $\Theta(\frac{1}{T \cdot T^{-1/2}}) = \Theta(T^{-1/2})$. The term in the parenthesis scales as $\ln(1 + \Theta(T)) - \Theta(1) = \Theta(\ln T)$. The overall order is:

$$T_2 = \Theta\left(\frac{1}{\widetilde{T}\sqrt{1 - \beta_2}}\right) \cdot \Theta(\ln T) = \Theta\left(T^{-1/2}\right) \cdot \Theta(\ln T) = \Theta\left(\frac{\ln T}{\sqrt{T}}\right).$$

**Term 3 (Second Logarithmic Term):** This term is $T_3 = \frac{E}{\widetilde{T}}\left(\ln(...) - T\ln(...)\right)$. First, we determine the asymptotic order of $E$, which is defined as:

$$E = \frac{d\eta L(1 + q_G)R(1 - \beta_1)^2}{(1 - \beta_1(1 + q_M))(1 - \frac{\beta_1(1 + q_M)}{\beta_2(1 - q_V)})(1 - \beta_2)} + \frac{2d\eta^2 L^2\beta_1(1 - \beta_1)}{(1 - \beta_1(1 + q_M))(1 - \frac{\beta_1(1 + q_M)}{\beta_2(1 - q_V)})(1 - \beta_2)^{\frac{3}{2}}}.$$

For the first part of $E$, the numerator scales as $\eta = \Theta(T^{-1/2})$ and the denominator is dominated by $(1 - \beta_2) = \Theta(T^{-1})$. This part is $\Theta(T^{-1/2})/\Theta(T^{-1}) = \Theta(T^{1/2})$. For the second part, the numerator scales as $\eta^2 = \Theta(T^{-1})$ and the denominator is dominated by $(1 - \beta_2)^{3/2} = \Theta(T^{-3/2})$. This part is $\Theta(T^{-1})/\Theta(T^{-3/2}) = \Theta(T^{1/2})$. Thus, $E = \Theta(T^{1/2})$. The logarithmic part scales as $\Theta(\ln T)$, so the entire term scales as:

$$T_3 = \Theta\left(\frac{E}{\widetilde{T}}\right) \cdot \Theta(\ln T) = \Theta\left(\frac{T^{1/2}}{T}\right) \cdot \Theta(\ln T) = \Theta\left(\frac{\ln T}{\sqrt{T}}\right).$$

**Term 4 (Moment Quantization Error):** We rewrite this term as:

$$T_4 = \frac{2(1 + q_G)dR^2 T(1 - \beta_1)}{\widetilde{T}\sqrt{1 - \beta_2}}Q, \tag{A.44}$$

where $Q = q_M \cdot \frac{\sqrt{r'(1 + r')}}{(1 + q_M)(1 - r')^{3/2}} + \frac{1}{\sqrt{1 - \frac{\beta_1^2}{\beta_2(1 - q_V)}}} - \frac{1}{T}\frac{1}{\sqrt{1 - \frac{\beta_1^2}{\beta_2(1 - q_V)}}}\left(\frac{1 - q_V}{q_V}(1 - (1 - q_V)^T)\right)$.

Our goal is to show that $T_4 = \mathcal{O}(T^{-1/2})$.

First, the pre-factor has an asymptotic order of:

$$\frac{2(1 + q_G)dR^2 T(1 - \beta_1)}{\widetilde{T}\sqrt{1 - \beta_2}} = \Theta\left(\frac{T}{T \cdot T^{-1/2}}\right) = \Theta(T^{1/2}).$$

The core of the analysis thus lies in determining the order of $Q$. We can rewrite $Q$ by combining its second and third components:

$$Q = q_M \cdot \frac{\sqrt{r'(1+r')}}{(1+q_M)(1-r')^{3/2}} + \frac{1}{\sqrt{1 - \frac{\beta_1^2}{\beta_2(1-q_V)}}} \left[ 1 - \frac{1}{T} \left( \frac{1-q_V}{q_V} (1 - (1-q_V)^T) \right) \right].$$

The first part of $Q$ is clearly $\mathcal{O}(q_M) = \mathcal{O}(T^{-1})$. The common factor in the second part, $\frac{1}{\sqrt{1-\ldots}}$, converges to a constant as $T \to \infty$, so it is $\mathcal{O}(1)$. The analysis therefore simplifies to finding the order of the bracketed term.

Let $x = q_V = \mathcal{O}(T^{-2})$. We perform a Taylor expansion on $(1-x)^T$:

$$(1-x)^T = 1 - Tx + \frac{T(T-1)}{2}x^2 + \mathcal{O}(T^3 x^3).$$

This allows us to analyze the term inside the bracket:

$$
\begin{aligned}
1 - \frac{1}{T} \left( \frac{1-x}{x} (1 - (1-x)^T) \right) &= 1 - \frac{1}{T} \frac{1-x}{x} \left( Tx - \frac{T(T-1)}{2}x^2 + \mathcal{O}(T^3 x^3) \right) \\
&= 1 - \frac{1}{T}(1-x) \left( T - \frac{T(T-1)}{2}x + \mathcal{O}(T^3 x^2) \right) \\
&= 1 - \frac{1}{T} \left( T - \frac{T(T-1)}{2}x - Tx + \mathcal{O}(T^3 x^2) \right) \\
&= 1 - \left( 1 - \frac{T(T+1)}{2T}x + \mathcal{O}(T^2 x^2) \right) \\
&= \frac{T+1}{2}x - \mathcal{O}(T^2 x^2).
\end{aligned}
$$

Substituting back $x = q_V = \mathcal{O}(T^{-2})$, the bracketed term has an order of:

$$\mathcal{O}(T \cdot q_V) = \mathcal{O}(T \cdot T^{-2}) = \mathcal{O}(T^{-1}).$$

Therefore, the entire second component of $Q$ is $\mathcal{O}(1) \cdot \mathcal{O}(T^{-1}) = \mathcal{O}(T^{-1})$. Combining both components of $Q$, we find its overall order:

$$Q = \mathcal{O}(T^{-1}) + \mathcal{O}(T^{-1}) = \mathcal{O}(T^{-1}).$$

Finally, we compute the order of Term 4 by combining the pre-factor and $Q$:

$$T_4 = \Theta(T^{1/2}) \cdot \mathcal{O}(T^{-1}) = \mathcal{O}(T^{-1/2}).$$

**Term 5 (Initial W/G Quantization Error):** This term is:

$$T_5 = \frac{2(1+q_G)dT}{\widetilde{T}\sqrt{\epsilon(1-\beta_2)}} \left( q_G R^3 + Lq_W R^2 ||\text{vec}(\mathbf{W}_0)||_2 \right)$$

The leading fraction scales as $\frac{T}{\widetilde{T}\sqrt{1-\beta_2}} = \frac{\Theta(T)}{\Theta(T)\Theta(T^{-1/2})} = \Theta(T^{1/2})$. The parenthesis scales with its dominant term $q_G = \mathcal{O}(T^{-1})$. The total order is:

$$T_5 = \Theta(T^{1/2}) \cdot \mathcal{O}(q_G + q_W) = \Theta(T^{1/2}) \cdot (\mathcal{O}(T^{-1}) + \mathcal{O}(T^{-2})) = \mathcal{O}(T^{-1/2}).$$

**Term 6 (Weight Growth Quantization Error):** This term is $T_6 = \frac{(1-\beta_1)d^{\frac{3}{2}}\eta Lq_W(1+q_G)R^2 T^2}{\widetilde{T}\sqrt{\epsilon}(1-\beta_2)\sqrt{1 - \frac{\beta_1^2(1+q_M)^2}{\beta_2(1-q_V)}}}$.

First, we analyze $\sqrt{\frac{1}{1 - \frac{\beta_1^2(1+q_M)^2}{\beta_2(1-q_V)}}}$. As $T \to \infty$, the denominator converges to the constant $1 - \beta_1^2$, so its contribution is $\mathcal{O}(1)$. The term's order is determined by the scaling of its other components: $\eta = \Theta(T^{-1/2})$, $q_W = \mathcal{O}(T^{-2})$, $\widetilde{T} = \Theta(T)$, and $(1-\beta_2) = \Theta(T^{-1})$. The total order is:

$$T_6 = \Theta \left( \frac{\eta \cdot q_W \cdot T^2}{\widetilde{T} \cdot (1-\beta_2)} \right) = \Theta \left( \frac{T^{-1/2} \cdot T^{-2} \cdot T^2}{T \cdot T^{-1}} \right) = \Theta(T^{-1/2}).$$

**Conclusion.** By comparing the asymptotic orders of all terms, we identify those that converge to zero at the slowest rate, as they will dominate the overall convergence bound. The orders are:

- Term 1, 4, 5, 6: $\mathcal{O}(T^{-1/2})$ or $\Theta(T^{-1/2})$.
- Term 2, 3: $\Theta(T^{-1/2} \ln T)$.

The dominant terms are the second and third, which are of order $\Theta(T^{-1/2} \ln T)$. These terms form the bottleneck that determines the overall convergence rate. Thus, under the specified parameter schedule, the expected squared gradient norm converges to zero at the following rate:

$$\mathbb{E}\left[||\nabla F(\mathbf{w}_\tau)||_2^2\right] = \Theta\left(\frac{\ln T}{\sqrt{T}}\right) = \widetilde{\mathcal{O}}\left(\frac{1}{\sqrt{T}}\right).$$

This matches the known convergence rate for full-precision Adam.

Furthermore, we derive the convergence rate for the expected gradient norm, $\mathbb{E}\left[||\nabla F(\mathbf{w}_\tau)||_2\right]$, from the rate of its squared value. We use Jensen's inequality, which states that for a convex function $\phi$ and a random variable $X$, $\phi(\mathbb{E}[X]) \leq \mathbb{E}[\phi(X)]$.

Let the random variable be $X = ||\nabla F(\mathbf{w}_\tau)||_2$ and the convex function be $\phi(x) = x^2$. Applying Jensen's inequality yields:

$$\left(\mathbb{E}\left[||\nabla F(\mathbf{w}_\tau)||_2\right]\right)^2 \leq \mathbb{E}\left[||\nabla F(\mathbf{w}_\tau)||_2^2\right].$$

By taking the square root of both sides, we obtain a bound on the expected norm:

$$\mathbb{E}\left[||\nabla F(\mathbf{w}_\tau)||_2\right] \leq \sqrt{\mathbb{E}\left[||\nabla F(\mathbf{w}_\tau)||_2^2\right]}.$$

Substituting our previously derived convergence rate:

$$\begin{aligned}
\mathbb{E}\left[||\nabla F(\mathbf{w}_\tau)||_2\right] &\leq \sqrt{\widetilde{\mathcal{O}}\left(\frac{1}{\sqrt{T}}\right)} \\
&= \widetilde{\mathcal{O}}\left(\sqrt{T^{-1/2}}\right) \\
&= \widetilde{\mathcal{O}}\left(T^{-1/4}\right).
\end{aligned}$$

Thus, the expected gradient norm converges to zero at a rate of $\widetilde{\mathcal{O}}(T^{-1/4})$. This finalizes the proof of the theorem.

### A.3 PROOF OF LEMMA A.2

**Lemma A.2** (The value range of $v_{t,i}$ and the upper bound of $|\frac{1}{\sqrt{\epsilon+v_{t,i}}} - \frac{1}{\sqrt{\epsilon+v'_{t,i}}}|$). Let $LB_{t,i} = \sum_{k=0}^{t} \beta_2^{t-k}(1-q_V)^{t-k}(\nabla_i f_k(\mathbf{w}_{k-1})+\delta_{k,i})^2$ and $UB_{t,i} = \sum_{k=0}^{t} \beta_2^{t-k}(1+q_V)^{t-k}(\nabla_i f_k(\mathbf{w}_{k-1})+\delta_{k,i})^2$. We have:

$$\sum_{k=0}^{t} \beta_2^{t-k}(1-q_V)^{t-k}(\nabla_i f_k(\mathbf{w}_{k-1})+\delta_{k,i})^2 \leq v_{t,i} \leq \sum_{k=0}^{t} \beta_2^{t-k}(1+q_V)^{t-k}(\nabla_i f_k(\mathbf{w}_{k-1})+\delta_{k,i})^2$$
$$(A.45)$$

$$\left|\frac{1}{\sqrt{\epsilon+v_{t,i}}} - \frac{1}{\sqrt{\epsilon+v'_{t,i}}}\right| \leq \max\left\{\frac{1}{\sqrt{\epsilon+LB_{t,i}}} - \frac{1}{\sqrt{\epsilon+v'_{t,i}}}, \frac{1}{\sqrt{\epsilon+v'_{t,i}}} - \frac{1}{\sqrt{\epsilon+UB_{t,i}}}\right\}$$
$$(A.46)$$

*Proof.* The proof consists of two parts.

**Part 1: Bounding $v_{t,i}$**

The update rule for the second moment estimate is $v_{t,i} = \beta_2(v_{t-1,i}+\theta_{t-1,i})+(\nabla_i f_t(\mathbf{w}_{t-1})+\delta_{t,i})^2$. The quantization noise is assumed to be a relative error, bounded by $|\theta_{t-1,i}| \le q_V|v_{t-1,i}|$. This implies that $(1-q_V)v_{t-1,i} \le v_{t-1,i}+\theta_{t-1,i} \le (1+q_V)v_{t-1,i}$.

Applying this to the update rule, we can establish the lower bound by recursively unrolling the inequality:

$$
\begin{aligned}
v_{t,i} &\ge \beta_2(1-q_V)v_{t-1,i} + (\nabla_i f_t(\mathbf{w}_{t-1}) + \delta_{t,i})^2 \\
&\ge \beta_2(1-q_V)\left[\beta_2(1-q_V)v_{t-2,i} + (\nabla_i f_{t-1}(\mathbf{w}_{t-2}) + \delta_{t-1,i})^2\right] + (\nabla_i f_t(\mathbf{w}_{t-1}) + \delta_{t,i})^2 \\
&= \ldots \\
&= \sum_{k=0}^{t} \beta_2^{t-k}(1-q_V)^{t-k}(\nabla_i f_k(\mathbf{w}_{k-1}) + \delta_{k,i})^2
\end{aligned}
\tag{A.47}
$$

Similarly, we can establish the upper bound:

$$
\begin{aligned}
v_{t,i} &\le \beta_2(1+q_V)v_{t-1,i} + (\nabla_i f_t(\mathbf{w}_{t-1}) + \delta_{t,i})^2 \\
&= \sum_{k=0}^{t} \beta_2^{t-k}(1+q_V)^{t-k}(\nabla_i f_k(\mathbf{w}_{k-1}) + \delta_{k,i})^2
\end{aligned}
\tag{A.48}
$$

This completes the proof of the first statement in the lemma.

**Part 2: Bounding the difference of the inverse square roots**

Let $v'_{t,i}$ be the idealized second moment estimate, updated without the quantization noise $\theta$. Its explicit form is:

$$
v'_{t,i} = \sum_{k=0}^{t} \beta_2^{t-k}(\nabla_i f_k(\mathbf{w}_{k-1}) + \delta_{k,i})^2
\tag{A.49}
$$

From Part 1, we know that $v_{t,i}$ is in the interval $[LB_{t,i}, UB_{t,i}]$, where $LB_{t,i}$ and $UB_{t,i}$ are the bounds established.

Now, we compare $v'_{t,i}$ with these bounds. Since $0 < \beta_2 < 1$ and we assume $0 < q_V < 1$, we have $\beta_2(1-q_V) < \beta_2 < \beta_2(1+q_V)$. This implies a term-by-term inequality, leading to:

$$
LB_{t,i} < v'_{t,i} < UB_{t,i}
\tag{A.50}
$$

Consider the function $f(y) = 1/\sqrt{\epsilon + y}$ for $y \ge 0$. This function is monotonically decreasing and convex. The value $v_{t,i}$ lies in the interval $[LB_{t,i}, UB_{t,i}]$, and $v'_{t,i}$ is a point within this interval. The maximum absolute difference $|f(v_{t,i}) - f(v'_{t,i})|$ must occur when $v_{t,i}$ is at one of the endpoints of the interval. Therefore, we can bound the difference as:

$$
\left|\frac{1}{\sqrt{\epsilon + v_{t,i}}} - \frac{1}{\sqrt{\epsilon + v'_{t,i}}}\right| \le \max\left\{\left|\frac{1}{\sqrt{\epsilon + LB_{t,i}}} - \frac{1}{\sqrt{\epsilon + v'_{t,i}}}\right|, \left|\frac{1}{\sqrt{\epsilon + UB_{t,i}}} - \frac{1}{\sqrt{\epsilon + v'_{t,i}}}\right|\right\}
\tag{A.51}
$$

Since $LB_{t,i} < v'_{t,i} < UB_{t,i}$ and the function is decreasing, we have $1/\sqrt{\epsilon + UB_{t,i}} < 1/\sqrt{\epsilon + v'_{t,i}} < 1/\sqrt{\epsilon + LB_{t,i}}$. We can therefore remove the absolute value signs:

$$
\left|\frac{1}{\sqrt{\epsilon + v_{t,i}}} - \frac{1}{\sqrt{\epsilon + v'_{t,i}}}\right| \le \max\left\{\frac{1}{\sqrt{\epsilon + LB_{t,i}}} - \frac{1}{\sqrt{\epsilon + v'_{t,i}}}, \frac{1}{\sqrt{\epsilon + v'_{t,i}}} - \frac{1}{\sqrt{\epsilon + UB_{t,i}}}\right\}
\tag{A.52}
$$

This completes the proof of the second statement. □

## A.4 PROOF OF LEMMA A.3

**Lemma A.3** (Bound on Discrete Error). Given two discrete-time systems defined for $t \geq 1$:

- System A: $a_t = k(a_{t-1} + c_{t-1}) + d_t$
- System B: $b_t = kb_{t-1} + d_t$

where the perturbation term $c_t$ is bounded by $|c_t| \leq q|a_t|$ for all $t$, and the constants $k, q$ satisfy $0 < k < 1$ and $q < k$.

Under zero initial conditions, where $a_0 = b_0 = 0$, the absolute error between the states of the two systems is bounded by:

$$|a_t - b_t| \leq \sum_{j=1}^{t-1} \left[ (k(1+q))^{t-j} - k^{t-j} \right] |d_j| \tag{A.53}$$

*Proof.* First, define the error as $e_t = a_t - b_t$. Subtracting the two system equations yields the error recurrence relation:

$$e_t = ke_{t-1} + kc_{t-1} \tag{A.54}$$

The explicit solution to this recurrence is $e_t = k^t e_0 + \sum_{j=0}^{t-1} k^{t-j} c_j$. Under the zero initial condition $a_0 = b_0 = 0$, this simplifies to:

$$e_t = \sum_{j=0}^{t-1} k^{t-j} c_j \tag{A.55}$$

Taking the absolute value and applying the given condition $|c_j| \leq q|a_j|$, we have:

$$|e_t| \leq \sum_{j=0}^{t-1} k^{t-j} |c_j| \leq q \sum_{j=0}^{t-1} k^{t-j} |a_j| \tag{A.56}$$

Since $a_0 = 0$, the sum starts from $j = 1$. The state $|a_j|$ can be bounded from its own recurrence $|a_t| \leq k(1+q)|a_{t-1}| + |d_t|$, which for $a_0 = 0$ unrolls to:

$$|a_j| \leq \sum_{i=1}^{j} (k(1+q))^{j-i} |d_i| \tag{A.57}$$

Substituting the bound for $|a_j|$ into the inequality for $|e_t|$ gives a double summation:

$$|e_t| \leq q \sum_{j=1}^{t-1} k^{t-j} \left( \sum_{i=1}^{j} (k(1+q))^{j-i} |d_i| \right) \tag{A.58}$$

By swapping the order of summation and evaluating the inner geometric series, we obtain the final result:

$$|a_t - b_t| \leq \sum_{j=1}^{t-1} \left[ (k(1+q))^{t-j} - k^{t-j} \right] |d_j| \tag{A.59}$$

$\square$

## A.5 PROOF OF LEMMA A.4

**Lemma A.4** (Finite Geometric Series Ratio Bounded by Infinite Sum). Let $(g_k)_{k=0}^{t}$ be a sequence of scalars for any finite $t \in \mathbb{N}$. Let the weights be terms of two geometric series, $A_k = a^k$ and $B_k = b^k$, where $a, b \in (0, 1)$ are the base ratios.

If the condition $a^2 < b$ holds, then the ratio of the weighted sum is bounded by a constant derived from the corresponding infinite series:

$$\frac{\sum_{k=0}^{t} a^k |g_k|}{\sqrt{\sum_{k=0}^{t} b^k g_k^2}} \leq \sqrt{\frac{1}{1 - a^2/b}} \tag{A.60}$$

*Proof.* Let the numerator be $N_t = \sum_{k=0}^{t} a^k |g_k|$ and the denominator be $D_t = \sqrt{\sum_{k=0}^{t} b^k g_k^2}$.

We rewrite the numerator as:

$$N_t = \sum_{k=0}^{t} \left( \frac{a^k}{\sqrt{b^k}} \right) \cdot \left( \sqrt{b^k} |g_k| \right) \tag{A.61}$$

Applying the Cauchy-Schwarz inequality to these finite sums, we get:

$$N_t^2 \leq \left( \sum_{k=0}^{t} \left( \frac{a^k}{\sqrt{b^k}} \right)^2 \right) \cdot \left( \sum_{k=0}^{t} \left( \sqrt{b^k} |g_k| \right)^2 \right)$$

$$= \left( \sum_{k=0}^{t} \frac{a^{2k}}{b^k} \right) \cdot \left( \sum_{k=0}^{t} b^k g_k^2 \right)$$

$$= \left( \sum_{k=0}^{t} \left( \frac{a^2}{b} \right)^k \right) \cdot D_t^2 \tag{A.62}$$

The first term is a finite geometric series. Since the condition $a^2 < b$ implies that the ratio $r = a^2/b$ is positive and less than 1, all terms in the series are positive. Therefore, the finite sum is always less than or equal to the sum of the infinite series:

$$\sum_{k=0}^{t} \left( \frac{a^2}{b} \right)^k \leq \sum_{k=0}^{\infty} \left( \frac{a^2}{b} \right)^k = \frac{1}{1 - a^2/b} \tag{A.63}$$

Substituting this upper bound back into the inequality for $N_t^2$, we have:

$$N_t^2 \leq \left( \frac{1}{1 - a^2/b} \right) \cdot D_t^2 \tag{A.64}$$

Taking the square root of both sides gives:

$$N_t \leq \sqrt{\frac{1}{1 - a^2/b}} \cdot D_t \tag{A.65}$$

Finally, dividing by $D_t$ yields the desired result for any finite $t$:

$$\frac{N_t}{D_t} = \frac{\sum_{k=0}^{t} a^k |g_k|}{\sqrt{\sum_{k=0}^{t} b^k g_k^2}} \leq \sqrt{\frac{1}{1 - a^2/b}} \tag{A.66}$$

$\square$

## A.6 PROOF OF LEMMA A.5

**Lemma A.5** (Bound on the Quantized Momentum Error Ratio)**.** Let $(g_k)_{k=0}^{t}$ be a sequence of scalars. Let the weights be $A_k = \beta_1^k((1 + q_M)^k - 1)$ and $B_k = (\beta_2(1 - q_V))^k$. If the condition $\beta_1^2(1 + q_M)^2 < \beta_2(1 - q_V)$ holds, then the ratio of the weighted sum is bounded by:

$$\frac{\sum_{k=0}^{t} A_k |g_k|}{\sqrt{\sum_{k=0}^{t} B_k g_k^2}} \leq q_M \cdot \frac{\sqrt{r'(1 + r')}}{(1 + q_M)(1 - r')^{3/2}}$$

where $r' = \frac{\beta_1^2(1+q_M)^2}{\beta_2(1-q_V)}$.

*Proof.* Following the proof of Lemma A.4, we apply the Cauchy-Schwarz inequality to get:

$$\left( \sum_{k=0}^{t} A_k |g_k| \right)^2 \leq \left( \sum_{k=0}^{t} \frac{A_k^2}{B_k} \right) \cdot \left( \sum_{k=0}^{t} B_k g_k^2 \right). \tag{A.67}$$

This implies that the ratio is bounded by the square root of the first term on the right-hand side. We now focus on bounding the term $\sum_{k=0}^{t} \frac{A_k^2}{B_k}$. First, we express the ratio $\frac{A_k^2}{B_k}$ as:

$$
\begin{aligned}
\frac{A_k^2}{B_k} &= \frac{(\beta_1^k((1+q_M)^k - 1))^2}{(\beta_2(1-q_V))^k} \\
&= \left(\frac{\beta_1^2}{\beta_2(1-q_V)}\right)^k ((1+q_M)^k - 1)^2.
\end{aligned}
\tag{A.68}
$$

To bound the term $((1+q_M)^k - 1)^2$, we first establish an inequality for $(1+q_M)^k - 1$ using the Mean Value Theorem. Let $f(x) = x^k$. For $q_M > 0$, by the Mean Value Theorem, there exists a $c \in (1, 1+q_M)$ such that:

$$
\frac{f(1+q_M) - f(1)}{(1+q_M) - 1} = f'(c) \implies (1+q_M)^k - 1 = q_M \cdot (kc^{k-1}).
\tag{A.69}
$$

Since $c < 1 + q_M$, and for $k \geq 1$, we have $c^{k-1} \leq (1+q_M)^{k-1}$. This leads to the inequality:

$$
(1+q_M)^k - 1 \leq k \cdot q_M \cdot (1+q_M)^{k-1}.
\tag{A.70}
$$

Squaring both sides of (A.70) gives:

$$
\begin{aligned}
((1+q_M)^k - 1)^2 &\leq k^2 q_M^2 (1+q_M)^{2(k-1)} \\
&= \frac{k^2 q_M^2}{(1+q_M)^2}(1+q_M)^{2k}.
\end{aligned}
\tag{A.71}
$$

Substituting this back into (A.68), and using the definition $r' = \frac{\beta_1^2(1+q_M)^2}{\beta_2(1-q_V)}$, we get:

$$
\begin{aligned}
\frac{A_k^2}{B_k} &\leq \left(\frac{\beta_1^2}{\beta_2(1-q_V)}\right)^k \frac{k^2 q_M^2}{(1+q_M)^2}(1+q_M)^{2k} \\
&= \frac{q_M^2}{(1+q_M)^2} k^2 \left(\frac{\beta_1^2(1+q_M)^2}{\beta_2(1-q_V)}\right)^k \\
&= \frac{q_M^2}{(1+q_M)^2} k^2 (r')^k.
\end{aligned}
\tag{A.72}
$$

Now we sum this term. The condition $r' < 1$ ensures the convergence of the infinite series. We first derive the closed-form expression for $\sum_{k=0}^{\infty} k^2 x^k$ for $|x| < 1$. We start with the geometric series:

$$
\sum_{k=0}^{\infty} x^k = \frac{1}{1-x}.
\tag{A.73}
$$

Differentiating with respect to $x$ and multiplying by $x$ gives:

$$
\sum_{k=0}^{\infty} k x^k = x \frac{d}{dx}\left(\frac{1}{1-x}\right) = \frac{x}{(1-x)^2}.
\tag{A.74}
$$

Differentiating one more time and multiplying by $x$ yields:

$$
\sum_{k=0}^{\infty} k^2 x^k = x \frac{d}{dx}\left(\frac{x}{(1-x)^2}\right) = x \frac{1(1-x)^2 - x(2(1-x)(-1))}{(1-x)^4} = \frac{x(1+x)}{(1-x)^3}.
\tag{A.75}
$$

Using this result with $x = r'$, we can bound the sum $\sum_{k=0}^{t} \frac{A_k^2}{B_k}$ by extending it to an infinite series:

$$
\begin{aligned}
\sum_{k=0}^{t} \frac{A_k^2}{B_k} &\leq \sum_{k=0}^{\infty} \frac{A_k^2}{B_k} \\
&\leq \sum_{k=0}^{\infty} \frac{q_M^2}{(1+q_M)^2} k^2 (r')^k
\end{aligned}
$$

$$= \frac{q_M^2}{(1+q_M)^2} \sum_{k=0}^{\infty} k^2 (r')^k$$

$$= \frac{q_M^2}{(1+q_M)^2} \frac{r'(1+r')}{(1-r')^3}. \tag{A.76}$$

Finally, taking the square root of (A.76) and substituting it back into the result from the Cauchy-Schwarz inequality (A.67) gives the desired bound:

$$\frac{\sum_{k=0}^{t} A_k |g_k|}{\sqrt{\sum_{k=0}^{t} B_k g_k^2}} \leq \sqrt{\sum_{k=0}^{t} \frac{A_k^2}{B_k}} \leq \sqrt{\frac{q_M^2}{(1+q_M)^2} \frac{r'(1+r')}{(1-r')^3}} = q_M \cdot \frac{\sqrt{r'(1+r')}}{(1+q_M)(1-r')^{3/2}}. \tag{A.77}$$

$\square$

## A.7 PROOF OF LEMMA A.6

**Lemma A.6** (Bound on the Quantized Gradient Estimator). Let the stochastic gradient be bounded in infinity norm almost surely by $\|\nabla f_t(\mathbf{w}; \gamma)\|_\infty \leq R - \sqrt{\epsilon}$ for any parameters $\mathbf{w}$. Let the gradient quantization operator satisfy the relative error model $|Q(z) - z| \leq q_G |z|$ for any scalar $z$. The quantized gradient estimator $\widehat{\mathbf{g}}_t$ is defined component-wise for $i \in [d]$ as:

$$\widehat{g}_{t,i} = \frac{1}{B} \sum_{j=1}^{B} [\nabla^Q f(\mathbf{w}_{t-1}^Q; \gamma_{t,j})]_i, \tag{A.78}$$

where we use $\nabla^Q f(\cdot)$ as shorthand for $Q(\nabla f(\cdot))$ and $[\cdot]_i$ to denote the i-th component. Then, the infinity norm of the estimator is bounded almost surely:

$$\|\widehat{\mathbf{g}}_t\|_\infty \leq (1 + q_G)(R - \sqrt{\epsilon}). \tag{A.79}$$

For notational simplicity in subsequent proofs, we will use the slightly looser bound $\|\widehat{\mathbf{g}}_t\|_\infty \leq (1 + q_G)R$.

*Proof.* We first bound the infinity norm of a single quantized gradient vector $\nabla^Q f(\cdot)$. For any component $i \in [d]$, we have:

$$\left| \nabla_i^Q f(\mathbf{w}_{t-1}^Q; \gamma_{t,j}) \right| = \left| \nabla_i f(\mathbf{w}_{t-1}^Q; \gamma_{t,j}) + \left( \nabla_i^Q f(\mathbf{w}_{t-1}^Q; \gamma_{t,j}) - \nabla_i f(\mathbf{w}_{t-1}^Q; \gamma_{t,j}) \right) \right|$$

$$\leq \left| \nabla_i f(\mathbf{w}_{t-1}^Q; \gamma_{t,j}) \right| + \left| \nabla_i^Q f(\mathbf{w}_{t-1}^Q; \gamma_{t,j}) - \nabla_i f(\mathbf{w}_{t-1}^Q; \gamma_{t,j}) \right|$$

$$\leq \left| \nabla_i f(\mathbf{w}_{t-1}^Q; \gamma_{t,j}) \right| + q_G \left| \nabla_i f(\mathbf{w}_{t-1}^Q; \gamma_{t,j}) \right|$$

$$= (1 + q_G) \left| \nabla_i f(\mathbf{w}_{t-1}^Q; \gamma_{t,j}) \right|. \tag{A.80}$$

Since this holds for any component, it also holds for the component with the maximum absolute value. Therefore, by taking the maximum over $i \in [d]$, we can bound the infinity norm:

$$\|\nabla^Q f(\mathbf{w}_{t-1}^Q; \gamma_{t,j})\|_\infty \leq (1 + q_G)\|\nabla f(\mathbf{w}_{t-1}^Q; \gamma_{t,j})\|_\infty$$

$$\leq (1 + q_G)(R - \sqrt{\epsilon}). \tag{A.81}$$

Finally, we apply the triangle inequality to the full estimator $\widehat{\mathbf{g}}_t$, which is the average over $B$ such vectors:

$$\|\widehat{\mathbf{g}}_t\|_\infty = \|\frac{1}{B} \sum_{j=1}^{B} \nabla^Q f(\mathbf{w}_{t-1}^Q; \gamma_{t,j})\|_\infty$$

$$\leq \frac{1}{B} \sum_{j=1}^{B} \|\nabla^Q f(\mathbf{w}_{t-1}^Q; \gamma_{t,j})\|_\infty$$

$$\leq \frac{1}{B} \sum_{j=1}^{B} (1 + q_G)(R - \sqrt{\epsilon})$$

$$= (1 + q_G)(R - \sqrt{\epsilon}). \tag{A.82}$$

This concludes the proof. $\square$

## A.8 PROOF OF LEMMA A.7

**Lemma A.7** (Bound on the Expected Gradient Error with Biased Quantization). Under the assumptions that the infinity norm of stochastic gradient is up bounded (Assumption 4.2), the objective $F$ is L-smooth (Assumption 4.3), and the quantization relative error model holds (Assumption 3.1), the magnitude of the conditional expectation of the total error term $\delta_{t,i}$ is bounded by:

$$|\mathbb{E}_{t-1}[\delta_{t,i}]| \leq q_G R + L q_W ||\mathbf{w}_{t-1}||_2.$$

*Proof.* We start from the decomposition of the conditional expectation of $\delta_{t,i}$, which we derived previously:

$$\mathbb{E}_{t-1}[\delta_{t,i}] = \mathbb{E}_\gamma \left[ \nabla_i^Q f(\mathbf{w}_{t-1}^Q; \gamma) - \nabla_i f(\mathbf{w}_{t-1}^Q; \gamma) \right] + \mathbb{E}_\gamma \left[ \nabla_i f(\mathbf{w}_{t-1}^Q; \gamma) - \nabla_i f(\mathbf{w}_{t-1}; \gamma) \right]$$

$$= \underbrace{\mathbb{E}_\gamma \left[ \nabla_i^Q f(\mathbf{w}_{t-1}^Q; \gamma) - \nabla_i f(\mathbf{w}_{t-1}^Q; \gamma) \right]}_{\text{Term I: Gradient Quantization Bias}} + \underbrace{\left( \nabla_i F(\mathbf{w}_{t-1}^Q) - \nabla_i F(\mathbf{w}_{t-1}) \right)}_{\text{Term II: Weight Quantization Bias}} \quad \text{(A.83)}$$

Using the triangle inequality, we can bound the magnitude as:

$$|\mathbb{E}_{t-1}[\delta_{t,i}]| \leq |\text{Term I}| + |\text{Term II}|. \quad \text{(A.84)}$$

We bound each term separately.

**Bounding Term I:** This term is the expected bias from the (potentially biased) gradient quantization. We first apply Jensen's inequality for absolute values, i.e., $|\mathbb{E}[X]| \leq \mathbb{E}[|X|]$:

$$|\text{Term I}| = \left| \mathbb{E}_\gamma \left[ \nabla_i^Q f(\mathbf{w}_{t-1}^Q; \gamma) - \nabla_i f(\mathbf{w}_{t-1}^Q; \gamma) \right] \right|$$

$$\leq \mathbb{E}_\gamma \left[ \left| \nabla_i^Q f(\mathbf{w}_{t-1}^Q; \gamma) - \nabla_i f(\mathbf{w}_{t-1}^Q; \gamma) \right| \right]. \quad \text{(A.85)}$$

By the relative error model for gradient quantization (Assumption 3.1 with factor $q_G$):

$$|\text{Term I}| \leq \mathbb{E}_\gamma \left[ q_G \left| \nabla_i f(\mathbf{w}_{t-1}^Q; \gamma) \right| \right] \leq q_G R. \quad \text{(A.86)}$$

**Bounding Term II:** This term represents the bias from weight quantization. Using the L-smoothness of $F$ (Assumption 4.3) and the relative error for weights (Assumption 3.1):

$$|\text{Term II}| = |\nabla_i F(\mathbf{w}_{t-1}^Q) - \nabla_i F(\mathbf{w}_{t-1})| \leq ||\nabla F(\mathbf{w}_{t-1}^Q) - \nabla F(\mathbf{w}_{t-1})||_2 \leq L q_W ||\mathbf{w}_{t-1}||_2. \quad \text{(A.87)}$$

**Combining the Bounds:** Summing the bounds for Term I and Term II, we arrive at the final result:

$$|\mathbb{E}_{t-1}[\delta_{t,i}]| \leq |\text{Term I}| + |\text{Term II}| \leq q_G R + L q_W ||\mathbf{w}_{t-1}||_2. \quad \text{(A.88)}$$

$\square$

## A.9 PROOF OF LEMMA A.8 (BOUND ON TERM D)

**Lemma A.8** (Bound on Term D). The term D, which captures the error from gradient drift as defined in (A.20), is bounded by:

$$|D| \leq \frac{\eta_t^2 L^2 \sqrt{1-\beta_1}}{4(1+q_G)R} \left( \sum_{l=1}^{t-1} ||\mathbf{u}_{t-l}||_2^2 \sum_{k=l}^{t-1} \beta_1^k \sqrt{k} \right) + \frac{(1+q_G)R}{\sqrt{1-\beta_1}} \sum_{k=0}^{t-1} \left( \frac{\beta_1}{\beta_2} \right)^k \sqrt{k+1} ||\mathbf{U}_{t-k}||_2^2. \quad \text{(A.89)}$$

*Proof.* We start with the definition of Term D:

$$D = \sum_{i \in [d]} \sum_{k=0}^{t-1} \beta_1^k (\mathcal{G}_{t,i} - \mathcal{G}_{t-k,i}) \frac{g_{t-k,i} + \delta_{t-k,i}}{\sqrt{\epsilon + v'_{t,i}}}$$

To tackle this, we employ the weighted Young's inequality, which states that for any $\lambda > 0$,

$$xy \leq \frac{\lambda}{2}x^2 + \frac{1}{2\lambda}y^2 \tag{A.90}$$

We apply this inequality to each product within the summation for Term D, setting

$$x = |\mathcal{G}_{t,i} - \mathcal{G}_{t-k,i}|, \quad y = \frac{|g_{t-k,i} + \delta_{t-k,i}|}{\sqrt{\epsilon + v'_{t,i}}}, \quad \text{and} \quad \lambda = \frac{\sqrt{1-\beta_1}}{2(1+q_G)R\sqrt{k+1}}.$$

This application gives us an initial bound on the magnitude of D:

$$|D| \leq \sum_{i \in [d]} \sum_{k=0}^{t-1} \beta_1^k \left( \frac{\sqrt{1-\beta_1}}{4(1+q_G)R\sqrt{k+1}} (\mathcal{G}_{t,i} - \mathcal{G}_{t-k,i})^2 + \frac{(1+q_G)R\sqrt{k+1}}{\sqrt{1-\beta_1}} \frac{(g_{t-k,i} + \delta_{t-k,i})^2}{\epsilon + v'_{t,i}} \right). \tag{A.91}$$

To simplify this expression further, we must establish bounds for two of its key components.

First, we can find a lower bound for the denominator term. For any coordinate $i \in [d]$, the recursive definition of $v'_{t,i}$ implies that $\epsilon + v'_{t,i} \geq \epsilon + \beta_2^k v'_{t-k,i} \geq \beta_2^k(\epsilon + v'_{t-k,i})$. This allows us to bound the fraction as:

$$\frac{(g_{t-k,i} + \delta_{t-k,i})^2}{\epsilon + v'_{t,i}} \leq \frac{1}{\beta_2^k} U_{t-k,i}^2. \tag{A.92}$$

Second, we bound the squared gradient difference using the L-smoothness of the objective function $F$.

$$||\mathcal{G}_t - \mathcal{G}_{t-k}||_2^2 \leq L^2 ||\mathbf{w}_{t-1} - \mathbf{w}_{t-k-1}||_2^2 = L^2 \left\| \sum_{l=1}^{k} \eta_{t-l} \mathbf{u}_{t-l} \right\|_2^2$$

$$\leq \eta_t^2 L^2 k \sum_{l=1}^{k} ||\mathbf{u}_{t-l}||_2^2. \tag{A.93}$$

The final step above follows from Jensen's inequality and the fact that the step size schedule $\eta_t$ is non-decreasing.

With these two intermediate results, (A.92) and (A.93), we can return to our main inequality (A.91). Substituting these bounds yields:

$$|D| \leq \left( \sum_{k=0}^{t-1} \frac{\eta_t^2 L^2 \sqrt{1-\beta_1} \beta_1^k}{4(1+q_G)R\sqrt{k+1}} \left( k \sum_{l=1}^{k} ||\mathbf{u}_{t-l}||_2^2 \right) \right) + \left( \sum_{k=0}^{t-1} \frac{(1+q_G)R\beta_1^k \sqrt{k+1}}{\sqrt{1-\beta_1}\beta_2^k} ||\mathbf{U}_{t-k}||_2^2 \right)$$

$$\leq \frac{\eta_t^2 L^2 \sqrt{1-\beta_1}}{4(1+q_G)R} \left( \sum_{k=0}^{t-1} \beta_1^k \sqrt{k} \sum_{l=1}^{k} ||\mathbf{u}_{t-l}||_2^2 \right) + \frac{(1+q_G)R}{\sqrt{1-\beta_1}} \sum_{k=0}^{t-1} \left( \frac{\beta_1}{\beta_2} \right)^k \sqrt{k+1} ||\mathbf{U}_{t-k}||_2^2.$$

Finally, by rearranging the order of summation in the first term, we arrive at our desired bound:

$$|D| \leq \frac{\eta_t^2 L^2 \sqrt{1-\beta_1}}{4(1+q_G)R} \left( \sum_{l=1}^{t-1} ||\mathbf{u}_{t-l}||_2^2 \sum_{k=l}^{t-1} \beta_1^k \sqrt{k} \right) + \frac{(1+q_G)R}{\sqrt{1-\beta_1}} \sum_{k=0}^{t-1} \left( \frac{\beta_1}{\beta_2} \right)^k \sqrt{k+1} ||\mathbf{U}_{t-k}||_2^2. \tag{A.94}$$

$\square$

## A.10 PROOF OF LEMMA A.9 (LOWER BOUND ON TERM C)

**Lemma A.9** (Lower Bound on the Expectation of Term C). The expectation of term C, defined in (A.20), is lower-bounded by:

$$\mathbb{E}[C] \geq \frac{1}{2} \left( \sum_{i \in [d]} \sum_{k=0}^{t-1} \beta_1^k \mathbb{E} \left[ \frac{\mathcal{G}_{t-k,i}^2}{\sqrt{\epsilon + \widetilde{v}_{t,k+1,i}}} \right] \right) - \frac{2(1+q_G)R}{\sqrt{1-\beta_1}} \left( \sum_{i \in [d]} \sum_{k=0}^{t-1} \left( \frac{\beta_1}{\beta_2} \right)^k \sqrt{k+1} \mathbb{E} \left[ ||\mathbf{U}_{t-k}||_2^2 \right] \right)$$

$$- d \sum_{k=0}^{t-1} \beta_1^k M_{t-k}. \tag{A.95}$$

where $M_{t-k} = \frac{q_G R^2 + L q_W R ||\mathbf{w}_{t-k-1}||_2}{\sqrt{\epsilon}}$.

*Proof.* We study the main term of the summation in C, i.e. for $i \in [d]$ and $k < t$:

$$\mathbb{E}\left[\mathcal{G}_{t-k,i}\frac{g_{t-k,i}+\delta_{t-k,i}}{\sqrt{\epsilon+v_{t,i}}}\right] = \mathbb{E}\left[\nabla_i F(\mathbf{w}_{t-k-1})\frac{\nabla_i f_{t-k}(\mathbf{w}_{t-k-1})+\delta_{t-k,i}}{\sqrt{\epsilon+v_{t,i}}}\right]. \tag{A.96}$$

We will further drop indices in the rest of the proof, noting $\mathcal{G} = \mathcal{G}_{t-k,i}$, $g = g_{t-k,i}$, $\delta = \delta_{t-k,i}$, $\widetilde{v} = \widetilde{v}_{t,k+1,i}$ and $v = v'_{t,i}$. Finally, let us note

$$s^2 = \sum_{j=t-k}^{t} \beta_2^{t-j}(g_{j,i}+\delta_{j,i})^2 \qquad \text{and} \qquad r^2 = \mathbb{E}_{t-k-1}\left[s^2\right]. \tag{A.97}$$

In particular we have $\widetilde{v} - v = r^2 - s^2$. With our new notations, we can rewrite (A.96) as

$$\mathbb{E}\left[\mathcal{G}\frac{g+\delta}{\sqrt{\epsilon+v}}\right] = \mathbb{E}\left[\mathcal{G}\frac{g+\delta}{\sqrt{\epsilon+\widetilde{v}}} + \mathcal{G}(g+\delta)\left(\frac{1}{\sqrt{\epsilon+v}} - \frac{1}{\sqrt{\epsilon+\widetilde{v}}}\right)\right]$$

$$= \mathbb{E}\left[\mathbb{E}_{t-k-1}\left[\mathcal{G}\frac{g+\delta}{\sqrt{\epsilon+\widetilde{v}}}\right] + \mathcal{G}(g+\delta)\frac{r^2-s^2}{\sqrt{\epsilon+v}\sqrt{\epsilon+\widetilde{v}}(\sqrt{\epsilon+v}+\sqrt{\epsilon+\widetilde{v}})}\right]$$

$$= \mathbb{E}\left[\frac{\mathcal{G}^2}{\sqrt{\epsilon+\widetilde{v}}}\right] + \mathbb{E}\left[\frac{\mathcal{G}\mathbb{E}_{t-k-1}\left[\delta\right]}{\sqrt{\epsilon+\widetilde{v}}}\right] + \mathbb{E}\left[\underbrace{\mathcal{G}(g+\delta)\frac{r^2-s^2}{\sqrt{\epsilon+v}\sqrt{\epsilon+\widetilde{v}}(\sqrt{\epsilon+v}+\sqrt{\epsilon+\widetilde{v}})}}_{E}\right]$$

$$\geq \mathbb{E}\left[\frac{\mathcal{G}^2}{\sqrt{\epsilon+\widetilde{v}}}\right] - \frac{q_G R^2 + L q_W R ||\mathbf{w}_{t-k-1}||_2}{\sqrt{\epsilon}} + \mathbb{E}\left[E\right]. \tag{A.98}$$

The inequality uses Lemma A.7 and the bound for $||\nabla F(\cdot)||_\infty$. We denote $\frac{q_G R^2 + L q_W R ||\mathbf{w}_{t-k-1}||_2}{\sqrt{\epsilon}} \triangleq M_{t-k}$.

Then we focus on $E$:

$$|E| \leq \underbrace{|\mathcal{G}(g+\delta)|\frac{r^2}{\sqrt{\epsilon+v}(\epsilon+\widetilde{v})}}_{\kappa} + \underbrace{|\mathcal{G}(g+\delta)|\frac{s^2}{(\epsilon+v)\sqrt{\epsilon+\widetilde{v}}}}_{\rho},$$

due to the fact that $\sqrt{\epsilon+v} + \sqrt{\epsilon+\widetilde{v}} \geq \max(\sqrt{\epsilon+v}, \sqrt{\epsilon+\widetilde{v}})$ and $|r^2 - s^2| \leq r^2 + s^2$.

Applying Young's inequality to $\kappa$ with

$$\lambda = \frac{\sqrt{1-\beta_1}\sqrt{\epsilon+\widetilde{v}}}{2}, \quad x = \frac{|\mathcal{G}|}{\sqrt{\epsilon+\widetilde{v}}}, \quad y = \frac{|g+\delta|r^2}{\sqrt{\epsilon+\widetilde{v}}\sqrt{\epsilon+v}},$$

we obtain

$$\kappa \leq \frac{\mathcal{G}^2}{4\sqrt{\epsilon+\widetilde{v}}} + \frac{1}{\sqrt{1-\beta_1}}\frac{(g+\delta)^2 r^4}{(\epsilon+\widetilde{v})^{3/2}(\epsilon+v)}.$$

Given that $\epsilon + \widetilde{v} \geq r^2$ and taking the conditional expectation, we can simplify as

$$\mathbb{E}_{t-k-1}\left[\kappa\right] \leq \frac{\mathcal{G}^2}{4\sqrt{\epsilon+\widetilde{v}}} + \frac{1}{\sqrt{1-\beta_1}}\frac{r^2}{\sqrt{\epsilon+\widetilde{v}}}\mathbb{E}_{t-k-1}\left[\frac{(g+\delta)^2}{\epsilon+v}\right]. \tag{A.99}$$

Now turning to $\rho$, we use Young's inequality with

$$\lambda = \frac{\sqrt{1-\beta_1}\sqrt{\epsilon+\widetilde{v}}}{2r^2}, \quad x = \frac{|\mathcal{G}s|}{\sqrt{\epsilon+\widetilde{v}}}, \quad y = \frac{|s(g+\delta)|}{\epsilon+v},$$

we obtain

$$\rho \leq \frac{\mathcal{G}^2}{4\sqrt{\epsilon + \widetilde{v}}} \frac{s^2}{r^2} + \frac{1}{\sqrt{1 - \beta_1}} \frac{r^2}{\sqrt{\epsilon + \widetilde{v}}} \frac{(g + \delta)^2 s^2}{(\epsilon + v)^2}. \tag{A.100}$$

Given that $\epsilon + v \geq s^2$, and $\mathbb{E}_{t-k-1}\left[\frac{s^2}{r^2}\right] = 1$, we obtain after taking the conditional expectation,

$$\mathbb{E}_{t-k-1}\left[\rho\right] \leq \frac{\mathcal{G}^2}{4\sqrt{\epsilon + \widetilde{v}}} + \frac{1}{\sqrt{1 - \beta_1}} \frac{r^2}{\sqrt{\epsilon + \widetilde{v}}} \mathbb{E}_{t-k-1}\left[\frac{(g + \delta)^2}{\epsilon + v}\right]. \tag{A.101}$$

Notice that in (A.100), we possibly divide by zero. It suffice to notice that if $r^2 = 0$ then $s^2 = 0$ a.s. so that $\rho = 0$ and (A.101) is still verified. Summing (A.99) and (A.101), we get

$$\mathbb{E}_{t-k-1}\left[|E|\right] \leq \frac{\mathcal{G}^2}{2\sqrt{\epsilon + \widetilde{v}}} + \frac{2}{\sqrt{1 - \beta_1}} \frac{r^2}{\sqrt{\epsilon + \widetilde{v}}} \mathbb{E}_{t-k-1}\left[\frac{(g + \delta)^2}{\epsilon + v}\right]. \tag{A.102}$$

Given that $r \leq \sqrt{\epsilon + \widetilde{v}}$ by definition of $\widetilde{v}$, and that $r \leq \sqrt{k+1}(1 + q_G)R$, reintroducing the indices we had dropped

$$\mathbb{E}_{t-k-1}\left[|E|\right] \leq \frac{\mathcal{G}_{t-k,i}^2}{2\sqrt{\epsilon + \widetilde{v}_{t,k+1,i}}} + \frac{2(1 + q_G)R}{\sqrt{1 - \beta_1}} \sqrt{k+1} \mathbb{E}_{t-k-1}\left[\frac{(g_{t-k,i} + \delta_{t-k,i})^2}{\epsilon + v_{t,i}'}\right]. \tag{A.103}$$

Taking the complete expectation and using that by definition $\epsilon + v_{t,i}' \geq \epsilon + \beta_2^k v_{t-k,i}' \geq \beta_2^k(\epsilon + v_{t-k,i}')$ we get

$$\mathbb{E}\left[|E|\right] \leq \frac{1}{2}\mathbb{E}\left[\frac{\mathcal{G}_{t-k,i}^2}{\sqrt{\epsilon + \widetilde{v}_{t,k+1,i}}}\right] + \frac{2(1 + q_G)R}{\sqrt{1 - \beta_1}\beta_2^k} \sqrt{k+1} \mathbb{E}\left[\frac{(g_{t-k,i} + \delta_{t-k,i})^2}{\epsilon + v_{t-k,i}'}\right]. \tag{A.104}$$

Injecting (A.104) into (A.98) gives us

$$\sum_{i\in[d]} \sum_{k=0}^{t-1} \beta_1^k \mathbb{E}\left[\mathcal{G}_{t-k,i} \frac{g_{t-k,i} + \delta_{t-k,i}}{\sqrt{\epsilon + v_{t,i}}}\right]$$

$$\geq \sum_{i\in[d]} \sum_{k=0}^{t-1} \beta_1^k \left(\mathbb{E}\left[\frac{\mathcal{G}_{t-k,i}^2}{\sqrt{\epsilon + \widetilde{v}_{t,k+1,i}}}\right] - \mathbb{E}\left[|E|\right] - M_{t-k}\right) \tag{A.105}$$

$$\geq \sum_{i\in[d]} \sum_{k=0}^{t-1} \beta_1^k \left(\mathbb{E}\left[\frac{\mathcal{G}_{t-k,i}^2}{\sqrt{\epsilon + \widetilde{v}_{t,k+1,i}}}\right] - \left(\frac{1}{2}\mathbb{E}\left[\frac{\mathcal{G}_{t-k,i}^2}{\sqrt{\epsilon + \widetilde{v}_{t,k,i}}}\right] + \frac{2(1 + q_G)R\sqrt{k+1}}{\sqrt{1 - \beta_1}\beta_2^k} \mathbb{E}\left[\frac{(g_{t-k,i} + \delta_{t-k,i})^2}{\epsilon + v_{t-k,i}'}\right] + M_{t-k}\right)\right)$$

$$\geq \frac{1}{2}\left(\sum_{i\in[d]} \sum_{k=0}^{t-1} \beta_1^k \mathbb{E}\left[\frac{\mathcal{G}_{t-k,i}^2}{\sqrt{\epsilon + \widetilde{v}_{t,k+1,i}}}\right]\right) - \frac{2(1 + q_G)R}{\sqrt{1 - \beta_1}}\left(\sum_{i\in[d]} \sum_{k=0}^{t-1} \left(\frac{\beta_1}{\beta_2}\right)^k \sqrt{k+1}\mathbb{E}\left[||\mathbf{U}_{t-k}||_2^2\right]\right) - d\sum_{k=0}^{t-1} \beta_1^k M_{t-k}. \tag{A.106}$$

This is the desired lower bound for $\mathbb{E}\left[C\right]$. $\qquad\square$

## A.11 PROOF OF LEMMA A.10

**Lemma A.10** (Lemma A.2 in Défossez et al. (2022)). Assume we have $0 < \beta_1 < \beta_2 \leq 1$ and a sequence of real numbers $(a_n)_{n\in\mathbb{N}^*}$. We define for all $n \in \mathbb{N}^*$:

$$b_n = \sum_{j=1}^{n} \beta_2^{n-j} a_j^2 \quad \text{and} \quad c_n = \sum_{j=1}^{n} \beta_1^{n-j} a_j.$$

Then for any $\epsilon > 0$, we have the following inequality:

$$\sum_{j=1}^{n} \frac{c_j^2}{\epsilon + b_j} \leq \frac{1}{(1 - \beta_1)(1 - \beta_1/\beta_2)}\left(\ln\left(1 + \frac{b_n}{\epsilon}\right) - n\ln(\beta_2)\right). \tag{A.107}$$

*Proof.* First, we use Jensen's inequality on $c_j^2$, noting that $\sum_{l=1}^{j} \beta_1^{j-l} = \frac{1-\beta_1^j}{1-\beta_1} \le \frac{1}{1-\beta_1}$, to get:

$$c_j^2 = \left( \sum_{l=1}^{j} \beta_1^{j-l} a_l \right)^2 \le \left( \sum_{l=1}^{j} \beta_1^{j-l} \right) \left( \sum_{l=1}^{j} \beta_1^{j-l} a_l^2 \right) \le \frac{1}{1-\beta_1} \sum_{l=1}^{j} \beta_1^{j-l} a_l^2.$$

Dividing by $\epsilon + b_j$ and using the fact that for $l \le j$, $b_j \ge \beta_2^{j-l} b_l$, which implies $\epsilon + b_j \ge \beta_2^{j-l}(\epsilon + b_l)$, we obtain:

$$\frac{c_j^2}{\epsilon + b_j} \le \frac{1}{1-\beta_1} \sum_{l=1}^{j} \beta_1^{j-l} \frac{a_l^2}{\epsilon + b_j} \le \frac{1}{1-\beta_1} \sum_{l=1}^{j} \left( \frac{\beta_1}{\beta_2} \right)^{j-l} \frac{a_l^2}{\epsilon + b_l}. \tag{A.108}$$

Now, we sum over $j \in [n]$ and swap the order of summation:

$$\sum_{j=1}^{n} \frac{c_j^2}{\epsilon + b_j} \le \frac{1}{1-\beta_1} \sum_{j=1}^{n} \sum_{l=1}^{j} \left( \frac{\beta_1}{\beta_2} \right)^{j-l} \frac{a_l^2}{\epsilon + b_l} = \frac{1}{1-\beta_1} \sum_{l=1}^{n} \frac{a_l^2}{\epsilon + b_l} \sum_{j=l}^{n} \left( \frac{\beta_1}{\beta_2} \right)^{j-l}$$

$$\le \frac{1}{(1-\beta_1)(1-\beta_1/\beta_2)} \sum_{l=1}^{n} \frac{a_l^2}{\epsilon + b_l}, \tag{A.109}$$

where the last step uses the sum of a geometric series, since $\beta_1/\beta_2 < 1$.

The next step is to bound the final sum. Let's denote $x_l = a_l^2$. The sum is $\sum_{l=1}^{n} \frac{x_l}{\epsilon + b_l}$, where $b_l = \sum_{k=1}^{l} \beta_2^{l-k} x_k$. Note that $b_l - x_l = \beta_2 b_{l-1}$ (with $b_0 = 0$). Using the inequality $\frac{x}{y} \le \ln(y) - \ln(y-x)$ for $0 < x < y$, we have:

$$\frac{x_l}{\epsilon + b_l} \le \ln(\epsilon + b_l) - \ln(\epsilon + b_l - x_l)$$

$$= \ln(\epsilon + b_l) - \ln(\epsilon + \beta_2 b_{l-1})$$

$$= \ln \left( \frac{\epsilon + b_l}{\epsilon + b_{l-1}} \right) + \ln \left( \frac{\epsilon + b_{l-1}}{\epsilon + \beta_2 b_{l-1}} \right).$$

Summing from $l = 1$ to $n$, the first term forms a telescoping series equal to $\ln(\epsilon + b_n) - \ln(\epsilon) = \ln(1 + b_n/\epsilon)$. For the second term, since $\beta_2 \le 1$ and $b_{l-1} \ge 0$, we have $\frac{\epsilon + \beta_2 b_{l-1}}{\epsilon + b_{l-1}} \ge \beta_2$, which implies $\ln \left( \frac{\epsilon + b_{l-1}}{\epsilon + \beta_2 b_{l-1}} \right) \le -\ln(\beta_2)$. Thus, summing over $l$ gives:

$$\sum_{l=1}^{n} \frac{a_l^2}{\epsilon + b_l} \le \ln \left( 1 + \frac{b_n}{\epsilon} \right) - n \ln(\beta_2). \tag{A.110}$$

This inequality is a useful result in itself, corresponding to the special case where $c_j^2$ is replaced by $a_j^2$ (or equivalently $\beta_1 \to 0$ and $a_j$ is replaced by $a_j^2$).

Finally, substituting the bound from (A.110) into (A.109) yields the desired result. □

## A.12 PROOF OF LEMMA A.11

**Lemma A.11** (Lemma A.3 in Défossez et al. (2022)). *For any scalar $\rho \in (0,1)$ and any integer $K \in \mathbb{N}$, the following bound holds for the finite geometric sum:*

$$\sum_{k=0}^{K-1} \rho^k \sqrt{k+1} \le \frac{2}{(1-\rho)^{3/2}}. \tag{A.111}$$

*Proof.* Let the sum be denoted by $S_K = \sum_{k=0}^{K-1} \rho^k \sqrt{k+1}$. We analyze the term $(1-\rho)S_K$:

$$(1-\rho)S_K = \sum_{k=0}^{K-1} \rho^k \sqrt{k+1} - \sum_{j=1}^{K} \rho^j \sqrt{j}$$

$$= 1 + \sum_{k=1}^{K-1} \rho^k (\sqrt{k+1} - \sqrt{k}) - \rho^K \sqrt{K}.$$

By the concavity of the square root function, $\sqrt{k+1} - \sqrt{k} \le \frac{1}{2\sqrt{k}}$. This implies:

$$(1-\rho)S_K \le 1 + \sum_{k=1}^{K-1} \frac{\rho^k}{2\sqrt{k}} \le 1 + \int_0^\infty \frac{\rho^t}{2\sqrt{t}} \mathrm{d}t.$$

The integral is a standard Gaussian integral form which evaluates to $\frac{\sqrt{\pi}}{2\sqrt{-\ln(\rho)}}$. Using the inequality $-\ln(\rho) \ge 1 - \rho$, we have:

$$(1-\rho)S_K \le 1 + \frac{\sqrt{\pi}}{2\sqrt{1-\rho}} \le \frac{2}{\sqrt{1-\rho}}.$$

Dividing by $(1-\rho)$ yields the desired result. □

## A.13 PROOF OF LEMMA A.12

**Lemma A.12** (Lemma A.4 in Défossez et al. (2022)). For any scalar $\rho \in (0,1)$ and any integer $K \in \mathbb{N}$, the following bound holds:

$$\sum_{k=0}^{K-1} \rho^k \sqrt{k}(k+1) \le \frac{4\rho}{(1-\rho)^{5/2}}. \tag{A.112}$$

*Proof.* Let the sum be denoted by $S_K = \sum_{k=0}^{K-1} \rho^k \sqrt{k}(k+1)$. We proceed by analyzing $(1-\rho)S_K$:

$$(1-\rho)S_K = \sum_{k=1}^{K-1} \rho^k \left[ \sqrt{k}(k+1) - k\sqrt{k-1} \right] - \rho^K k\sqrt{K-1}$$
$$\le \sum_{k=1}^{K-1} \rho^k (2\sqrt{k}),$$

where the inequality holds because $\sqrt{k}(k+1) - k\sqrt{k-1} \le 2\sqrt{k}$. Re-indexing the sum gives:

$$(1-\rho)S_K \le 2\rho \sum_{k=1}^{K-1} \rho^{k-1}\sqrt{k} = 2\rho \sum_{j=0}^{K-2} \rho^j \sqrt{j+1}.$$

Applying the result from Lemma A.11 to the final sum, we get:

$$(1-\rho)S_K \le 2\rho \left( \frac{2}{(1-\rho)^{3/2}} \right) = \frac{4\rho}{(1-\rho)^{3/2}}.$$

Dividing both sides by $(1-\rho)$ completes the proof. □

## A.14 PROOF OF LEMMA A.13

**Lemma A.13** (Upper bound of $\sum_{t=1}^T \mathbb{E}\left[\|\mathbf{u}_t\|_2^2\right]$). Under the condition that $\beta_1(1+q_M) < \beta_2(1-q_V)$, the expected sum of squared updates over $T$ iterations is bounded by:

$$\sum_{t=1}^T \mathbb{E}\left[\|\mathbf{u}_t\|_2^2\right] \le \frac{d}{(1-\beta_1(1+q_M))(1-\frac{\beta_1(1+q_M)}{\beta_2(1-q_V)})}$$
$$\times \left( \ln\left(1 + \frac{((1+q_G)R)^2}{\epsilon(1-\beta_2(1-q_V))}\right) - T\ln(\beta_2(1-q_V)) \right). \tag{A.113}$$

*Proof.* The proof proceeds by first expanding the term of interest, applying bounds on the moment estimates derived from their recurrence relations, and then leveraging Lemma A.10 to bound the resulting sum.

We begin by expanding the definition of $||\mathbf{u}_t||_2^2$, separating the sum over the dimension $d$, and taking the expectation:

$$\sum_{t=1}^{T} \mathbb{E}\left[||\mathbf{u}_t||_2^2\right] = \sum_{t=1}^{T} \mathbb{E}\left[\sum_{i \in [d]} \frac{m_{t,i}^2}{\epsilon + v_{t,i}}\right] = \sum_{i \in [d]} \sum_{t=1}^{T} \mathbb{E}\left[\frac{m_{t,i}^2}{\epsilon + v_{t,i}}\right]. \tag{A.114}$$

For each coordinate $i$, we bound the numerator $m_{t,i}^2$ from above and the denominator $\epsilon + v_{t,i}$ from below. By unrolling the recurrence for $m_{t,i}$ and applying the triangle inequality along with the relative error model, we get an upper bound on its magnitude:

$$|m_{t,i}| \le \sum_{k=1}^{t} \beta_1^{t-k} (1 + q_M)^{t-k} |\nabla_i f_k(\mathbf{w}_{k-1}) + \delta_{k,i}|. \tag{A.115}$$

For the denominator, Lemma A.2 provides a lower bound for $v_{t,i}$:

$$v_{t,i} \ge \sum_{k=1}^{t} \beta_2^{t-k} (1 - q_V)^{t-k} (\nabla_i f_k(\mathbf{w}_{k-1}) + \delta_{k,i})^2. \tag{A.116}$$

Substituting these into the sum gives the inequality:

$$\sum_{t=1}^{T} \mathbb{E}\left[||\mathbf{u}_t||_2^2\right] \le \sum_{i \in [d]} \sum_{t=1}^{T} \mathbb{E}\left[\frac{\left(\sum_{k=1}^{t} \beta_1^{t-k}(1+q_M)^{t-k}|\widehat{g}_{k,i}|\right)^2}{\epsilon + \sum_{k=1}^{t} \beta_2^{t-k}(1-q_V)^{t-k}\widehat{g}_{k,i}^2}\right], \tag{A.117}$$

where for brevity we denote $\widehat{g}_{k,i} = \nabla_i f_k(\mathbf{w}_{k-1}) + \delta_{k,i}$.

The inner sum over $t$ for each coordinate $i$ in (A.117) perfectly matches the form required by Lemma A.10. To apply it, we make the following substitutions into the lemma's notation:

- Let the sequence $(a_j)_{j \in \mathbb{N}^*}$ be $a_k = |\widehat{g}_{k,i}|$.

- Let the effective decay factors be $\beta_1' = \beta_1(1 + q_M)$ and $\beta_2' = \beta_2(1 - q_V)$. The lemma's condition $\beta_1' < \beta_2'$ is satisfied by our assumption.

With these substitutions, the numerator term becomes $\left(\sum_{k=1}^{t}(\beta_1')^{t-k}a_k\right)^2 = c_t^2$ and the sum in the denominator becomes $\sum_{k=1}^{t}(\beta_2')^{t-k}a_k^2 = b_t$. Applying Lemma A.10 to the sum over $t$ for a fixed $i$ yields:

$$\sum_{t=1}^{T} \mathbb{E}\left[\frac{c_t^2}{\epsilon + b_t}\right] \le \frac{1}{(1-\beta_1')(1-\beta_1'/\beta_2')}\left(\ln\left(1 + \frac{b_T}{\epsilon}\right) - T\ln(\beta_2')\right). \tag{A.118}$$

The final step is to find an upper bound for $b_T$. By definition:

$$b_T = \sum_{k=1}^{T}(\beta_2')^{T-k}a_k^2 = \sum_{k=1}^{T}(\beta_2(1-q_V))^{T-k}\widehat{g}_{k,i}^2. \tag{A.119}$$

From Lemma A.6, we have a uniform bound on the quantized gradient estimator, $|\widehat{g}_{k,i}| \le (1+q_G)R$. Therefore:

$$b_T \le \sum_{k=1}^{T}(\beta_2(1-q_V))^{T-k}((1+q_G)R)^2$$

$$= ((1+q_G)R)^2 \sum_{j=0}^{T-1}(\beta_2(1-q_V))^j$$

$$\leq ((1 + q_G)R)^2 \frac{1}{1 - \beta_2(1 - q_V)}. \tag{A.120}$$

Substituting the bound for $b_T$ back into (A.118), and re-inserting the definitions of $\beta_1'$ and $\beta_2'$, we obtain the bound for a single coordinate $i$. As this bound is identical for all $d$ coordinates, we multiply by $d$ to get the final result stated in (A.113). This completes the proof. □

### A.15 PROOF OF LEMMA A.14 (BOUND ON TERM M)

**Lemma A.14** (Bound on Term M). The term M, representing the accumulated quantization bias from (A.26), is bounded by:

$$M \leq \frac{\eta_T dT}{\sqrt{\epsilon}(1 - \beta_1)} \left( q_G R^2 + Lq_W R ||\mathbf{w}_0||_2 \right) + \frac{\eta_T^2 dL q_W U R T^2}{2\sqrt{\epsilon}(1 - \beta_1)}, \tag{A.121}$$

where $U = \sqrt{\frac{d}{1 - \frac{\beta_1^2(1 + q_M)^2}{\beta_2(1 - q_V)}}}$.

*Proof.* First, we establish a uniform bound on the update norm $||\mathbf{u}_t||_2$ using Lemma A.4:

$$||\mathbf{u}_t||_2 = \sqrt{\sum_{i \in [d]} \frac{m_{t,i}^2}{\epsilon + v_{t,i}}}$$

$$\leq \sqrt{\sum_{i \in [d]} \frac{(\sum_{k=0}^{t} \beta_1^{t-k}(1 + q_M)^{t-k}|\nabla_i f_k(\mathbf{w}_{k-1}) + \delta_{k,i}|)^2}{(\sum_{k=0}^{t} \beta_2^{t-k}(1 - q_V)^{t-k}(\nabla_i f_k(\mathbf{w}_{k-1}) + \delta_{k,i})^2)}}$$

$$\leq \sqrt{\frac{d}{1 - \frac{\beta_1^2(1 + q_M)^2}{\beta_2(1 - q_V)}}} \triangleq U \tag{A.122}$$

Now, let's recall the definition of $M$:

$$M = \eta_T d \sum_{t=1}^{T} \mathbb{E}\left[ \sum_{k=0}^{t-1} \beta_1^k M_{t-k} \right], \quad \text{where} \quad M_{t-k} = \frac{q_G R^2 + Lq_W R ||\mathbf{w}_{t-k-1}||_2}{\sqrt{\epsilon}}.$$

We can split $M$ into two components: a constant part $M_{\text{const}}$ and a weight-dependent part $M_{\text{weights}}$.

$$M_{\text{const}} = \frac{\eta_T d q_G R^2}{\sqrt{\epsilon}} \sum_{t=1}^{T} \sum_{k=0}^{t-1} \beta_1^k$$

$$M_{\text{weights}} = \frac{\eta_T d L q_W R}{\sqrt{\epsilon}} \sum_{t=1}^{T} \sum_{k=0}^{t-1} \beta_1^k \mathbb{E}\left[ ||\mathbf{w}_{t-k-1}||_2 \right]$$

Bounding $M_{\text{const}}$ is straightforward. The inner sum is a geometric series bounded by $\frac{1}{1-\beta_1}$, so the double summation is bounded by $\frac{T}{1-\beta_1}$.

$$M_{\text{const}} \leq \frac{\eta_T d q_G R^2 T}{\sqrt{\epsilon}(1 - \beta_1)}. \tag{A.123}$$

The main challenge is to bound $M_{\text{weights}}$. To do this, we first need a bound on the expected weight norm $\mathbb{E}[||\mathbf{w}_j||_2]$. From the update rule $\mathbf{w}_j = \mathbf{w}_{j-1} - \eta_j \mathbf{u}_j$, we can unroll the recursion:

$$\mathbf{w}_j = \mathbf{w}_0 - \sum_{l=1}^{j} \eta_l \mathbf{u}_l.$$

Applying the triangle inequality and taking the expectation, we get:

$$\mathbb{E}[||\mathbf{w}_j||_2] \leq ||\mathbf{w}_0||_2 + \mathbb{E}\left[ \sum_{l=1}^{j} \eta_l ||\mathbf{u}_l||_2 \right].$$

Using the uniform bound $||\mathbf{u}_l||_2 \leq U$, we have:

$$\mathbb{E}\left[||\mathbf{w}_j||_2\right] \leq ||\mathbf{w}_0||_2 + U \sum_{l=1}^{j} \eta_l \leq ||\mathbf{w}_0||_2 + U j \eta_T.$$

Now we substitute this bound back into the expression for $M_{\text{weights}}$. We first swap the order of summation. Let $j = t - k - 1$. For a fixed $j \in \{0, \ldots, T-1\}$, the term $\mathbb{E}\left[||\mathbf{w}_j||_2\right]$ appears when $k = t - j - 1$. This is valid for $t$ from $j+1$ to $T$.

$$\sum_{t=1}^{T} \sum_{k=0}^{t-1} \beta_1^k \mathbb{E}\left[||\mathbf{w}_{t-k-1}||_2\right] = \sum_{j=0}^{T-1} \mathbb{E}\left[||\mathbf{w}_j||_2\right] \sum_{t=j+1}^{T} \beta_1^{t-j-1}$$

$$= \sum_{j=0}^{T-1} \mathbb{E}\left[||\mathbf{w}_j||_2\right] \sum_{m=0}^{T-j-1} \beta_1^m$$

$$\leq \frac{1}{1-\beta_1} \sum_{j=0}^{T-1} \mathbb{E}\left[||\mathbf{w}_j||_2\right].$$

Next, we substitute the linear bound:

$$\frac{1}{1-\beta_1} \sum_{j=0}^{T-1} \mathbb{E}\left[||\mathbf{w}_j||_2\right] \leq \frac{1}{1-\beta_1} \sum_{j=0}^{T-1} \left(||\mathbf{w}_0||_2 + j \cdot U \cdot \eta_T\right)$$

$$= \frac{1}{1-\beta_1} \left(T||\mathbf{w}_0||_2 + U \eta_T \sum_{j=0}^{T-1} j\right).$$

The sum of the first $T-1$ integers is $\frac{(T-1)T}{2} < \frac{T^2}{2}$. This gives:

$$\leq \frac{1}{1-\beta_1} \left(T||\mathbf{w}_0||_2 + \frac{T^2}{2} U \eta_T\right).$$

Finally, we assemble the complete bound for $M_{\text{weights}}$:

$$M_{\text{weights}} \leq \frac{\eta_T d L q_W R}{\sqrt{\epsilon}(1-\beta_1)} \left(T||\mathbf{w}_0||_2 + \frac{T^2}{2} U \eta_T\right).$$

Combining $M_{\text{const}}$ with $M_{\text{weights}}$, we get the final bound for $M$.

$$M \leq \frac{\eta_T d T}{\sqrt{\epsilon}(1-\beta_1)} \left(q_G R^2 + L q_W R ||\mathbf{w}_0||_2\right) + \frac{\eta_T^2 d L q_W U R T^2}{2\sqrt{\epsilon}(1-\beta_1)}$$

(A.124)

$\square$

# B  PROOF OF THEOREM 4.6

## B.1  PRELIMINARIES

The momentum of the quantized Muon (Algorithm 1, 3) is defined as

$$\mathbf{M}_t = \beta \mathbf{M}_{t-1}^Q + (1-\beta) \frac{1}{B} \sum_{i=1}^{B} \nabla^Q f(\mathbf{W}_t^Q; \boldsymbol{\xi}_{t,i}), \qquad \mathbf{M}_0 = \frac{1}{B} \sum_{i=1}^{B} \nabla^Q f(\mathbf{W}_0^Q; \boldsymbol{\xi}_{0,i}). \qquad \text{(B.1)}$$

We define the following auxiliary variables for analysis:

$$\mathbf{C}_t = \beta \mathbf{C}_{t-1} + (1-\beta) \nabla F(\mathbf{W}_t), \qquad\qquad\qquad \mathbf{C}_0 = \nabla F(\mathbf{W}_0) \qquad \text{(B.2)}$$

$$\mathbf{X}_t = \beta \mathbf{X}_{t-1} + (1-\beta) \frac{1}{B} \sum_{i=1}^{B} \nabla f(\mathbf{W}_t; \boldsymbol{\xi}_{t,i}), \qquad \mathbf{X}_0 = \frac{1}{B} \sum_{i=1}^{B} \nabla f(\mathbf{W}_0; \boldsymbol{\xi}_{0,i}) \qquad \text{(B.3)}$$

$$\mathbf{Y}_t = \beta\mathbf{Y}_{t-1} + (1-\beta)\frac{1}{B}\sum_{i=1}^{B}\nabla^Q f(\mathbf{W}_t;\boldsymbol{\xi}_{t,i}), \qquad \mathbf{Y}_0 = \frac{1}{B}\sum_{i=1}^{B}\nabla^Q f(\mathbf{W}_0;\boldsymbol{\xi}_{0,i}) \qquad \text{(B.4)}$$

$$\mathbf{Z}_t = \beta\mathbf{Z}_{t-1} + (1-\beta)\frac{1}{B}\sum_{i=1}^{B}\nabla^Q f(\mathbf{W}_t^Q;\boldsymbol{\xi}_{t,i}), \qquad \mathbf{Z}_0 = \frac{1}{B}\sum_{i=1}^{B}\nabla^Q f(\mathbf{W}_0^Q;\boldsymbol{\xi}_{0,i}). \qquad \text{(B.5)}$$

We also define the following relative quantization errors $q_G, q_W, q_M$ according to Assumption 3.1 and Lemma B.2, i.e., for any $t \in \{0, 1, \dots, T-1\}$ and $i \in \{1, 2, \dots, B\}$,

$$\|\nabla^Q f(\mathbf{W}_t;\boldsymbol{\xi}_{t,i}) - \nabla f(\mathbf{W}_t;\boldsymbol{\xi}_{t,i})\|_F \leq q_G\|\nabla f(\mathbf{W}_t;\boldsymbol{\xi}_{t,i})\|_F,$$

$$\|\nabla^Q F(\mathbf{W}_t) - \nabla F(\mathbf{W}_t)\|_F \leq q_G\|\nabla F(\mathbf{W}_t)\|_F,$$

$$\|\nabla^Q f(\mathbf{W}_t^Q;\boldsymbol{\xi}_{t,i}) - \nabla f(\mathbf{W}_t^Q;\boldsymbol{\xi}_{t,i})\|_F \leq q_G\|\nabla f(\mathbf{W}_t^Q;\boldsymbol{\xi}_{t,i})\|_F,$$

$$\|\nabla^Q F(\mathbf{W}_t^Q) - \nabla F(\mathbf{W}_t^Q)\|_F \leq q_G\|\nabla F(\mathbf{W}_t^Q)\|_F,$$

$$\|\mathbf{W}_t^Q - \mathbf{W}_t\|_F \leq q_W\|\mathbf{W}_t\|_F,$$

$$\|\mathbf{M}_t^Q - \mathbf{M}_t\|_F \leq q_M\|\mathbf{M}_t\|_F. \qquad \text{(B.6)}$$

## B.2 Proof of Theorem 4.6

*Proof.* Set $\eta_t = \eta$, denote $r = \min\{m, n\}$, and according to the $L$-smoothness of $F(\cdot)$, we have

$$\mathbb{E}[F(\mathbf{W}_t) - F(\mathbf{W}_{t+1})]$$

$$\geq \mathbb{E}[\langle\nabla F(\mathbf{W}_t), \mathbf{W}_t - \mathbf{W}_{t+1}\rangle - \frac{L}{2}\|\mathbf{W}_t - \mathbf{W}_{t+1}\|_F^2]$$

$$= \mathbb{E}[\eta\langle\nabla F(\mathbf{W}_t), \mathbf{U}_t\mathbf{V}_t^\top\rangle - \frac{L}{2}\eta^2\|\mathbf{U}_t\mathbf{V}_t^\top\|_F^2]$$

$$\geq \mathbb{E}[\eta\langle\nabla F(\mathbf{W}_t), \mathbf{U}_t\mathbf{V}_t^\top\rangle] - \frac{L}{2}\eta^2 r$$

$$= \mathbb{E}[\eta\langle\mathbf{M}_t, \mathbf{U}_t\mathbf{V}_t^\top\rangle + \eta\langle\nabla F(\mathbf{W}_t) - \mathbf{M}_t, \mathbf{U}_t\mathbf{V}_t^\top\rangle] - \frac{L}{2}\eta^2 r$$

$$\geq \mathbb{E}[\eta\|\mathbf{M}_t\|_* - \eta\|\nabla F(\mathbf{W}_t) - \mathbf{M}_t\|_F \cdot \|\mathbf{U}_t\mathbf{V}_t^\top\|_F] - \frac{L}{2}\eta^2 r$$

$$\geq \mathbb{E}[\eta\|\nabla F(\mathbf{W}_t)\|_* - \eta\|\nabla F(\mathbf{W}_t) - \mathbf{M}_t\|_* - \eta\sqrt{r}\|\nabla F(\mathbf{W}_t) - \mathbf{M}_t\|_F] - \frac{L}{2}\eta^2 r$$

$$\geq \mathbb{E}[\eta\|\nabla F(\mathbf{W}_t)\|_* - 2\eta\sqrt{r}\|\nabla F(\mathbf{W}_t) - \mathbf{M}_t\|_F] - \frac{L}{2}\eta^2 r. \qquad \text{(B.7)}$$

The second inequality is due to $\|\mathbf{U}_t\mathbf{V}_t^\top\|_F^2 = \mathrm{tr}(\mathbf{V}_t\mathbf{U}_t^\top\mathbf{U}_t\mathbf{V}_t^\top) \leq r = \min\{m, n\}$. The third inequality is due to $\mathbf{M}_t = \mathbf{U}_t\mathbf{S}_t\mathbf{V}_t^\top$ and Cauchy-Schwarz inequality. The last inequality we used the fact that $\|\mathbf{A}\|_* \leq \sqrt{r}\|\mathbf{A}\|_F$ for any $\mathbf{A} \in \mathbb{R}^{m \times n}$.

Summing Eq. (B.7) over $t = 0, 1, \dots, T-1$, we get

$$\frac{1}{T}\sum_{t=0}^{T-1}\mathbb{E}[\|\nabla F(\mathbf{W}_t)\|_*]$$

$$\leq \frac{\mathbb{E}[F(\mathbf{W}_0) - F(\mathbf{W}_T)]}{T\eta} + \frac{2\sqrt{r}}{T}\sum_{t=0}^{T-1}\mathbb{E}[\|\nabla F(\mathbf{W}_t) - \mathbf{M}_t\|_F] + \frac{\eta Lr}{2}. \qquad \text{(B.8)}$$

Next, we focus on term $\mathbb{E}[\|\nabla F(\mathbf{W}_t) - \mathbf{M}_t\|_F]$. With auxiliary variables defined in Eq. (B.2)-(B.5), we have

$$\mathbb{E}[\|\nabla F(\mathbf{W}_t) - \mathbf{M}_t\|_F]$$

$$\leq \mathbb{E}[\|\nabla F(\mathbf{W}_t) - \mathbf{C}_t\|_F + \|\mathbf{C}_t - \mathbf{X}_t\|_F + \|\mathbf{X}_t - \mathbf{Y}_t\|_F + \|\mathbf{Y}_t - \mathbf{Z}_t\|_F + \|\mathbf{Z}_t - \mathbf{M}_t\|_F].$$

By Lemmas B.3, B.4, B.5, B.6, and B.7, we have

$$\mathbb{E}[\|\nabla F(\mathbf{W}_t) - \mathbf{M}_t\|_F]$$
$$\leq \mathbb{E}[\|\nabla F(\mathbf{W}_t) - \mathbf{C}_t\|_F + \|\mathbf{C}_t - \mathbf{X}_t\|_F + \|\mathbf{X}_t - \mathbf{Y}_t\|_F + \|\mathbf{Y}_t - \mathbf{Z}_t\|_F + \|\mathbf{Z}_t - \mathbf{M}_t\|_F]$$
$$\leq \frac{\beta L\eta\sqrt{r}}{1-\beta} + \beta^t \frac{3\sigma}{\sqrt{B}} + \sqrt{\frac{1-\beta}{1+\beta}} \frac{3\sigma}{\sqrt{B}} + 3q_G(\sigma + G) + 3q_G T\eta\sqrt{r}L + q_W(1+q_G)DL+$$
$$q_W(1+q_G)T\eta\sqrt{r}L + \frac{q_M\beta}{1-\beta(1+q_M)}\left(\frac{\sigma}{\sqrt{B}} + \sqrt{\frac{1-\beta}{1+\beta}} \cdot \frac{\sigma}{\sqrt{B}} + G + q_G(\sigma + G) +\right.$$
$$\left. q_W(1+q_G)DL + (1+q_W)(1+q_G)T\eta\sqrt{r}L\right).$$

Summing over $t = 0, 1, \ldots, T - 1$, we get

$$\frac{1}{T}\sum_{t=0}^{T-1}\mathbb{E}[\|\nabla F(\mathbf{W}_t) - \mathbf{M}_t\|_F]$$
$$\leq \frac{\beta L\eta\sqrt{r}}{1-\beta} + \frac{3\sigma}{T(1-\beta)\sqrt{B}} + \sqrt{\frac{1-\beta}{1+\beta}} \frac{3\sigma}{\sqrt{B}} + 3q_G(\sigma + G) + 3q_G T\eta\sqrt{r}L + q_W(1+q_G)DL+$$
$$q_W(1+q_G)T\eta\sqrt{r}L + \frac{q_M\beta}{1-\beta(1+q_M)}\left(\frac{\sigma}{\sqrt{B}} + \sqrt{\frac{1-\beta}{1+\beta}} \cdot \frac{\sigma}{\sqrt{B}} + G + q_G(\sigma + G) +\right.$$
$$\left. q_W(1+q_G)DL + (1+q_W)(1+q_G)T\eta\sqrt{r}L\right). \tag{B.9}$$

Substitute (B.9) into (B.8), with Assumption 3.1, we have

$$\frac{1}{T}\sum_{t=0}^{T-1}\mathbb{E}[\|\nabla F(\mathbf{W}_t)\|_*]$$
$$\leq \frac{\mathbb{E}[F(\mathbf{W}_0) - F(\mathbf{W}_T)]}{\eta T} + \frac{L\eta r}{2} + \frac{2\sqrt{r}}{T}\sum_{t=0}^{T-1}\mathbb{E}[\|\nabla F(\mathbf{W}_t) - \mathbf{M}_t\|_F]$$
$$\leq \frac{\mathbb{E}[F(\mathbf{W}_0) - F(\mathbf{W}_T)]}{\eta T} + \frac{L\eta r}{2} + \frac{2\beta L\eta r}{1-\beta} + \frac{6\sigma\sqrt{r}}{T(1-\beta)\sqrt{B}} + \sqrt{\frac{1-\beta}{1+\beta}} \frac{6\sigma\sqrt{r}}{\sqrt{B}}+$$
$$6q_G\sqrt{r}(\sigma + G) + 6q_G T\eta r L + 2q_W(1+q_G)DL\sqrt{r} + 2q_W(1+q_G)T\eta r L+$$
$$\frac{2q_M\beta\sqrt{r}}{1-\beta(1+q_M)}\left(\frac{\sigma}{\sqrt{B}} + \sqrt{\frac{1-\beta}{1+\beta}} \cdot \frac{\sigma}{\sqrt{B}} + G + q_G(\sigma + G) +\right.$$
$$\left. q_W(1+q_G)DL + (1+q_W)(1+q_G)T\eta\sqrt{r}L\right)$$
$$\leq \frac{\mathbb{E}[F(\mathbf{W}_0) - F(\mathbf{W}_T)]}{\eta T} + \frac{L\eta r}{2} + \frac{2\beta L\eta r}{1-\beta} + \frac{6\sigma\sqrt{r}}{T(1-\beta)\sqrt{B}} + \sqrt{\frac{1-\beta}{1+\beta}} \frac{6\sigma\sqrt{r}}{\sqrt{B}}+$$
$$\Theta\left(q_G + q_W + q_G T\eta + q_W T\eta + \frac{q_M\beta}{1-\beta(1+q_M)}\left(1 + \sqrt{1-\beta} + q_G + q_W + T\eta\right)\right).$$

Let $F(\mathbf{W}_0) - F^* \leq \Delta$, where $\Delta > 0$ is a constant. By setting $B = 1$, $1 - \beta = \Theta(T^{-1/2})$, $\eta = \Theta((1-\beta)^{1/2}T^{-1/2})$, we have $T\eta = \Theta(T^{1/4})$. Then we have

$$\frac{\mathbb{E}[F(\mathbf{W}_0) - F(\mathbf{W}_T)]}{\eta T} + \frac{L\eta r}{2} + \frac{2\beta L\eta r}{1-\beta} + \frac{6\sigma\sqrt{r}}{T(1-\beta)\sqrt{B}} + \sqrt{\frac{1-\beta}{1+\beta}} \frac{6\sigma\sqrt{r}}{\sqrt{B}} = \mathcal{O}(\frac{1}{T^{1/4}}).$$

Moreover, with condition $\beta(1+q_M) < 1$, suppose $1-\beta = C_\beta T^{-1/2}$, $C_\beta > 0$ is a constant. Choose $q_M = C_M T^{-1/2}$, where $C_M < C_\beta, C_M > 0$ is a constant, then we have

$$\beta(1 + q_M) = (1 - C_\beta T^{-1/2})(1 + C_M T^{-1/2}) = 1 - (C_\beta - C_M)T^{-1/2} - C_\beta C_M T^{-1} < 1.$$

Thus, by setting $q_G = \mathcal{O}(T^{-1/2})$, $q_W = \mathcal{O}(T^{-1/2})$, $q_M = \mathcal{O}(T^{-1/2})$, we have

$$q_G + q_W + q_G T\eta + q_W T\eta + \frac{q_M\beta}{1 - \beta(1+q_M)}\left(1 + \sqrt{1-\beta} + q_G + q_W + T\eta\right)$$
$$=\mathcal{O}(T^{-1/2} + T^{-1/2}T^{1/4} + T^{-1/2}(1 + T^{-1/4} + T^{-1/2} + T^{1/4}))$$
$$=\mathcal{O}(T^{-1/4}),$$

where we used the fact $\frac{q_M\beta}{1-\beta(1+q_M)} = \mathcal{O}(q_M\beta(1 + \beta(1+q_M))) = \mathcal{O}(T^{-1/2})$, and $\beta(1+q_M) < 1$.

Combining the above results, with the fact that $\|\mathbf{A}\|_* \geq \|\mathbf{A}\|_F$ for any matrix $\mathbf{A}$, we complete the proof.

$\square$

### B.3 Proof of Lemma B.1

**Lemma B.1** (Bound of $\|\mathbf{W}\|_F$ and $\|\nabla F(\mathbf{W})\|_F$ for Muon). Suppose Assumptions 4.3 and 4.4 hold. The iterates of Muon satisfy that for any $t \geq 0$,

$$\|\mathbf{W}_t\|_F \leq D + t\eta\sqrt{r}, \quad \|\nabla F(\mathbf{W}_t)\|_F \leq G + t\eta\sqrt{r}L.$$

*Proof of Lemma B.1.* According to the update of Muon, we have

$$\|\mathbf{W}_t\|_F$$
$$=\|\mathbf{W}_{t-1} - \eta\mathbf{U}_t\mathbf{V}_t^\top\|_F$$
$$\leq\|\mathbf{W}_{t-1}\|_F + \eta\|\mathbf{U}_t\mathbf{V}_t^\top\|_F$$
$$=\|\mathbf{W}_{t-1}\|_F + \eta\sqrt{\text{tr}(\mathbf{V}_t\mathbf{U}_t^\top\mathbf{U}_t\mathbf{V}_t^\top)}$$
$$\leq\|\mathbf{W}_{t-1}\|_F + \eta\sqrt{r}$$
$$\leq\|\mathbf{W}_0\|_F + t\eta\sqrt{r}$$
$$\leq D + t\eta\sqrt{r}.$$

The third inequality is because $\mathbf{U}_t$ and $\mathbf{V}_t$ are orthogonal matrices, and the last inequality is due to Assumption 4.4.

$$\|\nabla F(\mathbf{W}_t)\|_F$$
$$\leq\|\nabla F(\mathbf{W}_0)\|_F + \sum_{k=0}^{t-1}\|\nabla F(\mathbf{W}_{k+1}) - \nabla F(\mathbf{W}_k)\|_F$$
$$\leq G + \sum_{k=0}^{t-1}L\|\mathbf{W}_{k+1} - \mathbf{W}_k\|_F$$
$$\leq G + \sum_{k=0}^{t-1}L\eta\sqrt{r}$$
$$=G + t\eta\sqrt{r}L.$$

The first inequality is due to the triangle inequality, the second inequality is due to Assumption 4.3, and the last inequality is due to the update of Muon. $\square$

### B.4 PROOF OF LEMMA B.2

**Lemma B.2.** Suppose Assumption 3.1 holds. For any matrix $\mathbf{X} \in \mathbb{R}^{m \times n}$ and its quantized version $\mathbf{X}^Q$, we have

$$\|\mathbf{X}^Q - \mathbf{X}\|_F \leq q\|\mathbf{X}\|_F.$$

*Proof of Lemma B.2.* According to Assumption 3.1, we have

$$\begin{aligned}
\|\mathbf{X}^Q - \mathbf{X}\|_F^2 &= \sum_{i=1}^m \sum_{j=1}^n |X_{ij}^Q - X_{ij}|^2 \\
&\leq \sum_{i=1}^m \sum_{j=1}^n q^2 |X_{ij}|^2 \\
&= q^2 \|\mathbf{X}\|_F^2.
\end{aligned}$$

Taking the square root on both sides, we complete the proof. $\qquad\square$

### B.5 PROOF OF LEMMA B.3

**Lemma B.3.** Suppose Assumptions 4.3 and 4.4 hold. For any $t \geq 0$, we have

$$\mathbb{E}[\|\nabla F(\mathbf{W}_t) - \mathbf{C}_t\|_F] \leq \frac{\beta L \eta \sqrt{r}}{1 - \beta}.$$

*Proof of Lemma B.3.* This proof is a standard technique for bounding the bias term of momentum. We have

$$\begin{aligned}
&\mathbb{E}[\|\nabla F(\mathbf{W}_t) - \mathbf{C}_t\|_F] \\
=&\mathbb{E}[\|\nabla F(\mathbf{W}_t) - (\beta \mathbf{C}_{t-1} + (1-\beta)\nabla F(\mathbf{W}_t))\|_F] \\
=&\mathbb{E}[\beta\|\nabla F(\mathbf{W}_t) - \mathbf{C}_{t-1}\|_F] \\
\leq&\mathbb{E}[\beta\|\nabla F(\mathbf{W}_{t-1}) - \mathbf{C}_{t-1}\|_F + \beta\|\nabla F(\mathbf{W}_{t-1}) - \nabla F(\mathbf{W}_t)\|_F] \\
\leq&\mathbb{E}[\beta\|\nabla F(\mathbf{W}_{t-1}) - \mathbf{C}_{t-1}\|_F + \beta L\|\mathbf{W}_{t-1} - \mathbf{W}_t\|_F] \\
=&\mathbb{E}[\beta\|\nabla F(\mathbf{W}_{t-1}) - \mathbf{C}_{t-1}\|_F + \beta L \eta\|\mathbf{U}_{t-1}\mathbf{V}_{t-1}^\top\|_F] \\
\leq&\mathbb{E}[\beta\|\nabla F(\mathbf{W}_{t-1}) - \mathbf{C}_{t-1}\|_F + \beta L \eta \sqrt{r}] \\
\leq&\beta^t\|\nabla F(\mathbf{W}_0) - \mathbf{C}_0\|_F + \sum_{i=1}^t \beta^i L \eta \sqrt{r} \\
\leq&\frac{\beta L \eta \sqrt{r}}{1 - \beta}.
\end{aligned}$$

$\qquad\square$

### B.6 PROOF OF LEMMA B.4

**Lemma B.4.** Suppose Assumptions 4.1 and 4.2 hold. For any $t \geq 0$, we have

$$\mathbb{E}[\|\mathbf{C}_t - \mathbf{X}_t\|_F] \leq \beta^t \frac{\sigma}{\sqrt{B}} + \sqrt{\frac{1-\beta}{1+\beta}} \frac{\sigma}{\sqrt{B}}.$$

*Proof of Lemma B.4.* Expanding $\mathbf{C}_t$ and $\mathbf{X}_t$ by their definitions in (B.2) and (B.3), we have

$$\mathbf{C}_t = \beta^t \mathbf{C}_0 + (1-\beta) \sum_{k=1}^t \beta^{t-k} \nabla F(\mathbf{W}_k),$$

$$\mathbf{X}_t = \beta^t \mathbf{X}_0 + (1-\beta) \sum_{k=1}^t \beta^{t-k} \frac{1}{B} \sum_{i=1}^B \nabla f(\mathbf{W}_k; \boldsymbol{\xi}_{k,i}).$$

Thus, we have

$$\mathbb{E}[\|\mathbf{C}_t - \mathbf{X}_t\|_F]$$

$$\leq \mathbb{E}[\|\beta^t(\mathbf{C}_0 - \mathbf{X}_0)\|_F] + \mathbb{E}[(1-\beta)\|\sum_{k=1}^{t} \beta^{t-k}(\nabla F(\mathbf{W}_k) - \frac{1}{B}\sum_{i=1}^{B} \nabla f(\mathbf{W}_k; \boldsymbol{\xi}_{k,i}))\|_F]$$

$$\leq \beta^t \mathbb{E}[\|\mathbf{C}_0 - \mathbf{X}_0\|_F] + \sqrt{\mathbb{E}[(1-\beta)^2 \|\sum_{k=1}^{t} \beta^{t-k}(\nabla F(\mathbf{W}_k) - \frac{1}{B}\sum_{i=1}^{B} \nabla f(\mathbf{W}_k; \boldsymbol{\xi}_{k,i}))\|_F^2]}$$

$$= \beta^t \mathbb{E}[\|\mathbf{C}_0 - \mathbf{X}_0\|_F] + \sqrt{\mathbb{E}[(1-\beta)^2 \sum_{k=1}^{t} \beta^{2(t-k)} \frac{1}{B^2} \sum_{i=1}^{B} \|\nabla F(\mathbf{W}_k; \boldsymbol{\xi}_{k,i}) - \nabla F(\mathbf{W}_k)\|_F^2]}$$

$$\leq \beta^t \mathbb{E}[\|\mathbf{C}_0 - \mathbf{X}_0\|_F] + \sqrt{(1-\beta)^2 \sum_{k=1}^{t} \beta^{2(t-k)} \frac{\sigma^2}{B}}$$

$$\leq \beta^t \frac{\sigma}{\sqrt{B}} + \sqrt{\frac{1-\beta}{1+\beta}} \frac{\sigma}{\sqrt{B}}.$$

The second inequality is due to Jensen's inequality, the first equality is due to the independence of $\boldsymbol{\xi}_{k,i}$ for different $k$ or $i$, and the third inequality is due to Assumptions 4.1 and 4.2. $\qquad\square$

## B.7 PROOF OF LEMMA B.5

**Lemma B.5.** Suppose Assumptions 4.1, 4.2 and 3.1 hold. For any $t \geq 0$, we have

$$\mathbb{E}[\|\mathbf{X}_t - \mathbf{Y}_t\|_F] \leq q_G(\sigma + G + t\eta\sqrt{r}L).$$

*Proof of Lemma B.5.* By the definition of $\mathbf{X}_t$ and $\mathbf{Y}_t$ in (B.3) and (B.4), we have

$$\mathbb{E}[\|\mathbf{X}_t - \mathbf{Y}_t\|_F]$$

$$\leq \mathbb{E}[\beta\|\mathbf{X}_{t-1} - \mathbf{Y}_{t-1}\|_F] + (1-\beta)\frac{1}{B}\sum_{i=1}^{B} \mathbb{E}[\|\nabla f(\mathbf{W}_t; \boldsymbol{\xi}_{t,i}) - \nabla^Q f(\mathbf{W}_t; \boldsymbol{\xi}_{t,i})\|_F]$$

$$\leq \mathbb{E}[\beta\|\mathbf{X}_{t-1} - \mathbf{Y}_{t-1}\|_F] + (1-\beta)\frac{1}{B}\sum_{i=1}^{B} \mathbb{E}[q_G\|\nabla f(\mathbf{W}_t; \boldsymbol{\xi}_{t,i})\|_F]$$

$$\leq \mathbb{E}[\beta\|\mathbf{X}_{t-1} - \mathbf{Y}_{t-1}\|_F] + (1-\beta)\mathbb{E}[q_G(\sigma + \|\nabla F(\mathbf{W}_t)\|_F)]$$

$$\leq \mathbb{E}[\beta\|\mathbf{X}_{t-1} - \mathbf{Y}_{t-1}\|_F] + (1-\beta)q_G(\sigma + G + t\eta\sqrt{r}L)$$

$$\leq \beta^t\|\mathbf{X}_0 - \mathbf{Y}_0\|_F + (1-\beta)q_G(\sigma + G + t\eta\sqrt{r}L)\sum_{k=0}^{t-1} \beta^k$$

$$\leq \beta^t q_G(\sigma + G) + (1-\beta^t)q_G(\sigma + G + t\eta\sqrt{r}L)$$

$$\leq q_G(\sigma + G) + (1-\beta^t)q_G t\eta\sqrt{r}L$$

$$\leq q_G(\sigma + G + t\eta\sqrt{r}L).$$

The second inequality is due to Assumption 3.1, Lemma B.2 and Definition B.6. The third inequality is due to Assumption 4.2. The fourth inequality is due to Lemma B.1. $\qquad\square$

## B.8 PROOF OF LEMMA B.6

**Lemma B.6.** Suppose Assumptions 4.1, 4.2, 4.3 and 3.1 hold. For any $t \geq 0$, we have

$$\mathbb{E}[\|\mathbf{Y}_t - \mathbf{Z}_t\|_F] \leq \beta^t \cdot \frac{2\sigma}{\sqrt{B}} + \sqrt{\frac{1-\beta}{1+\beta}} \cdot \frac{2\sigma}{\sqrt{B}} + 2q_G(\sigma + G) + 2q_G t\eta\sqrt{r}L +$$

$$q_W(1+q_G)DL + q_W(1+q_G)t\eta\sqrt{r}L,$$

$$\mathbb{E}[\|\mathbf{Z}_t\|_F] \le \frac{\sigma}{\sqrt{B}} + \sqrt{\frac{1-\beta}{1+\beta} \cdot \frac{\sigma}{\sqrt{B}}} + G + q_G(\sigma + G) + q_W(1+q_G)DL +$$
$$(1+q_W)(1+q_G)t\eta\sqrt{r}L.$$

*Proof of Lemma B.6.* By the definition of $\mathbf{Y}_t$ and $\mathbf{Z}_t$ in (B.4) and (B.5), we have

$$\mathbf{Y}_t = \beta^t \mathbf{Y}_0 + (1-\beta)\sum_{k=1}^t \beta^{t-k}\frac{1}{B}\sum_{i=1}^B \nabla^Q f(\mathbf{W}_k; \boldsymbol{\xi}_{k,i}),$$

$$\mathbf{Z}_t = \beta^t \mathbf{Z}_0 + (1-\beta)\sum_{k=1}^t \beta^{t-k}\frac{1}{B}\sum_{i=1}^B \nabla^Q f(\mathbf{W}_k^Q; \boldsymbol{\xi}_{k,i}).$$

Thus, by the triangle inequality, we have

$$\mathbb{E}[\|\mathbf{Y}_t - \mathbf{Z}_t\|_F]$$

$$\le \mathbb{E}[\beta^t\|\mathbf{Y}_0 - \mathbf{Z}_0\|_F] + (1-\beta)\mathbb{E}[\|\sum_{k=1}^t \beta^{t-k} \cdot (\frac{1}{B}\sum_{i=1}^B \nabla^Q f(\mathbf{W}_k; \boldsymbol{\xi}_{k,i}) - \nabla^Q f(\mathbf{W}_k^Q; \boldsymbol{\xi}_{k,i}))\|_F]$$

$$\le \mathbb{E}[\beta^t\|\mathbf{Y}_0 - \mathbf{Z}_0\|_F] + \underbrace{(1-\beta)\mathbb{E}[\|\sum_{k=1}^t \beta^{t-k} \cdot (\frac{1}{B}\sum_{i=1}^B \nabla^Q f(\mathbf{W}_k; \boldsymbol{\xi}_{k,i}) - \nabla f(\mathbf{W}_k; \boldsymbol{\xi}_{k,i})\|_F]}_{A} +$$

$$\underbrace{(1-\beta)\mathbb{E}[\|\sum_{k=1}^t \beta^{t-k} \cdot (\frac{1}{B}\sum_{i=1}^B \nabla f(\mathbf{W}_k; \boldsymbol{\xi}_{k,i}) - \nabla F(\mathbf{W}_k))\|_F]}_{C} +$$

$$\underbrace{(1-\beta)\mathbb{E}[\|\sum_{k=1}^t \beta^{t-k} \cdot (\nabla F(\mathbf{W}_k) - \nabla F(\mathbf{W}_k^Q))\|_F]}_{H} +$$

$$\underbrace{(1-\beta)\mathbb{E}[\|\sum_{k=1}^t \beta^{t-k} \cdot (\nabla F(\mathbf{W}_k^Q) - \frac{1}{B}\sum_{i=1}^B \nabla f(\mathbf{W}_k^Q; \boldsymbol{\xi}_{k,i}))\|_F]}_{I} +$$

$$\underbrace{(1-\beta)\mathbb{E}[\|\sum_{k=1}^t \beta^{t-k} \cdot (\frac{1}{B}\sum_{i=1}^B \nabla f(\mathbf{W}_k^Q; \boldsymbol{\xi}_{k,i}) - \nabla^Q f(\mathbf{W}_k^Q; \boldsymbol{\xi}_{k,i}))\|_F]}_{J}. \tag{B.10}$$

Next, we bound each term in (B.10) one by one.

**Bound on $\beta^t \mathbb{E}[\|\mathbf{Y}_0 - \mathbf{Z}_0\|_F]$.** By the definitions of $\mathbf{Y}_0$ and $\mathbf{Z}_0$ in (B.4) and (B.5), we have

$$\mathbf{Y}_0 = \frac{1}{B}\sum_{i=1}^B \nabla^Q f(\mathbf{W}_0; \boldsymbol{\xi}_{0,i}),$$

$$\mathbf{Z}_0 = \frac{1}{B}\sum_{i=1}^B \nabla^Q f(\mathbf{W}_0^Q; \boldsymbol{\xi}_{0,i}).$$

Thus, we have

$$\beta^t \mathbb{E}[\|\mathbf{Y}_0 - \mathbf{Z}_0\|_F]$$

$$
\begin{aligned}
&= \beta^t \mathbb{E}[\| \frac{1}{B} \sum_{i=1}^{B} \nabla^Q f(\mathbf{W}_0; \boldsymbol{\xi}_{0,i}) - \frac{1}{B} \sum_{i=1}^{B} \nabla^Q f(\mathbf{W}_0^Q; \boldsymbol{\xi}_{0,i}) \|_F] \\
&\leq \beta^t \frac{1}{B} \sum_{i=1}^{B} \mathbb{E}[\| \nabla^Q f(\mathbf{W}_0; \boldsymbol{\xi}_{0,i}) - \nabla f(\mathbf{W}_0; \boldsymbol{\xi}_{0,i}) \|_F] + \\
&\quad \beta^t \mathbb{E}[\| \frac{1}{B} \sum_{i=1}^{B} \nabla f(\mathbf{W}_0; \boldsymbol{\xi}_{0,i}) - \nabla F(\mathbf{W}_0) \|_F] + \\
&\quad \beta^t \mathbb{E}[\| \nabla F(\mathbf{W}_0) - \nabla F(\mathbf{W}_0^Q) \|_F] + \\
&\quad \beta^t \mathbb{E}[\| \frac{1}{B} \sum_{i=1}^{B} \nabla F(\mathbf{W}_0^Q) - \nabla f(\mathbf{W}_0^Q; \boldsymbol{\xi}_{0,i}) \|_F] + \\
&\quad \beta^t \frac{1}{B} \sum_{i=1}^{B} \mathbb{E}[\| \nabla f(\mathbf{W}_0^Q; \boldsymbol{\xi}_{0,i}) - \nabla^Q f(\mathbf{W}_0^Q; \boldsymbol{\xi}_{0,i}) \|_F] \\
&\leq \beta^t \frac{1}{B} \sum_{i=1}^{B} q_G \mathbb{E}[\| \nabla f(\mathbf{W}_0; \boldsymbol{\xi}_{0,i}) \|_F] + \beta^t \frac{\sigma}{\sqrt{B}} + \beta^t L q_W \mathbb{E}[\| \mathbf{W}_0 \|_F] + \beta^t \frac{\sigma}{\sqrt{B}} + \\
&\quad \beta^t \frac{1}{B} \sum_{i=1}^{B} q_G \mathbb{E}[\| \nabla f(\mathbf{W}_0^Q; \boldsymbol{\xi}_{0,i}) \|_F] \\
&\leq \beta^t \frac{1}{B} \sum_{i=1}^{B} q_G(\sigma + G) + \beta^t \frac{2\sigma}{\sqrt{B}} + \beta^t q_W D L + \beta^t \frac{1}{B} \sum_{i=1}^{B} q_G(\sigma + q_W D L + G) \\
&= \beta^t (2 q_G(\sigma + G) + q_W D L (1 + q_G) + \frac{2\sigma}{\sqrt{B}}).
\end{aligned} \tag{B.11}
$$

The first inequality is due to the triangle inequality. The second inequality we used Definition B.6 for the first and last terms, Assumption 4.1, 4.2 and Jensen's inequality for the second and fourth terms, and Assumption 4.3 and Definition B.6 for the third term. The third inequality is due to Assumption 4.2, 4.4 and Definition B.6.

**Bound on A.**

$$
\begin{aligned}
A &= (1-\beta) \mathbb{E}[\| \sum_{k=1}^{t} \beta^{t-k} \cdot (\frac{1}{B} \sum_{i=1}^{B} \nabla^Q f(\mathbf{W}_k; \boldsymbol{\xi}_{k,i}) - \nabla f(\mathbf{W}_k; \boldsymbol{\xi}_{k,i}) \|_F] \\
&\leq (1-\beta) \sum_{k=1}^{t} \beta^{t-k} \frac{1}{B} \sum_{i=1}^{B} q_G \mathbb{E}[\| \nabla f(\mathbf{W}_k; \boldsymbol{\xi}_{k,i}) \|_F] \\
&\leq (1-\beta) \sum_{k=1}^{t} \beta^{t-k} \frac{1}{B} \sum_{i=1}^{B} q_G(\sigma + \mathbb{E}[\| \nabla F(\mathbf{W}_k) \|_F]) \\
&\leq (1-\beta) \sum_{k=1}^{t} \beta^{t-k} \frac{1}{B} \sum_{i=1}^{B} q_G(\sigma + G + t\eta\sqrt{r}L) \\
&= (1-\beta^t) q_G(\sigma + G + t\eta\sqrt{r}L).
\end{aligned} \tag{B.12}
$$

The first inequality is due to Definition B.6 and the triangle inequality. The second inequality is due to Assumption 4.2. The third inequality is due to Lemma B.1.

**Bound on C.** Similar to Lemma B.4, we have

$$
C = (1-\beta) \mathbb{E}[\| \sum_{k=1}^{t} \beta^{t-k} \cdot (\frac{1}{B} \sum_{i=1}^{B} \nabla f(\mathbf{W}_k; \boldsymbol{\xi}_{k,i}) - \nabla F(\mathbf{W}_k)) \|_F]
$$

$$\leq (1-\beta)\sqrt{\mathbb{E}[\|\sum_{k=1}^{t}\beta^{t-k}\cdot(\frac{1}{B}\sum_{i=1}^{B}\nabla f(\mathbf{W}_k;\boldsymbol{\xi}_{k,i})-\nabla F(\mathbf{W}_k))\|_F^2]}$$

$$= (1-\beta)\sqrt{\sum_{k=1}^{t}\beta^{2(t-k)}\frac{1}{B^2}\sum_{i=1}^{B}\mathbb{E}[\|\nabla f(\mathbf{W}_k;\boldsymbol{\xi}_{k,i})-\nabla F(\mathbf{W}_k)\|_F^2]}$$

$$\leq (1-\beta)\sqrt{\sum_{k=1}^{t}\beta^{2(t-k)}\frac{1}{B^2}\sum_{i=1}^{B}\sigma^2}$$

$$= (1-\beta)\sqrt{\frac{1-\beta^{2t}}{1-\beta^2}\cdot\frac{\sigma^2}{B}}$$

$$\leq \sqrt{\frac{1-\beta}{1+\beta}\cdot\frac{\sigma}{\sqrt{B}}}. \tag{B.13}$$

**Bound on H.**

$$H = (1-\beta)\mathbb{E}[\|\sum_{k=1}^{t}\beta^{t-k}\cdot(\nabla F(\mathbf{W}_k)-\nabla F(\mathbf{W}_k^Q))\|_F]$$

$$\leq (1-\beta)\sum_{k=1}^{t}\beta^{t-k}\mathbb{E}[\|\nabla F(\mathbf{W}_k)-\nabla F(\mathbf{W}_k^Q)\|_F]$$

$$\leq (1-\beta)\sum_{k=1}^{t}\beta^{t-k}L\mathbb{E}[\|\mathbf{W}_k-\mathbf{W}_k^Q\|_F]$$

$$\leq (1-\beta)\sum_{k=1}^{t}\beta^{t-k}Lq_W\mathbb{E}[\|\mathbf{W}_k\|_F]$$

$$\leq (1-\beta)\sum_{k=1}^{t}\beta^{t-k}Lq_W(D+t\eta\sqrt{r})$$

$$\leq (1-\beta^t)q_W L(D+t\eta\sqrt{r}). \tag{B.14}$$

The first inequality is due to the triangle inequality. The second inequality is due to Assumption 4.3. The third inequality is due to Definition B.6. The fourth inequality is due to Lemma B.1.

**Bound on I.**  Similar to Lemma B.4, we have

$$I = (1-\beta)\mathbb{E}[\|\sum_{k=1}^{t}\beta^{t-k}\cdot(\nabla F(\mathbf{W}_k^Q)-\frac{1}{B}\sum_{i=1}^{B}\nabla f(\mathbf{W}_k^Q;\boldsymbol{\xi}_{k,i}))\|_F]$$

$$\leq (1-\beta)\sqrt{\mathbb{E}[\|\sum_{k=1}^{t}\beta^{t-k}\cdot(\nabla F(\mathbf{W}_k^Q)-\frac{1}{B}\sum_{i=1}^{B}\nabla f(\mathbf{W}_k^Q;\boldsymbol{\xi}_{k,i}))\|_F^2]}$$

$$= (1-\beta)\sqrt{\sum_{k=1}^{t}\beta^{2(t-k)}\frac{1}{B^2}\sum_{i=1}^{B}\mathbb{E}[\|\nabla f(\mathbf{W}_k^Q;\boldsymbol{\xi}_{k,i})-\nabla F(\mathbf{W}_k^Q)\|_F^2]}$$

$$\leq (1-\beta)\sqrt{\sum_{k=1}^{t}\beta^{2(t-k)}\frac{1}{B^2}\sum_{i=1}^{B}\sigma^2}$$

$$= (1-\beta)\sqrt{\frac{1-\beta^{2t}}{1-\beta^2}\cdot\frac{\sigma^2}{B}}$$

$$\le \sqrt{\frac{1-\beta}{1+\beta}} \cdot \frac{\sigma}{\sqrt{B}}. \tag{B.15}$$

**Bound on $\mathbf{J}$.**

$$\begin{aligned}
J =&(1-\beta)\mathbb{E}[\| \sum_{k=1}^{t} \beta^{t-k} \cdot (\frac{1}{B}\sum_{i=1}^{B} \nabla f(\mathbf{W}_k^Q;\boldsymbol{\xi}_{k,i}) - \nabla^Q f(\mathbf{W}_k^Q;\boldsymbol{\xi}_{k,i}))\|_F] \\
\le&(1-\beta)\sum_{k=1}^{t} \beta^{t-k}\frac{1}{B}\sum_{i=1}^{B} q_G \mathbb{E}[\|\nabla f(\mathbf{W}_k^Q;\boldsymbol{\xi}_{k,i})\|_F] \\
\le&(1-\beta)\sum_{k=1}^{t} \beta^{t-k}\frac{1}{B}\sum_{i=1}^{B} q_G(\sigma + \mathbb{E}[\|\nabla F(\mathbf{W}_k^Q)\|_F]) \\
\le&(1-\beta)\sum_{k=1}^{t} \beta^{t-k}\frac{1}{B}\sum_{i=1}^{B} q_G(\sigma + L\mathbb{E}[\|\mathbf{W}_k^Q - \mathbf{W}_k\|_F] + \mathbb{E}[\|\nabla F(\mathbf{W}_k)\|_F]) \\
\le&(1-\beta)\sum_{k=1}^{t} \beta^{t-k}\frac{1}{B}\sum_{i=1}^{B} q_G(\sigma + Lq_W \mathbb{E}[\|\mathbf{W}_k\|_F] + \mathbb{E}[\|\nabla F(\mathbf{W}_k)\|_F]) \\
\le&(1-\beta)\sum_{k=1}^{t} \beta^{t-k}\frac{1}{B}\sum_{i=1}^{B} q_G(\sigma + Lq_W(D + t\eta\sqrt{r}) + G + t\eta\sqrt{r}L) \\
\le&(1-\beta^t)q_G(\sigma + G + q_W DL + (1+q_W)t\eta\sqrt{r}L). \tag{B.16}
\end{aligned}$$

The first inequality is due to Definition B.6 and the triangle inequality. The second inequality is due to Assumption 4.2. The third inequality is due to Assumption 4.3 and the triangle inequality. The fourth inequality is due to Definition B.6. The fifth inequality is due to Lemma B.1.

**Bound on $\mathbb{E}[\|\mathbf{Y}_t - \mathbf{Z}_t\|_F]$.** Substituting (B.11), (B.12), (B.13), (B.14), (B.15) and (B.16) into (B.10), we have

$$\begin{aligned}
\mathbb{E}[\|\mathbf{Y}_t - \mathbf{Z}_t\|_F] \le& \beta^t \cdot \frac{2\sigma}{\sqrt{B}} + \sqrt{\frac{1-\beta}{1+\beta}} \cdot \frac{2\sigma}{\sqrt{B}} + 2q_G(\sigma + G) + (1-\beta^t)2q_G t\eta\sqrt{r}L + \\
& q_W(1+q_G)DL + (1-\beta^t)q_W(1+q_G)t\eta\sqrt{r}L. \\
\le& \beta^t \cdot \frac{2\sigma}{\sqrt{B}} + \sqrt{\frac{1-\beta}{1+\beta}} \cdot \frac{2\sigma}{\sqrt{B}} + 2q_G(\sigma + G) + 2q_G t\eta\sqrt{r}L + \\
& q_W(1+q_G)DL + q_W(1+q_G)t\eta\sqrt{r}L.
\end{aligned}$$

**Bound on $\mathbb{E}[\|\mathbf{Z}_t\|_F]$.** By the definition of $\mathbf{Z}_t$ in (B.5), we have

$$\begin{aligned}
&\mathbb{E}[\|\mathbf{Z}_t\|_F] \\
\le& \mathbb{E}[\beta^t\|\mathbf{Z}_0\|_F] + \mathbb{E}[(1-\beta)\| \sum_{k=1}^{t} \beta^{t-k} \cdot \frac{1}{B}\sum_{i=1}^{B} \nabla^Q f(\mathbf{W}_k^Q;\boldsymbol{\xi}_{k,i})\|_F] \\
\le& \beta^t\mathbb{E}[\|\mathbf{Z}_0\|_F] + \mathbb{E}[(1-\beta)\sum_{k=1}^{t} \beta^{t-k} \cdot \frac{1}{B}\sum_{i=1}^{B} \|\nabla^Q f(\mathbf{W}_k^Q;\boldsymbol{\xi}_{k,i}) - \nabla f(\mathbf{W}_k^Q;\boldsymbol{\xi}_{k,i})\|_F] + \\
& \mathbb{E}[(1-\beta)\| \sum_{k=1}^{t} \beta^{t-k} \cdot (\frac{1}{B}\sum_{i=1}^{B} \nabla f(\mathbf{W}_k^Q;\boldsymbol{\xi}_{k,i}) - \nabla F(\mathbf{W}_k^Q))\|_F] + \\
& \mathbb{E}[(1-\beta)\sum_{k=1}^{t} \beta^{t-k} \cdot (\|\nabla F(\mathbf{W}_k^Q) - \nabla F(\mathbf{W}_k)\|_F + \|\nabla F(\mathbf{W}_k)\|_F)]
\end{aligned}$$

$$\leq \beta^t \mathbb{E}[\|\mathbf{Z}_0\|_F] + (1-\beta) \sum_{k=1}^{t} \beta^{t-k} \frac{1}{B} \sum_{i=1}^{B} q_G \mathbb{E}[\|\nabla f(\mathbf{W}_k^Q; \boldsymbol{\xi}_{k,i})\|_F] +$$

$$\sqrt{\frac{1-\beta}{1+\beta}} \cdot \frac{\sigma}{\sqrt{B}} + (1-\beta) q_W L \sum_{k=1}^{t} \beta^{t-k} \mathbb{E}[\|\mathbf{W}_k\|_F] + (1-\beta) \sum_{k=1}^{t} \beta^{t-k} \mathbb{E}[\|\nabla F(\mathbf{W}_k)\|_F]$$

$$\leq \beta^t \mathbb{E}[\|\mathbf{Z}_0\|_F] + (1-\beta) \sum_{k=1}^{t} \beta^{t-k} \frac{1}{B} \sum_{i=1}^{B} q_G(\sigma + q_W L \mathbb{E}[\|\mathbf{W}_k\|_F] + \mathbb{E}[\|\nabla F(\mathbf{W}_k)\|_F]) +$$

$$\sqrt{\frac{1-\beta}{1+\beta}} \cdot \frac{\sigma}{\sqrt{B}} + (1-\beta) q_W L \sum_{k=1}^{t} \beta^{t-k} \mathbb{E}[\|\mathbf{W}_k\|_F] + (1-\beta) \sum_{k=1}^{t} \beta^{t-k} \mathbb{E}[\|\nabla F(\mathbf{W}_k)\|_F]$$

$$\leq \beta^t \mathbb{E}[\|\mathbf{Z}_0\|_F] + (1-\beta^t) q_G(\sigma + q_W L(D + t\eta\sqrt{r}) + G + t\eta\sqrt{r}L) +$$

$$\sqrt{\frac{1-\beta}{1+\beta}} \cdot \frac{\sigma}{\sqrt{B}} + (1-\beta^t) q_W L(D + t\eta\sqrt{r}) + (1-\beta^t)(G + t\eta\sqrt{r}L)$$

$$\leq \beta^t(q_G(\sigma + q_W DL + G) + \frac{\sigma}{\sqrt{B}} + q_W DL + G +$$

$$(1-\beta^t) q_G(\sigma + q_W L(D + t\eta\sqrt{r}) + G + t\eta\sqrt{r}L) +$$

$$\sqrt{\frac{1-\beta}{1+\beta}} \cdot \frac{\sigma}{\sqrt{B}} + (1-\beta^t) q_W L(D + t\eta\sqrt{r}) + (1-\beta^t)(G + t\eta\sqrt{r}L)$$

$$\leq \beta^t \frac{\sigma}{\sqrt{B}} + \sqrt{\frac{1-\beta}{1+\beta}} \cdot \frac{\sigma}{\sqrt{B}} + G + q_G(\sigma + G) + q_W(1+q_G)DL +$$

$$(1-\beta^t)(1+q_W)(1+q_G)t\eta\sqrt{r}L$$

$$\leq \frac{\sigma}{\sqrt{B}} + \sqrt{\frac{1-\beta}{1+\beta}} \cdot \frac{\sigma}{\sqrt{B}} + G + q_G(\sigma + G) + q_W(1+q_G)DL + (1+q_W)(1+q_G)t\eta\sqrt{r}L.$$

The first and second inequalities are due to the triangle inequality. The third inequality we used Definition B.6 for the second term, Jensen's inequality, Assumptions 4.1, 4.2 for the third term, Assumption 4.3 and Definition B.6 for the fourth term. The fourth inequality we used triangle inequality, Assumptions 4.2, 4.3, Definition B.6. The fifth inequality is due to Lemma B.1.

$\square$

## B.9 PROOF OF LEMMA B.7

**Lemma B.7.** Suppose Assumptions 4.1, 4.2, 4.3 and 3.1 hold. For any $t \geq 0$, if $\beta(1 + q_M) < 1$, we have

$$\mathbb{E}[\|\mathbf{Z}_t - \mathbf{M}_t\|_F] \leq \frac{q_M \beta}{1 - \beta(1 + q_M)} \left( \frac{\sigma}{\sqrt{B}} + \sqrt{\frac{1-\beta}{1+\beta}} \cdot \frac{\sigma}{\sqrt{B}} + G + q_G(\sigma + G) + \right.$$

$$\left. q_W(1+q_G)DL + (1+q_W)(1+q_G)t\eta\sqrt{r}L \right).$$

*Proof of Lemma B.7.* By the definitions of $\mathbf{Z}_t$ and $\mathbf{M}_t$ in (B.5) and (B.1), we have

$$\mathbb{E}[\|\mathbf{Z}_t - \mathbf{M}_t\|_F]$$
$$\leq \mathbb{E}[\beta\|\mathbf{Z}_{t-1} - \mathbf{M}_{t-1}^Q\|_F]$$
$$\leq \mathbb{E}[\beta\|\mathbf{Z}_{t-1} - \mathbf{M}_{t-1}\|_F + \beta\|\mathbf{M}_{t-1} - \mathbf{M}_{t-1}^Q\|_F]$$
$$\leq \mathbb{E}[\beta\|\mathbf{Z}_{t-1} - \mathbf{M}_{t-1}\|_F + q_M\beta\|\mathbf{M}_{t-1}\|_F]$$
$$\leq \mathbb{E}[\beta(1+q_M)\|\mathbf{Z}_{t-1} - \mathbf{M}_{t-1}\|_F + q_M\beta\|\mathbf{Z}_{t-1}\|_F]$$

$$\leq q_M \beta \sum_{k=0}^{t-1} \beta^k (1 + q_M)^k \|\mathbf{Z}_{t-k-1}\|_F$$

$$\leq \frac{q_M \beta}{1 - \beta(1 + q_M)} \left( \frac{\sigma}{\sqrt{B}} + \sqrt{\frac{1-\beta}{(1+\beta)}} \cdot \frac{\sigma}{\sqrt{B}} + G + q_G(\sigma + G) + \right.$$

$$\left. q_W(1 + q_G)DL + (1 + q_W)(1 + q_G)t\eta\sqrt{r}L \right). \tag{B.17}$$

The second inequality is due to the triangle inequality. The third inequality is due to Definition B.6. The fourth inequality is due to the triangle inequality. The fifth inequality is due to $\mathbf{Z}_0 = \mathbf{M}_0$. The last inequality we used Lemma B.6 □

## C ADDITIONAL EXPERIMENTS AND DETAILS

### C.1 IMITATING QUANTIZATION AND DEQUANTIZATION

We emulate floating-point quantization and dequantization by reducing the mantissa length from its original precision (52 bits for `float64` and 23 bits for `float32`) to $M$ bits, while keeping the exponent and sign bits unchanged. This design choice is motivated by the fact that practical scaling techniques can effectively prevent overflow and underflow (Peng et al., 2023). After truncating the mantissa, we apply stochastic rounding to the nearest two representable values, and then dequantize the result back to standard `float32` or `float64`.

### C.2 SYNTHETIC EXPERIMENTS

We conduct synthetic experiments on the Rosenbrock function, defined as

$$F(\mathbf{W}) = \sum_{j=1}^{n-1} \left( 100\|\mathbf{W}_{j+1} - \mathbf{W}_j^2\|_F^2 + \|\mathbf{1}_m - \mathbf{W}_j\|_F^2 \right),$$

where $\mathbf{W} = [\mathbf{W}_1, \mathbf{W}_2, \ldots, \mathbf{W}_d] \in \mathbb{R}^{m \times n}$ is the weight matrix. The global minimum is at $\mathbf{W}^* = [\mathbf{1}_m, \mathbf{1}_m, \ldots, \mathbf{1}_m]$ with $F(\mathbf{W}^*) = 0$. We set $m = 50$, $d = 100$, and initialize $\mathbf{W}_0 \sim \mathcal{N}(\mathbf{1}_{m \times n}, 0.1^2\mathbf{I})$. For Muon, we apply the default hyperparameters in the Newton-Schulz iteration to compute the zeroth power / orthogonalization of $G$ (Jordan et al., 2024), using double precision.

Figure 3 shows the gradient norms of Adam with different quantization errors on the Rosenbrock function. Figure 4 shows the gradient norms of Muon with different quantization errors on the Rosenbrock function.

Figure 5 shows the function values of Adam with different quantization errors on the Rosenbrock function. Figure 6 shows the function values of Muon with different quantization errors on the Rosenbrock function. The relative quantization error is defined as $\frac{\|\mathbf{X} - Q(\mathbf{X})\|_F}{\|\mathbf{X}\|_F}$, measuring the average quantization error of $\mathbf{X}$, where $Q(\cdot)$ is the quantization operator.

Figure 7 shows the effect of quantizing the second moment in Adam to different mantissa lengths $M$, with all other components kept in FP32. As $\beta_2 \to 1$, the optimizer exhibits larger converged gradient norms and becomes more sensitive to quantization errors induced by reduced $M$. This phenomenon aligns with our theoretical analysis in Theorem 4.5, which highlights the amplification of quantization errors by the inverse square root of historical gradient variances in Adam when $\beta_2$ is close to 1.

### C.3 CIFAR-10 EXPERIMENTS

We conduct real-data experiments on the CIFAR-10 dataset (Krizhevsky et al., 2009) using a 4-layer fully connected network (FCN). The architecture is as follows: an input layer with 3072 neurons (corresponding to $32 \times 32 \times 3$ images), followed by three hidden layers with 512, 256, and 64 neurons, respectively, and an output layer with 10 neurons for classification. ReLU activations are

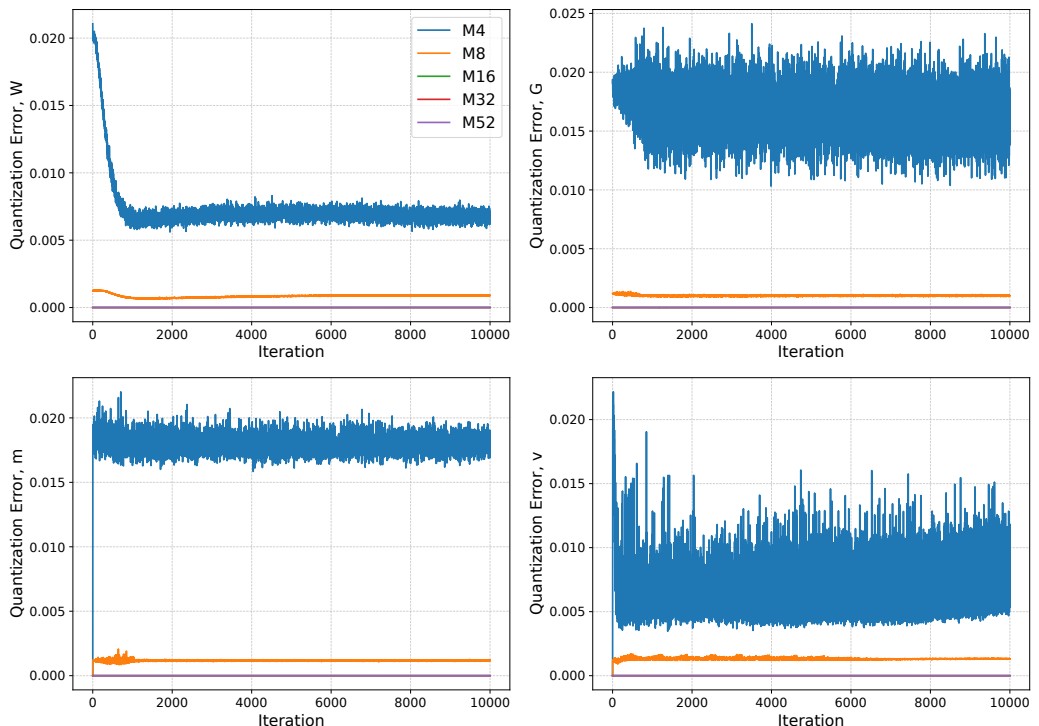

Figure 5: Rosenbrock: Adam relative quantization error of different mantissa bits ($M$). Weights error (top left), Gradient error (top right), First moment error (bottom left), Second moment error (bottom right). These results show that the more mantissa bits, the smaller the relative quantization error. Combining with Figure 3, we can see that the more mantissa bits, the smaller quantization error, the better convergence performance (Theorem 4.5).

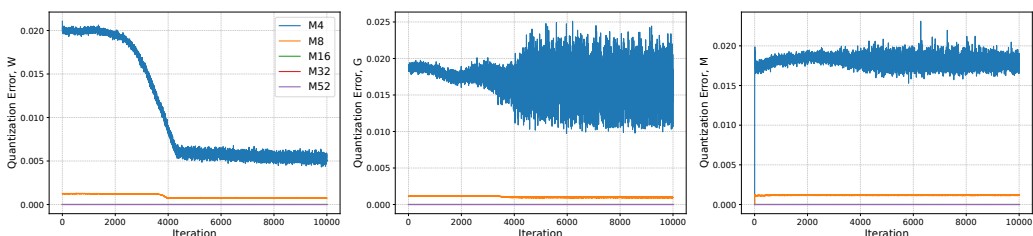

Figure 6: Rosenbrock: Muon relative quantization error of different mantissa bits ($M$). Weights error (left), Gradient error (middle), Momentum error (right). These results show that the more mantissa bits, the smaller the relative quantization error. Combining with Figure 4, we can see that the more mantissa bits, the smaller quantization error, the better convergence performance (Theorem 4.6).

used for all hidden layers, and the network is trained with the cross-entropy loss for 100 epochs. We evaluate both Adam and Muon under varying quantization precisions.

For Adam, we use mantissa bit-lengths $M \in \{1, 2, 3, 7, 10, 23\}$, batch size $B = 256$, learning rate $\eta = 1.5 \times 10^{-4}$, $\beta_1 = 0.95$, $\beta_2 = 0.999$, $\epsilon = 10^{-8}$, and weight decay $0.1$. For Muon, vector parameters are updated using Adam, while matrix parameters are updated with Muon's orthogonalization step. We choose mantissa bit-lengths $M \in \{2, 3, 7, 10, 23\}$, batch size $B = 512$, learning rate $\eta = 0.001$, $\beta = 0.99$, weight decay $0.1$, and 5 Newton–Schulz iterations, following the iteration hyperparameters in Jordan et al. (2024). The auxiliary Adam optimizer in Muon uses learning rate $\eta = 2 \times 10^{-4}$, $\beta_1 = 0.9$, $\beta_2 = 0.999$, $\epsilon = 10^{-8}$, and weight decay $0.05$.

Figure 8 shows the gradient norms of Adam with different quantization errors on CIFAR-10. Figure 9 shows the gradient norms of Muon with different quantization errors on CIFAR-10. Figure 10

shows the quantization errors of Adam with different precision on CIFAR-10. Figure 11 shows the quantization errors of Muon with different precision on CIFAR-10. The relative quantization error is defined as $\frac{\|\mathbf{X}-Q(\mathbf{X})\|_F}{\|\mathbf{X}\|_F}$, measuring the average quantization error of $\mathbf{X}$, where $Q(\cdot)$ is the quantization operator.

### C.4 NANOGPT EXPERIMENTS

We evaluate the impact of quantization on training the `nanoGPT` model on the OpenWebText dataset (Gokaslan et al., 2019). The model has $\sim 26.4M$ parameters, with weight tying between the embedding and the output layer (`lm_head`). Its architecture includes 4 transformer layers, each with 4 attention heads, embedding dimension 384, without dropout, and no bias terms. The dataset contains $\sim 655.4M$ tokens, and we use a block size (context length) of 512. Training is performed with a batch size of 32 and gradient accumulation of 4, resulting in an effective batch size of 128. Models are trained for up to 10,000 iterations.

**Optimizer and Training Settings.** We experiment with both AdamW and Muon optimizers, as summarized below:

- **AdamW:** learning rate $3 \times 10^{-4}$, weight decay 0.1, $\beta_1 = 0.9$, $\beta_2 = 0.95$, $\epsilon = 10^{-8}$, gradient clipping norm 1.0. Learning rate decay is disabled.
- **Muon:** 2D parameters in transformer blocks ($\sim 7M$) are updated with Muon's orthogonalization-based step (Newton-Schulz iteration), while all remaining parameters ($\sim 19M$, including embeddings, layer norms, and output layer) are updated with AdamW. Muon hyperparameters are: learning rate $3 \times 10^{-2}$, $\beta = 0.95$ (with Nesterov momentum), Newton-Schulz steps 5, $\epsilon = 1 \times 10^{-7}$ for NS iteration. Auxiliary AdamW: learning rate $6 \times 10^{-3}$, $\beta_1 = 0.9$, $\beta_2 = 0.95$, $\epsilon = 10^{-8}$, weight decay 0.01, gradient clipping norm 1.0.

**Quantization.** Following the procedure in Section C, we apply mantissa truncation to weights, gradients, and optimizer states. We vary the mantissa length $M \in \{1, 2, 10, 23\}$, keeping exponent and sign bits in full precision.

**Results.** Figures 12 and 13 show training and validation loss dynamics for nanoGPT under different quantization precisions. We observe that:

- Lower mantissa lengths (e.g., $M = 2$) induce slightly slower convergence and higher final training loss, consistent with the observed gradient norm amplification in Theorem 4.5 and Theorem 4.6.
- Muon exhibits greater robustness to low-precision quantization compared to AdamW, achieving lower training and validation loss at $M = 2$. This aligns with our theoretical findings that Muon's quantization error amplification is less sensitive than Adam's.
- As the mantissa length increases, both AdamW and Muon converge to almost identical training and validation loss, indicating that higher precision mitigates quantization-induced degradation.

Overall, these results on nanoGPT extend the findings from synthetic (Rosenbrock) and CIFAR-10 experiments to a real large-scale language modeling setting, highlighting the interplay between quantization precision, optimizer dynamics, and convergence stability.

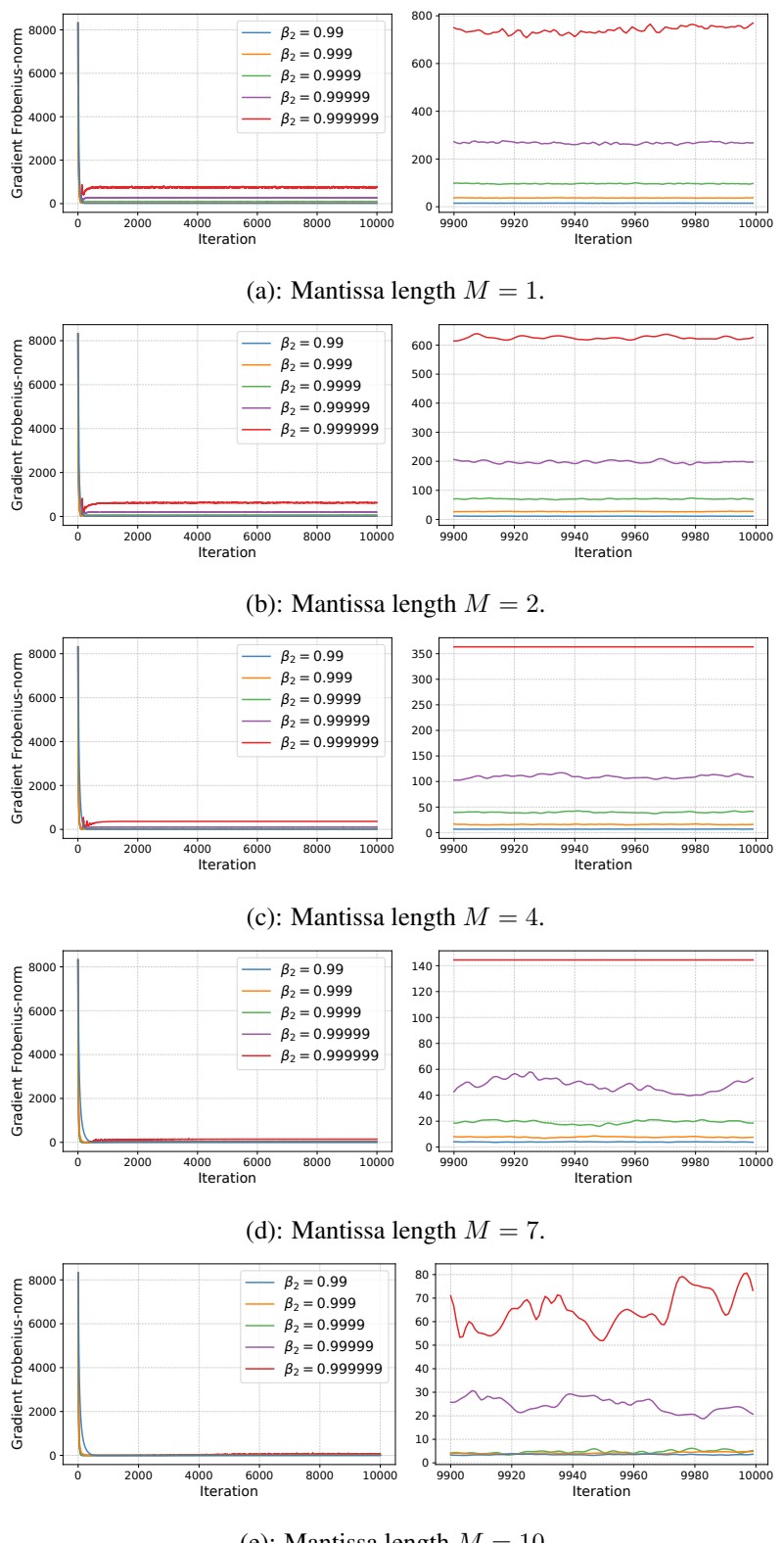

(a): Mantissa length $M = 1$.

(b): Mantissa length $M = 2$.

(c): Mantissa length $M = 4$.

(d): Mantissa length $M = 7$.

(e): Mantissa length $M = 10$.

Figure 7: Rosenbrock: Effect of quantizing second moment in Adam to different mantissa lengths $M$, with all other components kept in FP32. As $\beta_2 \to 1$, the optimizer exhibits larger converged gradient norms and becomes more sensitive to quantization errors induced by reduced $M$.

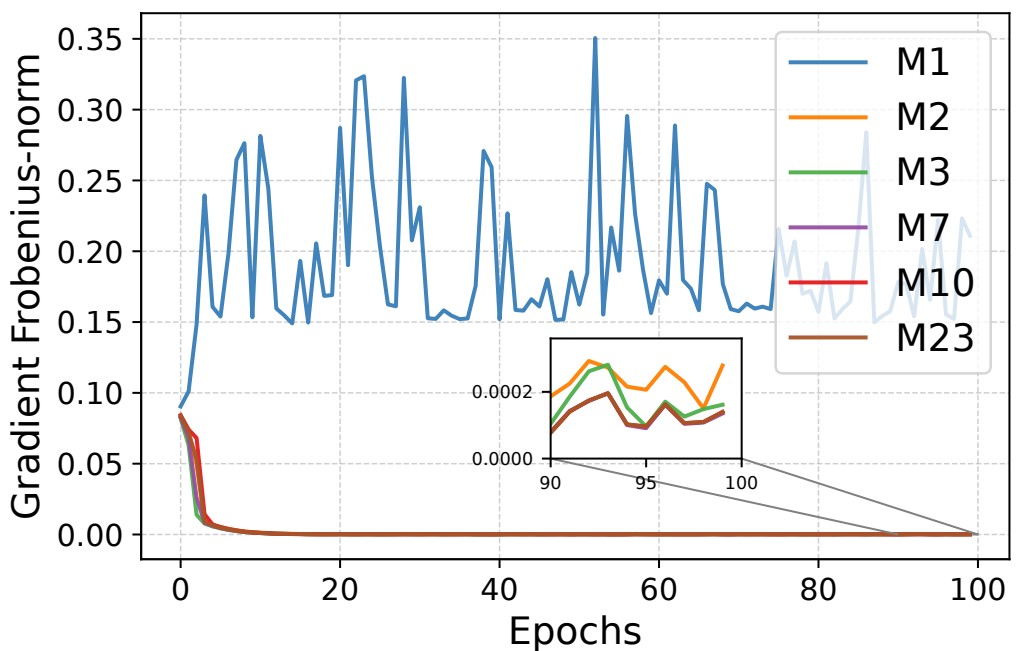

Figure 8: CIFAR-10: Adam gradient norms under different mantissa precisions $M$. Larger mantissa bit-lengths lead to smaller converged gradient norms. Together with Figure 10, this demonstrates that higher precision reduces quantization error and improves convergence, consistent with Theorem 4.5.

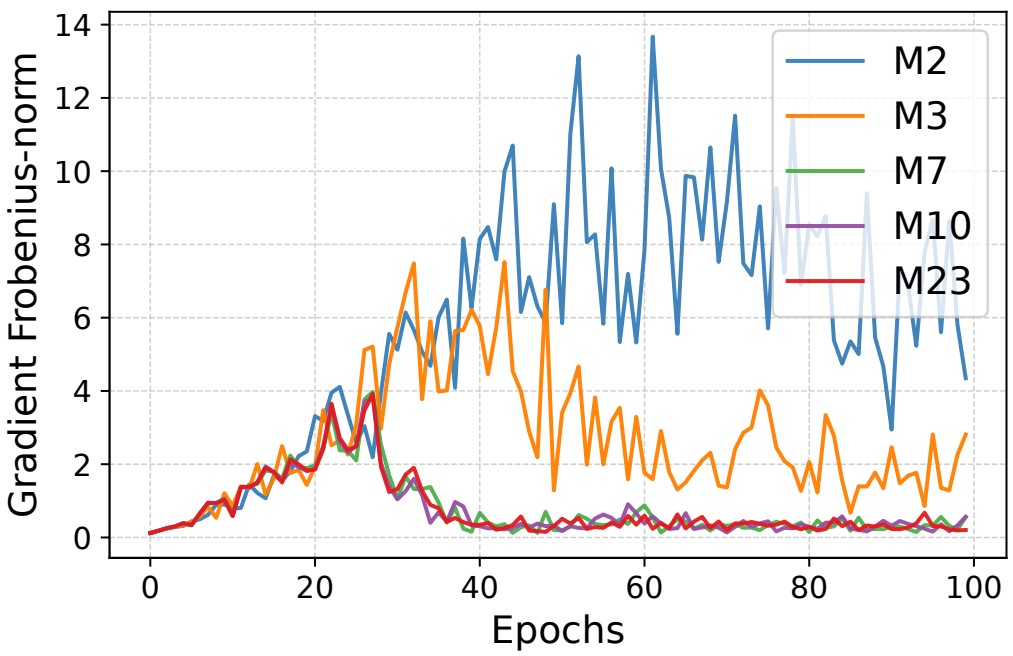

Figure 9: CIFAR-10: Muon gradient norms under different mantissa precisions $M$. Larger mantissa bit-lengths lead to smaller converged gradient norms. Together with Figure 11, this demonstrates that higher precision reduces quantization error and improves convergence, consistent with Theorem 4.6.

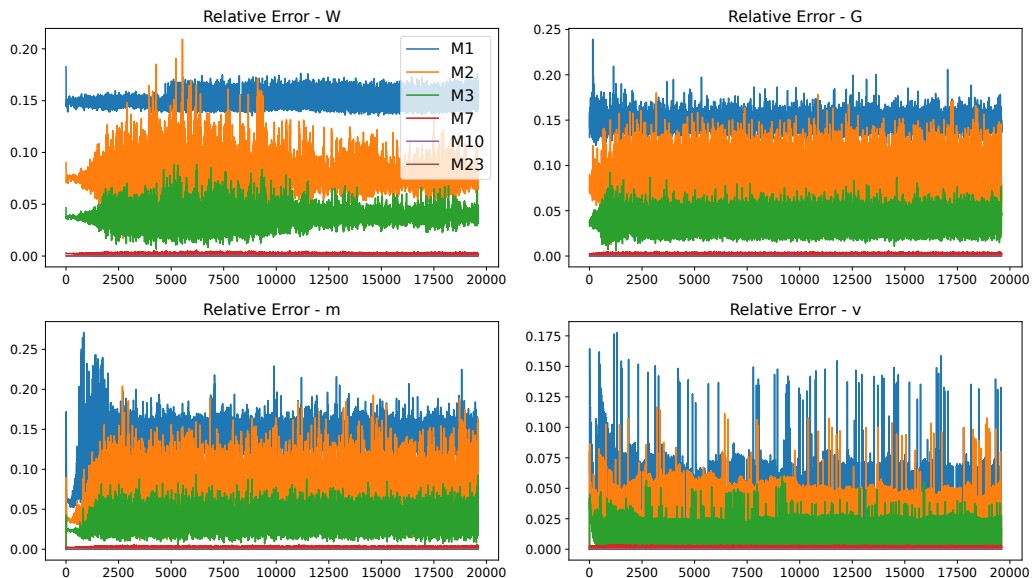

Figure 10: CIFAR-10: Adam relative quantization error of different mantissa bits ($M$). Weights error (top left), Gradient error (top right), First moment error (bottom left), Second moment error (bottom right). These results show that the more mantissa bits, the smaller the relative quantization error. Combining with Figure 8, we can see that the more mantissa bits, the smaller quantization error, the better convergence performance (Theorem 4.5).

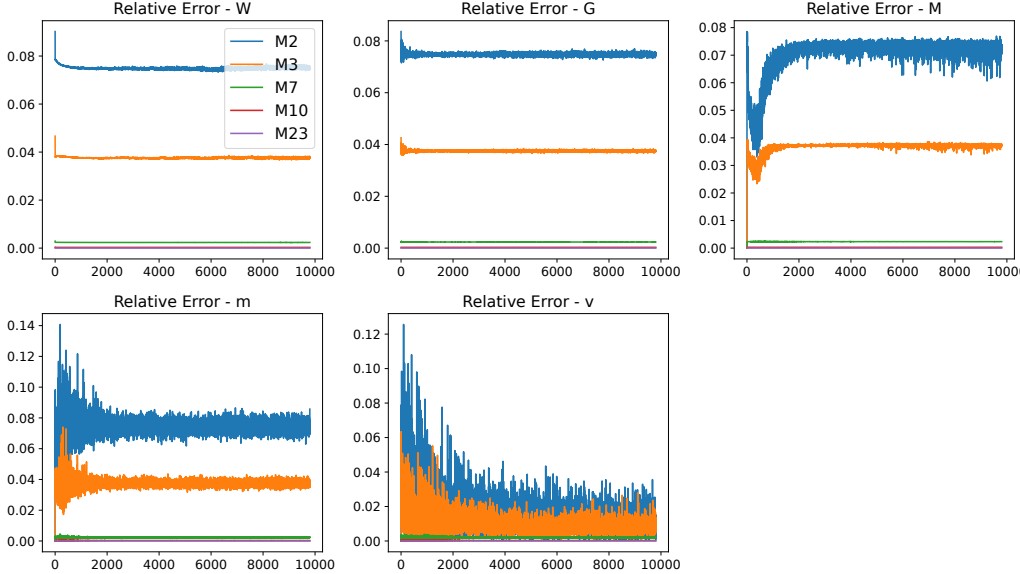

Figure 11: CIFAR-10: Muon with auxiliary Adam relative quantization error of different mantissa bits ($M$). Weights error (top left), Gradient error (top middle), Momentum error (top right), Auxiliary Adam first moment error (bottom left), Auxiliary Adam second moment error (bottom middle). These results show that the more mantissa bits, the smaller the relative quantization error. Combining with Figure 9, we can see that the more mantissa bits, the smaller quantization error, the better convergence performance (Theorem 4.6).

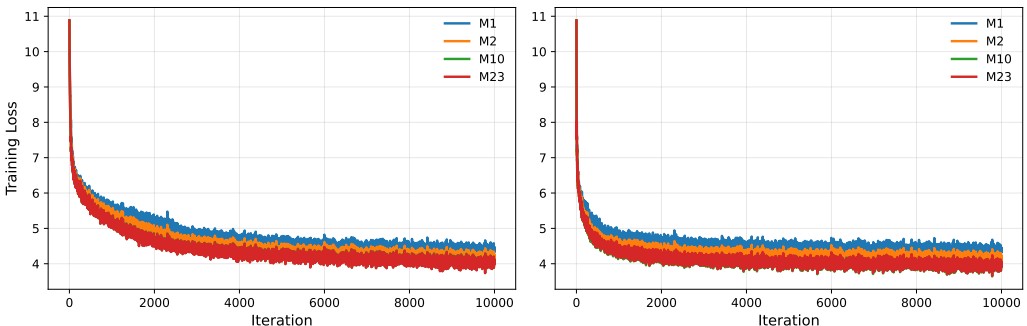

Figure 12: Training loss of nanoGPT on OpenWebText with varying mantissa lengths $M$. Lower $M$ slightly increases the training loss due to amplified quantization error.

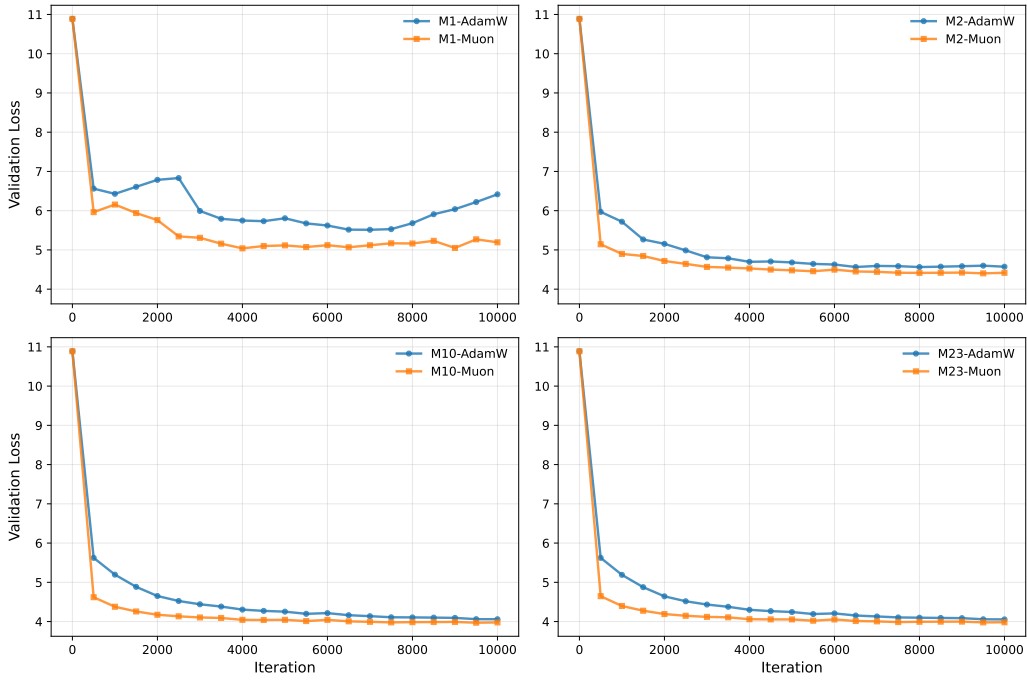

Figure 13: Validation loss of nanoGPT on OpenWebText under varying mantissa precisions $M$. Higher precision reduces quantization error and improves validation performance, particularly at low $M$. Notably, Muon exhibits greater robustness to low-precision quantization compared to AdamW, suggesting its potential advantage for low-precision training of large language models.

