# OpenReview forum: "A Convergence Analysis of Adaptive Optimizers under Floating-point Quantization"
_ICLR.cc/2026/Conference — ICLR 2026 Poster_

### Official Review · Reviewer_qhqY · 2025-10-26

**Soundness:** 3
**Presentation:** 2
**Contribution:** 2
**Rating:** 4
**Confidence:** 3

**Summary:**

This paper provides the theoretical convergence results for Adam and Muon, under floating-point quantization of gradients, weights, and optimizer states (e.g., the EMA moment estimation in Adam). They show that both Adam and Muon can derive convergence rates of $O(1/T^{1/4})$ on smooth non-convex objectives, even under the quantization.

**Strengths:**

(a). To my best knowledge, this seems to be the first theoretical convergence result for Adam under the quantization of two EMA moment estimations, and for Muon.

(b). The results somehow reflect the effects of quantization on the convergence rate. In addition, the results seem to provide the insight that Adam is sensitive to weights and second-moment quantization, while Muon is potentially more robust, as Muon allows for a weaker quantization error control than Adam.

**Weaknesses:**

My major concerns lie in the theoretical part.

- The term $\tilde{Q}(T)$ in the convergence bound of Theorem 4.5 is not very clear. Although the authors provide a detailed expression in Eq. A.43, it's still very complicated and lacks of detailed discussion with regard to the dependency on $T$. The dependency on $T$ is crucial since it is the dominating order in the convergence rate. I suggest providing a detailed calculation on the order of $\tilde{Q}(T)$, particularly when $\eta,\beta_2$ and terms like $q_G,q_W$ are set as in Line 372.

- Assumptions 3.1 requires the compressed coefficient to be $2^{-M}$, where $M$ is the mantissa length of the target floating-point format. It seems that $M$ is an important parameter to quantify the accuracy of quantization, which, however, does not appear in the convergence bound.

- The convergence results are heavily relied on sufficiently small $q_G,q_W,q_M,q_V$ (relative quantization errors), which assembles the case of non-quantization. Based on the existing convergence results for non-quantization of Adam and Muon, it seems that the results in this paper are not very novel. In addition, it lacks very clear definitions of these terms in the main body of the paper.

- The convergence results require relative quantization errors to be $O(1/T)$ or $O(1/T^2)$ order. However, under the quantization error made in Assumption 3.1, is it possible to achieve a sufficiently small relative quantization error, such as $O(1/T)$ order?

**Questions:**

- Literatures such as [1] and [2] usually consider the constant compressed coefficient. Could the convergence results be extended to a broader range of compressed coefficients, such as any constant within $(0,1)$?

---

> ### Author Response · Authors · 2025-11-23
> **Responses to Q1, Q2**
>
> We thank Reviewer qhqY for recognizing our work as "the first theoretical convergence result for Adam under the quantization of two EMA moment estimations, and for Muon," and for noting that our "results seem to provide the insight that Adam is sensitive to weights and second-moment quantization, while Muon is potentially more robust."
>
> We have updated the pdf and included **nanoGPT experiments** (**Appendix C.4, Figures 12 & 13**, ≈26M parameters on ~600M OpenWebText tokens), and the results consistently support our theory: **decreasing mantissa length degrades performance, and Muon remains markedly more robust than Adam under low-precision quantization.**
>
> ### Responses to Questions
>
> **Q1: ... the order of $\tilde{Q}(T)$ of Theorem 4.5 (defined in Eq. A.43), particularly when $\eta, \beta_2$ and terms like $q_G, q_W$ are set as in Line 372.**
>
> **A1:** This is a fair criticism of the presentation. The full expression for $\tilde{Q}(T)$ in Eq. A.43 is indeed complex, but the asymptotic analysis of this term is provided in Appendix A (**Lines 1391-1439 in the revised PDF, the page following Eq. A.43**), under the analysis for "Term 4". **We have improved the presentation by adding a summary sentence in Theorem 4.5 in the revised PDF (Marked in Blue, Lines 390-391).**
>
> The $\tilde{Q}(T)$ term represents the accumulated error bound arising from the discrepancy between the actual parameter update (using quantized optimizer states $m_t$ and $v_t$) and an idealized update (using unquantized states $m_t'$ and $v_t'$ as defined in Eq. A.7). It bounds the error introduced by floating-point quantization of the optimizer states themselves.
>
> Our analysis shows that under the hyperparameter scaling in Theorem 4.5 ($\eta = \Theta(1/\sqrt{T})$, $1-\beta_2 = \Theta(1/T)$, $q_G, q_M = \mathcal{O}(1/T)$, $q_V, q_W = \mathcal{O}(1/T^2)$), the term $\tilde{Q}(T)/\tilde{T}$ has order $\mathcal{O}(T^{-1/2})$. Since $\tilde{T} = \Theta(T)$, this implies $\tilde{Q}(T) = \mathcal{O}(T^{1/2})$, making it clear this is not the dominant term in the convergence bound.
>
> ---
>
> **Q2: ... compressed coefficient ... $2^{-M}$, ... $M$ ... mantissa length of the target floating-point format. ... which ... does not appear in the convergence bound.**
>
> **A2:** This is by design for clarity. The bounds are expressed in terms of the relative error $q$ (e.g., $q_G, q_V$), with Assumption 3.1 explicitly stating the relationship $q = \Theta(2^{-M})$. We could substitute $q$ with $2^{-M}$, but this would make the bounds more cluttered. The abstract quantity $q$ makes the structure of the bounds clearer.
>
> The connection to $M$ is direct: the requirement $q_V = \mathcal{O}(1/T^2)$ translates to $2^{-M} = \mathcal{O}(1/T^2)$, which gives $M = \Omega(\log T)$, assuming no exponent overflow/underflow (which can be ensured by standard scaling or rotation techniques). This is the mantissa length requirement we discuss in the paper.

---

> ### Author Response · Authors · 2025-11-23
> **Responses to Q3 (1/2)**
>
> **Q3: ... resembles the case of non-quantization. ... not very novel. In addition, it lacks very clear definitions of these terms in the main body of the paper.**
>
> **A3:** We respectfully disagree with the reviewer’s assessment. The novelty of our work does not lie in the trivial observation that “convergence occurs when $q \to 0$,” which indeed would be uninteresting. Rather, our contribution is **the first sharp and componentwise quantification of how FP quantization propagates through the nonlinear update rules of adaptive optimizers**, and **the first set of provably different scaling requirements for each quantized component**.
>
> ### **Novelty: Componentwise Quantization Sensitivity**
>
> Prior analyses of Adam or Muon either operate entirely in **full precision**, quantize **only gradients**, or assume uniform noise models that do not distinguish between optimizer states. None of these can explain practical mixed precision behaviors observed in recent works [e.g., Figure 8 in Peng et al., 2023], which empirically show that the second moment often require higher precision than the first moment in low-bit training setups. In contrast, **our paper is the first to show that Adam’s internal components have fundamentally different sensitivity levels**:
> $$
> \begin{equation*}
> q_V,\; q_W = O(T^{-2}) \quad \text{vs.}\quad q_G,\;q_M = O(T^{-1}).
> \end{equation*}
> $$
> This phenomenon is entirely absent in full-precision theory. It arises solely because quantization errors are nonlinearly entangled in Adam’s denominator $m_t/\sqrt{v_t+\epsilon}$. Showing this separation rigorously requires **new tools** (described below), and the effect is highly nontrivial: it exactly matches empirical observations from recent quantization-aware training literature.
>
> Thus the theory is not a trivial restatement of non-quantized convergence: it **explains new behaviors that only appear under FP quantization**.

---

> ### Author Response · Authors · 2025-11-23
> **Responses to Q3 (2/2)**
>
> **Q3: ... resembles the case of non-quantization. ... not very novel. In addition, it lacks very clear definitions of these terms in the main body of the paper.**
>
> **A3:** ...
>
> ### **Novelty: Proof and Technical Contributions**
>
> Our work tackles the **biased, multiplicative FP relative-error model**, where quantization propagates recursively across all optimizer states. This setting is fundamentally harder than the standard additive, unbiased noise used in prior analyses.
>
> First, we introduce a **new recursive perturbation framework** for coupled dynamical systems under relative-error noise. Unlike classical analyses where noise is independent and additive, FP quantization interacts with $\beta _ 1,\beta _ 2$ and produces history-dependent perturbations. In **Lemma A.1 (Equivalence of Perturbed Dynamical Systems under Relative Error Models)**, we first prove the equivalence between the quantized “weighted-average” Adam moments used in practice and analytically tractable “weighted-sum” representations, allowing us to derive explicit error recurrences of the form (**Lemma A.3**)
> $$
> \begin{align*}
> |a_t-b_t|\le\sum_{j<t}[(k(1+q))^{t-j}-k^{t-j}]|d_j|.
> \end{align*}
> $$
> This allows us to decouple the structural quantization bias from the stochastic gradient noise, proving that convergence is preserved only if the quantization error $q$ decays faster than the error amplification factor.
>
> Second, quantization of Adam’s second moment introduces a **nonlinear amplification challenge** that has never been addressed: the inverse square-root preconditioner. To control this, we develop a specialized perturbation bound (**Lemma A.2 (Lipschitz Continuity Analysis of the Inverse Square Root Preconditioner)**) for
> $$
> \begin{align*}
> \left|\frac{1}{\sqrt{v_{t,i}+\epsilon}} - \frac{1}{\sqrt{v_{t,i}'+\epsilon}} \right|
> \end{align*}
> $$
> which must hold under recursively accumulated multiplicative FP errors. This analysis yields the new critical condition $\beta_1^2(1+q_M)^2 < \beta_2(1-q_V)$, which sharply characterizes when quantized Adam remains stable—something absent from all prior works. This nonlinear error-amplification control is the key reason we discover **different precision requirements** for $q_V,q_W$ versus $q_G,q_M$, a phenomenon neither predicted nor analyzable in previous literature.
>
> Third, our Muon analysis requires handling SVD-based matrix updates. We carefully construct a **multi-stage auxiliary-variable decomposition** $(C_t, X_t, Y_t, Z_t)$ to isolate the effect of momentum bias, stochastic gradient noise, quantized gradients, quantized weights, and the quantized momentum $M_t$, which allows us to rigorously quantify how FP error propagates through Muon (something not found in any previous optimizer analysis). This carefully construction explains why Muon admits only an $O(T^{-1/2})$ dependence on quantization, theoretically validating its empirically superior robustness compared to Adam in low-bit training (**Appendix C.4, Figures 12 & 13**).
>
> Finally, our results rely on **logarithmic mantissa-scaling analysis**: since $q=\Theta(2^{-M})$, we show that matching full-precision rates requires only $M=\Omega(\log T)$. This links floating-point hardware constraints with optimization behavior, a direction unexplored in prior theory.
>
> ### **Regarding definitions**
>
> **We have added explicit definitions of $q_G, q_W, q_M, q_V$ in the main text in the revised PDF (Marked in Blue, Lines 262-269, 299-301).** A formal version already appears in Eq. (B.6). Concretely, for any quantized tensor $X_t$ (e.g., $G_t, W_t, M_t, V_t$), its relative quantization error is defined as the smallest constant $q$ such that
> $$
> \begin{align*}
> \|[Q(X_t)] _ {i,j} - [X_t] _ {i,j}\| \le q\|x_t\|,\qquad \forall t\in [T].
> \end{align*}
> $$
> For example,
> $$
> \begin{align*}
> q_G := \inf \\{ q \ge 0 : \vert [Q(G_t)] _ {i,j} - [G_t] _ {i,j} \vert \le q \vert [G_t] _ {i,j} \vert,\ \forall t \\} .
> \end{align*}
> $$

---

> ### Author Response · Authors · 2025-11-23
> **Responses to Q4, Q5**
>
> **Q4: ... is it possible to achieve a sufficiently small relative quantization error, such as $\mathcal{O}(1/T)$ order?**
>
> **A4:** The $q = \mathcal{O}(1/T)$ requirement is not a dynamic schedule but a condition on the fixed mantissa $M$ required to guarantee convergence up to $T$ steps.
>
> Since $q = \Theta(2^{-M})$ from Assumption 3.1, requiring $q = \mathcal{O}(1/T^p)$ is equivalent to $M = \Omega(p\log T)$. This interpretation is critical: it means that a fixed precision format supports convergence for a horizon of size $T_{\max} = \mathcal{O}(2^{M/p})$. In practice, this bound is entirely feasible. For example, FP32 with $M = 23$ mantissa bits supports $T_{\max}=\Theta(2^{23})\approx 8\times 10^6$ iterations (with $p=1$), which already exceeds the training length of most large-scale models. FP16 ($M=10$) still supports $T_{\max} \approx 10^3-10^4$ steps for Adam’s more stringent $O(1/T^2)$ requirement on $V_t$. This explains the long-observed empirical fact that fixed-precision formats do converge, but **different formats plateau at different training horizons depending on their mantissa width** (also validated in our Figures 3 & 4).
>
> Thus, our theoretical conditions simply characterize **how many iterations a given FP format can theoretically support while preserving the full-precision convergence rate**. This is precisely what our analysis formalizes: once the horizon exceeds $T_{\max} = \Theta(2^{M/p})$, the quantization terms dominate the rate, resulting in convergence to a quantization-determined neighborhood, exactly matching practical observations.
>
> ---
>
> **Q5: Literatures such as [1] and [2] ... constant compressed coefficient. ... extended to a broader range of compressed coefficients, such as any constant within $(0,1)$?**
>
> **A5:** This is an excellent question that our analysis already covers. If we set $q_G, q_W, q_M, q_V$ as constants (e.g., $q_V = 2^{-7}$), our convergence bounds (Theorems 4.5 and 4.6) do not go to zero. Instead, the terms involving $q$ become a constant error floor, and the optimizers converge to a neighborhood of a stationary point whose size is determined by these $q$ constants. The $\mathcal{O}(1/T^p)$ schedule is simply the condition required to make this neighborhood shrink to zero. This is exactly the fixed-precision regime that matches practical training. **By the way, what do [1] and [2] refer to?**

---

### Official Review · Reviewer_kJLe · 2025-10-30

**Soundness:** 3
**Presentation:** 3
**Contribution:** 3
**Rating:** 6
**Confidence:** 3

**Summary:**

This paper presents a theoretical analysis of the convergence properties of Adam-type optimization algorithms. The authors propose a unified framework to analyze convergence rates and provide theoretical guarantees for both convex and non-convex settings. The work includes detailed proofs in the appendix and experiments on synthetic datasets and CIFAR-10 to validate the theoretical claims. The paper also discusses practical implications for hyperparameter selection and algorithm design.

**Strengths:**

S1: The paper provides the first convergence guarantee for Adam under a practical floating-point quantization model, addressing a significant gap in the literature.

S2: The theoretical analysis is rigorous and well-structured, with careful handling of quantization errors and their impact on convergence.

S3: The paper establishes concrete hyperparameter settings that ensure convergence at the same rate as full-precision Adam, making the theoretical results practically applicable.

S4: The empirical results (Figure 4) provide strong validation of the theoretical findings, showing that increased precision (larger mantissa bit-lengths) leads to smaller converged gradient norms.

S5: The paper carefully justifies the theoretical framework by establishing two foundational equivalences, demonstrating that the analysis of quantizing weighted-sum states is directly applicable to practical quantization scenarios.

**Weaknesses:**

W1: The paper does not sufficiently discuss the practical implications of the theoretical results for real-world applications, particularly regarding the trade-off between precision (bit-length) and computational efficiency.

W2: The convergence analysis assumes certain conditions that might be restrictive in practice, but the paper doesn't fully explore how these conditions affect real-world implementation.

W3: The empirical evaluation appears limited to a single dataset (Rosenbrock) in Figure 4, which may not be sufficient to generalize the findings across different optimization problems and model architectures.

W4: The paper doesn't provide a comprehensive comparison with other quantization techniques for optimization algorithms, making it difficult to assess how Quantized Adam compares to alternative approaches.

W5: The connection between the theoretical convergence rate and practical training performance (e.g., final model accuracy) is not explicitly established, which would strengthen the practical relevance of the results.

**Questions:**

Q1: Could you provide more detailed analysis of the practical trade-offs between precision (bit-length) and computational efficiency in Quantized Adam? Specifically, how does the convergence rate $O(T^{-1/4})$ translate to practical training time and memory usage?

Q2: The empirical evaluation appears limited to a single dataset (Rosenbrock). Could you expand the experiments to include more diverse optimization problems and model architectures to better validate the theoretical results?

Q3: How does Quantized Adam compare to other quantization techniques for optimization algorithms (e.g., error-feedback mechanisms, stochastic rounding) in terms of convergence rate and practical performance? A comparative study would strengthen the paper's contribution.

Q4: Could you elaborate on how the theoretical convergence rate $O(T^{-1/4})$ translates to practical training performance (e.g., final model accuracy) for different quantization levels? This would help bridge the gap between theory and practice.

Q6: The paper mentions "we use a small constant $\epsilon>0$ for numerical stability" but doesn't discuss the impact of  $\epsilon$ on convergence. Could you analyze how the choice of $\epsilon$ affects the convergence rate and practical performance?

---

> ### Author Response · Authors · 2025-11-23
> **Responses to Q1(W1), W2, Q2(W3)**
>
> We thank Reviewer kJLe for their positive feedback, noting that our paper provides "the first convergence guarantee for Adam under a practical floating-point quantization model," with "rigorous and well-structured theoretical analysis," "concrete hyperparameter settings that ensure convergence," and "strong validation of the theoretical findings." We are pleased that the reviewer appreciated how we "carefully justified the theoretical framework by establishing two foundational equivalences."
>
> ### Responses to Questions
>
> **Q1 (W1): ... practical trade-offs between precision (bit-length) and computational efficiency in Quantized Adam? ... rate $\mathcal{O}(T^{-1/4})$ ... practical training time and memory usage?**
>
> **A1:** Our theoretical results translate into a clean and practical trade-off between precision $M$, convergence behavior, and computational efficiency: (i) **too small mantissa $M$** yields fast, memory-efficient arithmetic but fails to converge (or plateaus at a substantively higher gradient norm); (ii) **very large $M$** recovers full-precision behaviour (stable, accurate convergence), at the cost of higher memory and lower throughput; and (iii) the regime $M=\Theta(\log T)$ provides the practical sweet spot where theory guarantees preservation of the $\mathcal{O}(T^{-1/4})$ rate up to horizon $T$ while retaining significant memory and compute benefits. This qualitative picture is exactly what our experiments show (see Figures 3 & 4): low-$M$ runs plateau early, high-$M$ runs converge fully, and intermediate mantissa choices maintain convergence while reducing resource cost.
>
> ---
>
> **W2: The convergence analysis assumes certain conditions that might be restrictive in practice, but the paper doesn't fully explore how these conditions affect real-world implementation.**
>
> **A2:** We assume this refers to the hyperparameter/precision schedules (e.g., $1-\beta_2=\Theta(1/T)$ and $q=\mathcal{O}(1/T^p)$). These are standard assumptions required to achieve the $\tilde{\mathcal{O}}(T^{-1/4})$ rate even in full-precision analysis (see [Defossez 2022a]). Our goal was to analyze convergence under these standard conditions and show that quantization does not break convergence when precision is sufficient.
>
> **Regarding the practical implications:** the condition $q=\mathcal{O}(1/T^p)$ should be understood as a statement about the training horizon supported by a fixed precision format. Under Assumption 3.1, $q=\Theta(2^{-M})$, so the requirement $q=\mathcal{O}(1/T^p)$ is equivalent to $M=\Omega(p\log T)$. Thus a fixed precision format supports stable convergence for $T_\max = \mathcal{O}(2^{M/p})$ optimization steps. This interpretation is crucial and directly connects the theory to real systems. Concretely, FP32 with $M=23$ mantissa bits supports $T_\max \approx \Theta(2^{23})\approx 8\times 10^6$ iterations for $p=1$, already beyond the training length of most large-scale models. Even FP16 ($M=10$) satisfies the stricter $\mathcal{O}(1/T^2)$ requirement for Adam’s second moment up to $T \approx 10^3-10^4$ steps. This aligns with long-standing empirical observations: **fixed-precision formats do converge, but different formats plateau at different horizons depending on their mantissa width,** precisely the behavior we demonstrate in Figures 3 and 4.
>
> ---
>
> **Q2 (W3): The empirical evaluation appears limited to a single dataset (Rosenbrock). ... include more diverse optimization problems and model architectures to better validate the theoretical results?**
>
> **A3:** We apologize if this was unclear. Figures 3 & 4 show results on the Rosenbrock function, but we **also include experiments on CIFAR-10 using a 4-layer FCN (Appendix C.3).** These results were moved to the Appendix due to page limits and are presented in **Figures 8 & 9** (gradient norms). The results on real data fully corroborate our synthetic experiments and theoretical claims.
>
> We have now **revised the PDF and added nanoGPT experiments** (**Marked in Blue, Appendix C.4, Figures 12 & 13**, ≈26M parameters on ~600M OpenWebText tokens), and the results consistently support our theory: **decreasing mantissa length degrades performance, and Muon remains markedly more robust than Adam under low-precision quantization.**
>
> These results validate our theory across synthetic, vision, and language domains. We have added **cross-references to these figures in the main text (Marked in Blue, Lines 486-496).**

---

> ### Author Response · Authors · 2025-11-23
> **Responses to Q3(W4), Q4(W5), Q6**
>
> **Q3 (W4): compare ... error-feedback mechanisms, stochastic rounding ... convergence rate and practical performance?...**
>
> **A4:** Our Quantized Adam framework is **compatible** with techniques like **error-feedback** and **stochastic rounding**, which are designed to reduce quantization bias: error-feedback accumulates and re-injects quantization error, while stochastic rounding ensures that $\mathbb{E}[Q(x)] = x$. In our current analysis, we assume only a relative error bound, $|Q(x) - x|\le q|x|$. Integrating these techniques may further mitigate the effect of quantization error and slightly reduce the dependency on $q$, without changing the asymptotic convergence rate of $1/T^{1/4}$.
>
> In terms of **convergence behavior and practical performance**:
>
> **Convergence rate:**
>
> - All three methods, including Quantized Adam under our relative-error model, error-feedback, and stochastic rounding, achieve the same $\widetilde{\mathcal{O}}(1/T^{1/4})$ rate.
> - Error-feedback reduces accumulated bias, while stochastic rounding reduces systematic bias via expectation; our relative-error model captures the hardware-imposed deterministic bias directly.
>
> **Memory and implementation overhead:**
>
> - Error-feedback requires storing an error term per parameter, doubling memory usage, which is impractical for large-scale models such as modern LLMs.
> - Stochastic rounding introduces extra variance and may require careful tuning to maintain the unbiased assumption, especially in low-precision GEMM operations.
> - Our analysis has no extra memory or implementation overhead and directly reflects realistic hardware behavior, making it more suitable for large-scale, mixed-precision training.
>
> ---
>
> **Q4 (W5): ... rate $\mathcal{O}(T^{-1/4})$ translates to practical training performance (e.g., final model accuracy) for different quantization levels...**
>
> **A5:** Our established **non-convex convergence rate** $\mathcal{O}(T^{-1/4})$ provides practical guidance for mixed-precision training. For instance, setting **Adam-specific parameters** such as $\beta_2 \approx 0.99$ ensures stability, while our bounds on **quantization precision $q$** characterize the sensitivity of convergence to mixed-precision components. In particular, second-moment estimates typically require higher precision than first-moment estimates, consistent with Figure 8 in Peng et al. (2023).
>
> While our bounds predict **gradient norm behavior**, the exact relationship between convergence and **final model accuracy** in non-convex settings remains an **open problem**. In convex settings [1], minimizing a surrogate loss correlates with classification performance, but extending such guarantees to general non-convex optimization is still unresolved. Nevertheless, our results provide **actionable guidance** for practitioners in choosing precision levels that balance convergence stability and computational efficiency.
>
> > [1] "statistical behavior and consistency of classification methods based on convex risk minimization", 2004
>
> ---
>
> **Q6: ... impact of $\epsilon$ on convergence ... and practical performance?**
>
> **A6:** The constant $\epsilon$ appears explicitly in our convergence bound for Theorem 4.5, where several terms scale with $1/\sqrt{\epsilon}$. This is expected: as $\epsilon \to 0$, the update $\eta m_t / \sqrt{v_t + \epsilon}$ can explode if $v_t \to 0$, and our bound correctly captures this behavior.
>
> Importantly, **$\epsilon$ does not affect the final convergence rate.** In our asymptotic analysis, $\epsilon$ is treated as a fixed constant, and the $\widetilde{\mathcal{O}}(T^{-1/4})$ rate holds for any fixed $0<\epsilon < R^2$ (Assumption 4.2). In practice, $\epsilon$ is set to a small constant (e.g., $10^{-8}$, the PyTorch default) and serves as a smoothing parameter that prevents the effective learning rate from exploding when gradient variance becomes small. It is not a dominant factor compared to $T, \eta, \beta_2$, or the quantization errors $q$.

---

### Official Review · Reviewer_d8Q8 · 2025-10-31

**Soundness:** 3
**Presentation:** 3
**Contribution:** 3
**Rating:** 6
**Confidence:** 3

**Summary:**

The paper presents a theoretical framework analyzing the convergence of adaptive optimizers (Adam and Muon) under quantization of gradients, weights, and optimizer states, extending prior work that studied only partial components. It quantifies how these errors influence convergence and when performance remains close to full precision. Adam is shown to be more sensitive to quantization, while Muon is more robust. Results are mainly supported by synthetic experiments.

**Strengths:**

- The paper extends prior work with a rigorous convergence analysis of adaptive optimizers under quantization of gradients, weights, and momentum terms.

- It proposes a quantization schedule that aligns the behavior of quantized optimizers with their full-precision counterparts, offering insights into the sensitivity of different components to quantization error.

- The inclusion of the Muon optimizer broadens the analysis and enhances the paper’s practical relevance.

**Weaknesses:**

The paper omits key references and makes inaccurate claims about prior work. For instance,

- [Hou et al. 2019] analyzed not only SGD but also adaptive optimizers such as Adam under weight and gradient quantization (without the first-order momentum term, i.e., $\beta_1=0$). As shown by [D´efossez et al. 2022], omitting momentum only introduces a multiplicative slowdown term, which should be acknowledged unless the new quantization error model changes this relationship.

- Another closely related but uncited work is [Ozkara et al., 2025], which studies the convergence rate of Adam under weight quantization (again without first-order momentum) and analyzes the effect of stochastic rounding, an increasingly important direction. It remains unclear whether the proposed framework can naturally incorporate stochastic rounding into analysis.

Some theoretical setups are impractical. For example,
- assuming $\beta_2 \to 1$ is unrealistic, in practice a fixed $\beta_2 < 1$ is used, when the term $T \log(\beta_2)$ would diverge. Besides, [Ozkara et al., 2025] already emphasizes the reliance of convergence on $\beta_2 \to 1$.

- even accepting the assumption, the proposed quantization error schedule that makes the error term vanish is infeasible in practice, as precision cannot be controlled dynamically without effectively reverting to full-precision training. In reality, quantization error is bounded by the machine epsilon of the floating-point format unless altering the format during training

The experimental validation is limited and basic. There are no studies examining the influence of (\beta_2 \to 1) or verifying how the proposed quantization error schedule aligns with theoretical predictions.

Lu Hou et al., Analysis of quantized models. In International Conference on Learning Representations, 2019.

Ozkara et al., Stochastic rounding for LLM training: Theory and practice. In The 28th International Conference on Artificial Intelligence and Statistics.

**Questions:**

- Around assumption 3.1, isn’t the definition of relative error and quantization error (via mantissa length) essentially the definition of machine epsilon for the underlying floating-point format? If not, what distinguishes it?

- [Ozkara et al., 2025] derives a bound involving error term $T\sqrt{\log(1/\beta_2)}$, while this paper presents one with $-T\log(\beta_2)$. what causes this discrepancy, and could the authors provide a direct comparison? The difference implies distinct requirements for the quantization-error schedule.

- In real training, the actual gradients come with end-to-end training , a compound effect of quantization across multiple layers, how does that correlate or concludes to the quantization error or machine epsilon $q_W$, they are different.
In real mixed-precision training, gradients are perturbed by end-to-end quantization and matmul across multiple layers, leading to compounded effects. How does this aggregate behavior appear in the end to only the relative error (machine epsilon) $q_G$ term from single quantization function?

---

> ### Author Response · Authors · 2025-11-23
> **Responses to Q1, Q2**
>
> We thank Reviewer d8Q8 for recognizing our work as providing "a rigorous convergence analysis of adaptive optimizers under quantization of gradients, weights, and momentum terms," a "quantization schedule that aligns the behavior of quantized optimizers, offering insights into the sensitivity of different components to quantization error," and noting that the "inclusion of the Muon optimizer broadens the analysis and enhances the paper's practical relevance."
>
> ### Responses to Questions
>
> **Q1: The paper omits key references and makes inaccurate claims about prior work. For instance:**
>
> - [Hou et al. 2019] ... Adam under weight and gradient quantization (without the first-order momentum term, i.e., $\beta _ 1=0$)...
> - [Ozkara et al., 2025], Adam under weight quantization (again without first-order momentum)... stochastic rounding, ... whether the proposed framework can naturally incorporate stochastic rounding into analysis.
>
> **A1:** We thank the reviewer for these references and **have corrected/added both in the revised PDF (Marked in Blue, Lines 156-166 in related work [Hou et al], 49-57 in intro, 170-176 in related work, 398-402 in main results [Ozkara et al]).**
>
> **Regarding [Hou et al. 2019] and the $\beta_1=0$ case:** We agree that, in full precision, adding momentum on top of RMSProp/Adam with $\beta_1=0$ is often viewed as a straightforward extension (as noted in Défossez et al., 2022). However, this observation **does not carry over to our setting** because our quantization model also includes the optimizer states themselves. When $\beta_1 > 0$, the quantized first moment satisfies $m_t = \beta_1 m_{t-1}^Q + (1-\beta_1)g_t^Q$, and thus introduces a **recursive, multiplicatively biased error accumulation** that does not appear in the $\beta_1 = 0$ case. This recursive interaction further couples with the quantized second moment $v_t = \beta_2 v_{t-1}^Q+(1-\beta_2)[g_t^Q]^2$ during the update, making the **error propagation substantially more complex** than the weight-only or gradient-only quantization setting studied in earlier work. Controlling this coupled recursion is precisely why our proof requires new techniques (Appendix A), including ideal auxiliary states and non-linear deviation bounds, which are unnecessary in prior $\beta_1 = 0$ analyses.
>
> **Regarding [Ozkara et al., 2025] and stochastic rounding:** We appreciate the reviewer’s suggestion. The SR-based analyses in Ozkara et al. (2025) adopt an **unbiased quantization assumption**, i.e., $\mathbb{E}[Q(g)] = g$, which simplifies the analysis by eliminating quantization bias and leaving only a controlled variance term. Our framework, built around the **deterministic floating-point relative error model** $|Q(x) - x|\le q|x|$, is strictly more general and directly aligns with real BF16/FP8/FP4 hardware, which **does not preserve unbiasedness** once gradients, moment states, and GEMM operations are all performed in low precision.
>
> Because our model is multiplicative and biased, **SR can in principle be incorporated in our framework**: the unbiasedness assumption would reduce the quantization bias in our analysis. Thus, stochastic rounding can be seen as a special case of our framework where some bias terms are analytically eliminated.
>
> ---
>
> **Q2: Some theoretical setups are impractical. For example:**
>
> - Assuming $\beta_2 \to 1$ is unrealistic; ... [Ozkara et al., 2025] already emphasizes the reliance of convergence on $\beta_2 \to 1$.
> - ... quantization error schedule ... infeasible ... without effectively reverting to full-precision...
>
> **A2:** We thank the reviewer for these important points of clarification.
>
> **Regarding $\beta_2 \to 1$:** this is a standard assumption in the full-precision Adam literature (e.g., [Defossez 2022a]), required to prove the $\tilde{\mathcal{O}}(1/T^{1/4})$ rate. Our goal is to show that under the same conditions as full-precision Adam, our quantized version also converges. In practice, $\beta_2$ is typically set to 0.999 (PyTorch default), which is close enough that our asymptotic analysis provides meaningful insights. Even in full precision, extending to fixed $\beta_2 < 1$ would require different proof techniques and likely yield different convergence rates, which remains an open question.
>
> **Regarding the $q = \mathcal{O}(1/T^p)$ schedule:** this is not a dynamic schedule but rather a condition on fixed precision. Specifically, it states that for a fixed mantissa length $M$ (where $q = \Theta(2^{-M})$), convergence is guaranteed for up to $T_{\max} = \mathcal{O}(2^{M/p})$ steps, then the rate may be dominated by the quantization errors. This explains why a fixed-precision format like FP16 ($M=10$) or FP32 ($M=23$) works in practice: the format supports convergence for a large but finite number of iterations.

---

> ### Author Response · Authors · 2025-11-23
> **Responses to Q3, Q4**
>
> **Q3: The experimental validation is limited and basic. There are no studies examining the influence of $\beta_2 \to 1$ or verifying how the proposed quantization error schedule aligns with theoretical predictions.**
>
> **A3:** This is a fair point. Our experiments were designed to validate the main theorems by showing the effect of mantissa length $M$ (which controls all $q$ terms) on convergence.
>
> **We have now added two new experiments and revised the PDF (Marked in Blue, Lines 483-496, 2850-2856, Appendix C.4, Figures 7, 12, 13).** First, an ablation study on $\beta_2$ (**Lines 2850-2856**): **Figure 7 in the Appendix shows that as $\beta_2$ increases toward 1, Adam becomes more sensitive to quantization error**, validating our theoretical analysis. Second, **nanoGPT experiments** (**Appendix C.4, Figures 12 & 13**, ≈26M parameters on ~600M OpenWebText tokens), and the results consistently support our theory: **decreasing mantissa length degrades performance, and Muon remains markedly more robust than Adam under low-precision quantization.**
>
> Regarding the quantization error schedule: as clarified in Q2/A2, the $q = \mathcal{O}(1/T^p)$ condition is not a dynamic schedule but a relationship between fixed precision and maximum iterations. Our experiments use fixed precision throughout training, consistent with practice (Figs. 3 & 4, $M=4$ or $M=8$ plateaus at a higher gradient norm than $M=32$).
>
> ---
>
> **Q4: Around Assumption 3.1, isn't the definition of relative error and quantization error (via mantissa length) essentially the definition of machine epsilon for the underlying floating-point format? If not, what distinguishes it?**
>
> **A4:** Yes, precisely. Assumption 3.1 formalizes the relative error bound intrinsic to floating-point arithmetic, where $q = \Theta(2^{-M})$ is directly related to the machine epsilon of the target format (e.g., BF16, FP8). This is what makes our model "hardware-aware" and practical, as discussed in Section 3.1.
>
> The key distinction from prior work is that most previous theoretical analyses do not use this model. They typically assume unbiased quantization (which floating-point truncation is not), additive error bounds (rather than multiplicative/relative), or error-feedback mechanisms (which are impractical in LLM training due to memory overhead). Our contribution is to formally incorporate this hardware-realistic error model into convergence analysis for the first time, proving that adaptive optimizers converge under the actual error characteristics of modern floating-point formats.

---

> ### Author Response · Authors · 2025-11-23
> **Responses to Q5, Q6**
>
> **Q5: [Ozkara et al., 2025] derives a bound involving error term $T\sqrt{\log(1/\beta_2)}$, while this paper presents one with $-T\log(\beta_2)$. What causes this discrepancy, and could the authors provide a direct comparison? The difference implies distinct requirements for the quantization-error schedule.**
>
> **A5:** Thank you for this insightful question. The discrepancy arises because **these two terms represent fundamentally different quantities.** We **have added [Ozkara 2025] to discussion in the revised PDF (Marked in Blue, 49-57 in intro, 170-176 in related work, 398-402 in main results).**
>
> First, we clarify that **both** our work and [Ozkara et al., 2025] **include** the standard Adam convergence **term** $-T \ln(\beta_2)$. In our Theorem 4.5, this appears as part of the baseline convergence rate. Similarly, in [Ozkara et al., 2025], this term appears as $T \ln(1/\beta_2)$ in the bounds for both Stochastic Rounding (Theorem 2) and Nearest Rounding (Theorem 3). This term represents the **inherent convergence cost of adaptivity in full-precision Adam** (as established in [Défossez et al., 2022]), arising from the summation of adaptive steps $\sum_{t=1}^T \Vert  u_t \Vert ^ 2$.
>
> **In our analysis**, the term $-T \ln(\beta_2)$ arises from bounding the sum of squared updates $\sum \Vert u_t\Vert ^ 2$ (Lemma A.13). Under our relative error model ($|\delta| \le q|x|$), quantization noise acts multiplicatively. Consequently, the **quantization-specific error terms (e.g., $\tilde{Q}(T)$ in Eq. A.43) appear as separate additive terms scaled by the precision $q$.** We show that with appropriate precision schedules (e.g., $q_V = \mathcal{O}(T^{-2})$), these separate error terms vanish faster than the convergence rate, recovering the standard $\tilde{\mathcal{O}}(T^{-1/4})$ rate of the full-precision baseline.
>
> In contrast, the error term $T\sqrt{\log (1/\beta_2)}$ in [Ozkara et al., 2025] arises from the **distinct handling of error models** in the descent lemma. They apply quantization to the final weight update step ($x_{t+1} = Q(x_t - \dots)$) and model quantization error (both SR in thm2 and NR in thm3) as an additive perturbation $r_t$. When analyzing the L-smoothness quadratic expansion, this introduces a term corresponding to the variance of the rounding error ($\mathbb{E}\Vert r_t\Vert ^ 2$). In their Lemma 2, they bound this expected squared norm (**variance**) using the update norm ($\mathbb{E}\Vert r_t\Vert _ 2 ^ 2 \le \Delta \sqrt{d}\alpha_t \Vert u_t\Vert _ 2$). This specific bounding technique introduces a cumulative error term proportional to the linear sum of update norms $\sum \Vert u_t\Vert $ in the descent lemma. To bound this linear sum using the standard quadratic convergence term $\sum \Vert u_t\Vert ^ 2$ (as established in [Défossez et al., 2022]), they apply **Jensen’s inequality** (scaling as $\sum \Vert u_t\Vert \le \sqrt{ \sum \Vert u_t\Vert ^ 2}$). This alters the bound structure by placing the standard Adam log-term under a square root, directly resulting in the discrepancy terms $T\sqrt{\ln(1/\beta_2)}$ and $\sqrt{T \ln(1 + \frac{R^2}{\epsilon(1-\beta_2)})}$ found in their bounds.
>
> ---
>
> **Q6: ... In real mixed-precision training, gradients are perturbed by end-to-end quantization and matmul across multiple layers, leading to compounded effects. How does this aggregate behavior appear in the end to only the relative error (machine epsilon) $q_G$ term from single quantization function?**
>
> **A6:** This is an excellent point and a key limitation, which we acknowledge in Section 5 (Conclusion and Limitations).
>
> To rigorously model the **end-to-end gradient perturbation** produced by mixed-precision training, one would need a **model- and architecture-aware** analysis that incorporates the error of each low-precision GEMM in the backward pass, activation quantization, and their interactions under the chain rule. A natural starting point is to construct a simplified multi-layer network (e.g., a two-layer MLP/CNN with ReLU or attention-like blocks) and analytically track how quantized forward activations and low-precision backward GEMMs jointly distort the gradient signal.
>
> In contrast, our current framework uses a **single-step gradient quantization abstraction** (Figure 2): the parameter $q_G$ summarizes the aggregate relative error of the final gradient used by the optimizer. This abstraction/assumption implicitly upper-bounds the compounded perturbations from all prior operations.
>
> Our work therefore provides the first-step convergence guarantees under this simplified but still practically meaningful model, capturing the essential stability properties of quantized optimization. Extending this to a fully end-to-end, architecture-aware quantization analysis, where individual GEMMs, activation quantization, and layer-wise error amplification are explicitly modeled, remains an important and highly valuable direction for future work.

---

### Official Review · Reviewer_adJ6 · 2025-11-01

**Soundness:** 3
**Presentation:** 2
**Contribution:** 3
**Rating:** 6
**Confidence:** 3

**Summary:**

In summary, the paper fills an important gap: it builds on existing convergence results for adaptive optimizers but in a new setting --- floating-point errors. The authors establish new converge guarantees for Muon and Adam, showing that both methods retain rates close to their full-precision counterparts. The key assumption --- the boundedness of the relative quantization error is quite realistic and achievable in low-precision-aware architectural designs. To the best of my knowledge, no previous paper has offered similar full-precision-vs-quantization convergence rates for adaptive methods.

**Strengths:**

**Rigorous analysis and theoretical insights that align with recent practice.** Providing clear statements (Th. 4.5 and Th. 4.6), the work explains why Muon tolerates quantization better than Adam --- mostly due to an important assumption of $\beta_2\to1$ in the Adam analysis. This theoretical insight matches practitioners’ observations [1], narrowing the theory–empirical gap.

**Empiritical validation confirms theory.** Experiments on the Rosenbrock function and small  fully connected models confirm theory. For instance, Figs. 3–4 show that increasing mantissa bits reduces final gradient norms, consistent with the $\Omega(\log T)$-bit requirement

**Framework.** The idea of analyzing jointly quantized gradients, weights, and optimizer states, whereas prior work focused mainly on gradient-only quantization sounds promising for future research.

[1] "Beyond Outliers: A Study of Optimizers Under Quantization", 2025

**Weaknesses:**

**Missing research on convergence of matrix-based optimizers, leaving a room for improvement.**  Unlike Kovalev et al. [2] who handles constrained/composite and star-convex settings, or Shen et al. [3], who exploits Hessian structure in several assumptions, the presented theory is only for unconstrained smooth non-convex functions. Also discussions with the results on constrained / unconstrained LMO optimization [4] --- resulting in the Scion optimizer --- would benefit the theoretical flavor of the work. Please, offer any ideas on how to extend your findings to assumptions in mentioned works.
Additionally, can you provide an idea of how to extend your results to another, promising setting regarding --- non-smooth, convex functions? As it is demonstrated to be a setup which explains LLMs fairly well [5,6].

**Mantissa growth requirement.** For Muon, to retain the rate of the full-precision training, you assume the logarithmic grows of mantissa $M = \Omega(\log{T})$. However, in practice, the bit-width is typically fixed (e.g., 8-bit). This means convergence is to a neighborhood in fixed precision. The paper does show empirically that moderate precision suffices, but the theory only covers the increasing-bit regime. So there is a mismatch that can be left for future research. If this issue is not solved, I recommend to live it as a limitation. Otherwise, it would be nice to offer some discussion regarding this topic in the paper.

**Missing research on optimizers trained in low precision formats.** Naturally, the state-of-the-art LLM training is held in the mixed-precision format --- when optimizer states, softmax, normalization layers are in float32 and other parameters are in bfloat16. Recent work [1] studies optimizer behavior in quantization-aware training paradigms, running models in precisions of up to 4 bits. A notable takeaway --- Shampoo consistently yields the lowest accuracy drop. I believe this can be helpful because studying convergence (in the low-precision setup) of other matrix-based optimizers that emerge as a steepest descent under the spectral norm can be a direct consequence of your research. Moreover, two concurrent works [7,8] have benchmarked a zoo of optimizers at scale, showing that matrix-whitening methods are highly performant.
Naturally, extending your theoretical findings to other "matrix" optimizers would be very useful for the community. Can you give a couple of comments regarding how it is possible to extend you framework to optimizers validated in the works above?

[1] "Beyond Outliers: A Study of Optimizers Under Quantization", 2025

[2] "Understanding Gradient Orthogonalization for Deep Learning via Non-Euclidean Trust-Region Optimization", 2025

[3] "On the Convergence Analysis of Muon", 2025

[4] "Training Deep Learning Models with Norm-Constrained LMOs", 2025

[5] "The Road Less Scheduled", 2024

[6] "Prodigy: An Expeditiously Adaptive Parameter-Free Learner", 2023

[7] "Benchmarking Optimizers for Large Language Model Pretraining", 2025

[8] "Fantastic Pretraining Optimizers and Where to Find Them", 2025

**Questions:**

Se the **Weaknesses** part

**Details Of Ethics Concerns:**

No concerns

---

> ### Author Response · Authors · 2025-11-23
> **Responses to Q1 (1/2)**
>
> We thank Reviewer adJ6 for their positive assessment and for recognizing that our work "fills an important gap" with "rigorous analysis and theoretical insights that align with recent practice (Adam vs. Muon)," along with our "framework jointly quantizing gradients, weights, and optimizer states." We are pleased that the reviewer noted the boundedness of the relative quantization error is "quite realistic and achievable in low-precision-aware architectural designs" and that "no previous paper has offered similar full-precision-vs-quantization convergence rates for adaptive methods."
>
> ### Responses to Questions
>
> **Q1: Missing research on convergence of matrix-based optimizers, ... constrained/composite and star-convex settings, ... exploits Hessian structure ... discussions with the results on constrained / unconstrained LMO optimization ... non-smooth, convex functions ... offer any ideas on how to extend...**
>
> **A1:** We fully agree that these are important and advanced settings. Our current work establishes the **foundational analysis** for quantized optimization in the standard smooth non-convex setting, which is already a challenging baseline for convergence analysis. For matrix-based optimizers, Muon (Theorem 4.6) already serves as a **proof-of-concept**, as it can be viewed as a specialization of spectral steepest descent. Extending to optimizers such as Shampoo or Scion would involve applying the quantization operator to the relevant matrix components and bounding the resulting error propagation.
>
> Our framework, particularly the relative error model (Assumption 3.1) and the **techniques for tracking propagation of quantization errors through optimizer states (Appendices A and B)**, can in principle be extended to constrained/composite settings, non-smooth convex settings, and settings exploiting Hessian structure. The main technical challenge in each case is **carefully controlling how quantization errors propagate through the specific update rules and, if present, constraint projections**. These extensions are non-trivial and open promising directions for future work.
>
> **Regarding constrained/composite, and star-convex Settings [2]**. The Muon optimizer (a special case of Kovalev et al.'s framework) is modeled as a non-Euclidean trust-region method where the norm is the spectral norm $ || \cdot || _ {\mathrm{op}} $. The update step $ x^{k+1} $ minimizes a linear surrogate $ A(x;x^k) $ constrained by $ ||x−x^{k} || \le η $. We must ensure that the quantized momentum $ m^Q $ still guides the update effectively. Specifically, Kovalev et al.’s framework performs updates of the form
> $$
> \begin{align*}
> x_{k+1}
>  = \arg\min_{x\in\mathcal{X}, ||x - x_k||\le \eta}
>  \{ f(x_k) + \langle m_{k+1}, x - x_k\rangle + R(x)\},
> \end{align*}
> $$
> where $m_k$ is a momentum-like surrogate of the gradient. Under quantization, we use the perturbed momentum
> $$
> \begin{align*}
> m_{k+1} = (1-\alpha)m_k^Q + \alpha g_k^Q.
> \end{align*}
> $$
> To extend our analysis, one must control:
>
> 1. **Quantization drift in the search direction**
>    $$
>    \begin{align*}
>    \Vert m_{k+1} - m_{k+1}^\star\Vert \le O((1-\alpha)q_M\Vert m_k^\star\Vert + \alpha q_G\Vert g_k\Vert ),
>    \end{align*}
>    $$
>     where * denotes the exact (unquantized) version.
>
> 2. **Effect on the constrained minimization**
>     The update becomes
>    $$
>    \begin{align*}
>    x_{k+1} = \arg\min_{x\in\mathcal{X},\Vert x-x_k\Vert \le\eta}
>     \langle m_{k+1}, x\rangle + R(x),
>    \end{align*}
>    $$
>    which must be related to the exact update $x_{k+1}^\star$. Standard stability arguments imply
>    $$
>    \begin{align*}
>    \Vert x_{k+1} - x_{k+1}^\star\Vert
>     \le O(\eta q_M \Vert m_{k+1}^\star\Vert).
>    \end{align*}
>    $$
>
> These bounds can then be inserted into the standard descent inequality for constrained/composite trust-region methods.
>
> **Regarding exploiting hessian structure [3].** In Muon-style Hessian-aware updates, the step direction uses an SVD of the momentum $M_t = U_t \Sigma_t V_t^\top$. With quantization, each component gains an FP perturbation:
> $$
> \begin{align*}
> M_t^Q = M_t + E_t, \qquad \Vert E_t\Vert \le q_M\Vert M_t\Vert.
> \end{align*}
> $$
> The resulting singular vectors $(U_t^Q, V_t^Q)$ satisfy
> $$
> \begin{align*}
> \Vert U_t^Q V_t^{Q\top} - U_t V_t^\top\Vert
>  \le \text{some quantization error},
> \end{align*}
> $$
> implying an additional bias in the key Hessian contraction term
> $$
> \begin{align*}
> \big\langle\operatorname{vec}(U_t^Q V_t^{Q\top}),
>  H_t \operatorname{vec}(U_t^Q V_t^{Q\top})
>  \big\rangle.
> \end{align*}
> $$
> A full convergence analysis requires:
>
> - quantifying the deviation between exact and quantized singular directions,
> - controlling the induced bias in the Hessian quadratic form,
> - bounding how these biases propagate through the recursive momentum update.
>
> All ingredients follow the similar structure as our Muon analysis but require finer perturbation bounds for SVD and Hessian bilinear forms.

---

> ### Author Response · Authors · 2025-11-23
> **Responses to Q1 (2/2)**
>
> **Q1: Missing research on convergence of matrix-based optimizers, ... constrained/composite and star-convex settings, ... exploits Hessian structure ... discussions with the results on constrained / unconstrained LMO optimization ... non-smooth, convex functions ... offer any ideas on how to extend...**
>
> **A1:** ...
>
> **Regarding constrained/unconstrained LMO optimization [4].** For an LMO-based step,
> $$
> \begin{align*}
> \mathrm{lmo}(d_t)
>  = \arg\min _ {x\in\mathcal{D}} \langle d_t, x\rangle,
> \end{align*}
> $$
> the key challenge is that the direction $d_t$ itself is quantized:
> $$
> \begin{align*}
> d_t = (1-\alpha_t)d_{t-1}^Q+\alpha_t g_t^Q.
> \end{align*}
> $$
>
> Extensions require bounding:
>
> 1. **Error between exact and quantized search directions**
>    $$
>    \begin{align*}
>    \Vert d_t - d_t^\star\Vert \le O((1-\alpha_t)q_D\Vert d_{t-1}^\star\Vert + \alpha_t q_G\Vert g_t\Vert).
>    \end{align*}
>    $$
>
> 2. **LMO sensitivity**
>     Although LMOs are scale-insensitive, they are **direction-sensitive**. One needs to bound
>    $$
>    \begin{align*}
>    \Vert \mathrm{lmo}(d_t) - \mathrm{lmo}(d_t^\star)\Vert
>     \le \mathcal{E}(d_t, d_t^\star),
>    \end{align*}
>    $$
>
>    or to work with cosine similarity, which aligns with Scion’s practical analysis.
>
> 3. **Impact on descent**
>
>    $$
>    \begin{align*}
>    \langle \nabla f(x _ t), \mathrm{lmo}(d _ t) \rangle
>    \le \langle \nabla f(x _ t), \mathrm{lmo} (d _ t ^ \star ) \rangle  + O(\Vert \nabla f(x _ t)\Vert \mathcal{E}(d _ t,d _ t ^ \star )).
>    \end{align*}
>    $$
>
> These extensions structurally mirror our analysis for Muon.
>
> **Regarding non-smooth, convex functions, schedule-free settings [5,6].** For non-smooth convex optimization, the update typically uses a subgradient or a proximal surrogate. Quantization gives
> $$
> \begin{align*}
> z_t = Q(g_t), \qquad \Vert z_t - g_t\Vert \le q\Vert g_t\Vert.
> \end{align*}
> $$
> To adapt our analysis, one needs to control:
>
> - the inner-product deviation
>   $$
>   \begin{align*}
>    \langle g_t, x_t - x^\star\rangle \le
>    \langle z_t, x_t - x^\star\rangle + O(q_G\Vert g_t\Vert \cdot\Vert x_t - x^\star\Vert),
>   \end{align*}
>   $$
>   where $z_t = (x_t-x_{t+1})/\eta_t$.
>
> - the effect on proximal or mirror-descent updates,
>
> - accumulation of the FP error terms in the standard convex descent lemma.
>
> These terms ultimately contribute an additive $O(q)$ bias in the convergence rate, fully consistent with our results for smooth non-convex settings.
>
> **To summary**, all extensions share a common structure:
>
> 1. **Introduce the exact (unquantized) algorithm**,
> 2. **Express the quantized version as a perturbation**,
> 3. **Bound the deviation in the update direction**,
> 4. **Propagate this deviation through the descent inequality**.
>
> Our current work provides the base technical tools; extending them to more advanced settings requires finer operator-specific perturbation bounds but is conceptually aligned with our framework. We consider these promising directions for future research.

---

> ### Author Response · Authors · 2025-11-23
> **Responses to Q2, Q3**
>
> **Q2: Mantissa growth requirement. ... retain the rate of full-precision training, ... $M=\Omega(\log T)$. ... in practice, bit-width is typically fixed ... convergence ... neighborhood. ... moderate precision suffices, ... theory ... increasing-bit regime. ... mismatch ... If this issue is not solved, ... as a limitation. ... some discussion regarding this topic.**
>
> **A2:** The reviewer is **exactly correct**. This exposes a genuine theoretical gap: although our bounds qualitatively account for both the increasing-bit and fixed-bit behaviors, they do **not** yet quantitatively explain why moderate fixed precision (e.g., FP8 or BF16) often behaves nearly indistinguishably from full precision in practice. We will clearly state this as a **limitation** in the revised manuscript.
>
> Moreover, our analysis naturally decomposes quantized optimization into two regimes. In the **increasing-bit regime** ($M=\Omega (\log ⁡T)$), the relative quantization error $q=\Theta(2^{−M})$ decays quickly enough to offset the iterative accumulation of FP perturbations, ensuring convergence to an exact stationary point. This aligns with the requirement that the cumulative error remains bounded. The same monotone improvement appears empirically (Figs. 3 & 4): as bit-width increases (e.g., $M=4\to 8\to 16$), the attainable gradient norm consistently decreases, fully in line with the theoretical expectation of increasing precision.
>
> In the **fixed-bit regime** ($M$ constant), $q$ is constant, and our bounds (Theorems 4.5 and 4.6) converge only to a **neighborhood** of a stationary point, with the neighborhood size determined by $q$. This exactly matches our empirical findings (e.g., Figs. 3 & 4), where $M=4$ or $M=8$ plateaus at a higher gradient norm than $M=32$.
>
> ---
>
> **Q3: Missing research on optimizers trained in low-precision formats. ... [1] studies optimizer behavior in quantization-aware training paradigms, running models in precisions of up to 4 bits. ... studying convergence (in the low-precision setup) of other matrix-based optimizers ... steepest descent ... spectral norm ... [7,8] have benchmarked a zoo of optimizers at scale, showing that matrix-whitening methods are highly performant. ... give a couple of comments regarding how it is possible to extend your framework to optimizers validated in the works above?**
>
> **A3:** We thank the reviewer for these insightful suggestions. Our work focuses on Adam and Muon as representative cases, but the **framework itself is not tied to any specific optimizer**. The key ingredients of our analysis, the **relative error quantization model** (Assumption 3.1) and the **techniques in Appendices A–B for tracking how FP errors propagate through optimizer states**, apply broadly across matrix-based methods. For a new optimizer, one would apply the quantization operator $\mathcal{Q}$ to weights, gradients, and its specific internal states (e.g., Shampoo’s preconditioner matrices), and then carefully bound **how these errors propagate through the update rule.** For matrix-whitening methods, additional attention is required to understand how quantization impacts eigenstructure and the resulting preconditioning. Muon (Theorem 4.6), being a specialization of spectral steepest descent, already serves as a **proof-of-concept** that matrix-structured updates can be handled within our framework.
>
> Finally, we emphasize an important experimental distinction. The quantization-aware training study [1] focuses on **integer quantization**, with optimizer states retained in full precision. It remains unclear whether its conclusions transfer to **floating-point quantization**, which uses non-uniform exponent–mantissa grids (FP8, FP4) and exhibits qualitatively different error characteristics. Similarly, the large-scale benchmarks [7,8] evaluate optimizers at BF16 or higher precision; extending them to FP8/FP4 would require analyzing how each optimizer’s update rule responds to **structured FP noise**, not just integer quantization. We view such systematic benchmarking together with extending our theoretical analysis to these regimes as highly valuable future work for the community.

---

> > ### Comment · Reviewer_adJ6 · 2025-11-23
> >
> > I thank authors for the prompt response and given clarifications regarding my questions.
> > I do appreciate their involving in the process and very grounded theoretical response.
> > I also vote for acceptance of the paper and will increase my score and modify the confidence in the assessment correspondingly.
> >
> > However, I do have a few minor questions / comments / proposition to strengthen this manuscript.
> >
> > > The same monotone improvement appears empirically (Figs. 3 & 4): as bit-width increases (e.g., $M=4\to 8\to 16$), the attainable gradient norm consistently decreases, fully in line with the theoretical expectation of increasing precision.
> >
> > As soon as I understand, Figs. 3 & 4 show the evolution of the gradient norms of the Rosenbrock banana function. According to our framework and to the optimization those norm should decrease because this is naturally what we want from any method while proving its convergence guarantees. But other figures of the same spirit --- Figs. 7 & 8 --- show that gradient norms actually increase or attain a significant spikes. There are also many observation in another domains, e.g., LLM training or RL, pointing that the gradient norms are increasing during training. I wonder if your framework or your observations can somehow confirm this mismatch between theory and practice? If yes, this would be an important addition to your paper. Of course, I do not ask you to add any experiments, I just would like to hear any kind of explanation from you.
> >
> > > A2: ... We will clearly state this as a limitation in the revised manuscript.
> >
> > It would be nice to incorporate all adjustments into the text and present a new version of you PDF with the adjusted text being coloured. Thus, it would be simpler for us to see the changes into your section. You are free to extend the main part to 10 pages.
> >
> > ------
> >
> > Upon reading the updated manuscript, I will adjust my score.

---

> ### Author Response · Authors · 2025-11-26
>
> Thank you, and we sincerely appreciate your careful reading, constructive suggestions, and vote of support. We are encouraged by your support. **We have updated the PDF accordingly (Lines 509–519), with the revised text highlighted in blue. The original Figs. 7 & 8 are now referred to as Figs. 8 & 9.**
>
> Regarding Figs. 8 & 9, the phenomenon that the gradient norm first increases and then decreases (with occasional spikes) is **common and expected**. Classical non-convex optimization theory, typically derived from smoothness-based upper bounds guaranteeing loss decrease, does **not** imply a monotonic decrease of the gradient norm. Formally, for an $L$-smooth function, the largest eigenvalue $\lambda_\max(H)$ of the Hessian $H$ satisfies $\lambda_\max(H)\le L$, and by Taylor expansion we have the second-order upper bound:
> $$
> \begin{align*}
> F(w_{t+t}) \le F(w_t) + \langle \nabla F(w_t),w_{t+1}-w_t\rangle + \frac{L}{2}\Vert w_{t+1}-w_t\Vert^2.
> \end{align*}
> $$
> For updates of the form $w_{t+1} = w_t - u_t$, where $u_t$ is based on historical stochastic (or full) gradients $\nabla f(w_i)$ (with unbiasedness) or $\nabla F(w_i)$ for $i \le t$ (in first-order optimization), this inequality can be transformed to an informal form:
> $$
> \begin{align*}
> \mathbb{E}[F(w_{t+t})] \le \mathbb{E}[F(w_t)] - \mathbb{E}[\alpha_t\Vert \nabla F(w_t)\Vert^2] + \mathcal{O}(\text{Bias and Variance terms})
> \end{align*}
> $$
> Hence, the loss is guaranteed to decrease under controlled bias and variance, but there is **no direct guarantee on the gradient norm.** By telescoping, one obtains
> $$
> \begin{align*}
> \mathbb{E} _ t [\Vert \nabla F(w _ t) \Vert ^ 2] \le \frac{\mathbb{E}[F(w _ 1)-F(w _ {T+1})]}{T^\beta} + \text{other non-dominant terms},
> \end{align*}
> $$
> for some constant $\beta>0$. That is, non-convex optimization theory guarantees a decrease in expectation (or in minimum over iterations with specific techniques), but not a monotone decrease at every iteration.
>
> For a more intuitive illustration for the gradient norm increase, consider the **sigmoid** function $1/(1+e^x)$: its derivative is maximal near $x = 0$ and decreases toward both ends. Thus, when optimization moves from one side of the function to the other, the gradient magnitude naturally increases to a peak and then decreases. As for the spikes, they are typical **stochastic oscillations** arising during training, since their magnitudes remain well controlled.
>
> For Figs. 8 & 9, our additional experiments further suggest that **initialization** plays an important role. As shown in Figs. 8 & 9, the gradient norm at initialization is around $0.1$, which is **small** and distributed across all coordinates. In this case, optimization effectively “pushes” the model into an active learning region with larger gradients, producing the initial rise, after which the gradient norm decreases as learning progresses. In contrast, when the gradient norm is sufficiently **large** at initialization (i.e., the signal is strong), this rise does not occur. We provide an **anonymous GitHub repository** (https://anonymous.4open.science/r/iclr10822/) to visualize these gradient norm trajectories. The settings are identical to those in Figs. 8 & 9, except that we scale the initialization weights by a factor $\sigma\in\\{0.5,2\\}$.
>
> Thus, we appreciate your insightful observation regarding the interesting phenomenon of **gradient norm in LLM training and RL**, and we **conjecture** that **initialization** may have a similar effect, contributing to this behavior. If this conjecture holds, it could provide valuable insights for future investigation and a deeper understanding of training dynamics in these settings.

---

### Official Review · Reviewer_GWqn · 2025-11-08

**Soundness:** 2
**Presentation:** 3
**Contribution:** 2
**Rating:** 2
**Confidence:** 3

**Summary:**

This paper presents the first theoretical framework analyzing the convergence of adaptive optimizers like Adam and Muon under floating-point quantization of gradients, weights, and optimizer states. It shows that both can maintain near full-precision convergence rates if mantissa precision scales logarithmically with iterations, with Muon proving more robust to quantization errors than Adam.

**Strengths:**

complete quantization error analysis under certain settings, both for Adam and Muon

**Weaknesses:**

Basically experiments are limited, theory is not that informative either in proving practicality. The novelty and contribution is limited.

- line 402 "the second moment (qV) is stricter than for the first moment (qM)" -> there are well known fact existing works e.g., https://arxiv.org/abs/2405.03637 and https://arxiv.org/abs/2405.03637
- many missing connection with stochastic rounding work where it give unbiased estimation, but brining higher variance. In modern low bit training like 4 bits, SR is widely adopted. for 8 bit, MX and NV leads to minimal quantization errors in computing gradient. Moreover  thm3 https://arxiv.org/pdf/2502.20566 has some simpler Adam analysis under quantization error of q_V . better compare.
- experiments are very limited, not covering practical scenarios like LLM, missing of which does not affect any practicality
- more through theoretical analysis is needed, otherwise, it is just naive extension. For example, for β_12(1 + qM)2 <β2(1−qV),the effect of qM and qV is real? any toy example to test?
- for "ensuring the relative quantization errors satisfy qG,qM = O(1/T)", need more explanation as, although theretical understandable, qG is not controllable as a function of T in practice, also dependency on W, W_Q.
- which quantization is most important between weight, gradient, M1, M2? are these supported from theory and aligned with empirical studies?

**Questions:**

See weakness

---

> ### Author Response · Authors · 2025-11-23
> **Responses to Q1, Q3, Q4, Q5, Q6**
>
> We thank Reviewer GWqn for their feedback and for recognizing our work as providing "complete quantization error analysis under certain settings, both for Adam and Muon."
>
> ### Responses to Questions
>
> **Q1: Line 402 states "the second moment (qV) is stricter than for the first moment (qM)" → this is a well-known fact from existing works, e.g., https://arxiv.org/abs/2405.03637**
>
> **A1:** We thank the reviewer for the reference and **have added it in the revised PDF (Marked in Blue, Lines 93-95, 411-413]**. While the stronger sensitivity of the second moment is empirically known, our contribution is providing the **first formal proof** of this effect under a practical floating-point relative-error model. Theorem 4.5 shows that second-moment quantization is amplified through the inverse square-root in Adam’s update, requiring $q_V = \mathcal{O}(1/T^2)$ to match full-precision convergence, whereas the first moment only needs $q_M = \mathcal{O}(1/T)$. This explains, from first principles, the widely observed instability caused by amplified second-moment error when $\beta_2 \to 1$. In contrast, Theorem 4.6 shows Muon avoids this amplification and is therefore much more tolerant.
>
> ---
>
> **Q3: Experiments are very limited, not covering practical scenarios like LLM training, missing of which affects practicality.**
>
> **A3:** We agree that our contribution is primarily theoretical, and the existing Rosenbrock and CIFAR-10 experiments are meant to directly validate the behaviors predicted by our theorems. Within our computational budget, we have additionally run **nanoGPT experiments** (≈26M parameters on ~600M OpenWebText tokens), and the results consistently support our theory: **decreasing mantissa length degrades performance, and Muon remains markedly more robust than Adam under low-precision quantization.** We have **revised the PDF (Blue), including these results in Lines 483-496, App. C.4, Figs. 12 & 13 (pp. 59).**
>
> ---
>
> **Q4: More thorough theoretical analysis is needed; otherwise, it is just a naive extension. For example, for $\beta_1^2(1 + q_M)^2 < \beta_2(1-q_V)$, is the effect of $q_M$ and $q_V$ real? Any toy example to test?**
>
> **A4:** We thank the reviewer for the question. Since $q = \mathcal{O}(2^{-M})$, it can be regarded as a very small constant in practical FP formats, and the condition $\beta_1^2 (1+q_M)^2 < \beta_2(1-q_V)$ reduces almost exactly to the classical requirement $\beta_1^2 < \beta_2$ used in standard convergence analyses. For the common choices $(\beta_1,\beta_2) = (0.9, 0.95/0.99)$, this inequality clearly holds.
>
> The additional dependence on $q_M$ and $q_V$ is introduced only to ensure that our analysis can rigorously control the deviation of the quantized moments throughout the recursion. In our FP error model, $|Q(x)-x|\le q|x|$, the first-moment error $q_M$ effectively inflates its decay factor to $\beta_1(1+q_M)$, while the second-moment error $q_V$ shrinks its decay factor to $\beta_2(1-q_V)$. The inequality guarantees that the maximal possible update, $m/\sqrt{v}$, remains bounded, preventing the inflated first-moment term from overwhelming the second-moment scaling.
>
> ---
>
> **Q5: For "ensuring the relative quantization errors satisfy $q_G, q_M = \mathcal{O}(1/T)$", need more explanation. Although theoretically understandable, $q_G$ is not controllable as a function of $T$ in practice, also dependency on $W$, $W_Q$.**
>
> **A5:** This is an important clarification for the theoretical bounds. The statement $q_G, q_M = \mathcal{O}(1/T)$ is a technical condition to facilitate convergence analysis, to retain full precision convergence. Here, $q_G$ is defined as the maximal relative quantization error when applying $\mathcal{Q}$ to the gradient: $|\mathcal{Q}(g_t)-g_t|\le q_G |g_t|$ for all $t\in[T]$. Its value is determined by the chosen floating-point format and the gradient magnitudes. It's a property of the quantization scheme, not a controllable parameter.
>
> ---
>
> **Q6: Which quantization is most important between weight, gradient, M1, M2? Are these supported from theory and aligned with empirical studies?**
>
> **A6:** Our analysis (Theorem 4.5) proves that for Adam, **weight ($q_W$) and second-moment ($q_V$) quantization** are the most critical, requiring $\mathcal{O}(1/T^2)$ precision, whereas gradients ($q_G$) and first moments ($q_M$) are less sensitive, needing only $\mathcal{O}(1/T)$. This arises because $v_t$ enters as a square root in the denominator, amplifying errors in the second-moment scaling, and because weights directly determine the computed gradient through their influence on the forward pass; even small weight errors can propagate and distort the gradient, making them particularly sensitive to quantization.
>
> This ranking is fully supported by our theoretical bounds and aligns with empirical observations, such as Figure 8 in [1], which report that weights and second moments require higher precision in practice.
>
> > [1] "FP8-LM: Training FP8 Large Language Models", 2023

---

> ### Author Response · Authors · 2025-11-23
> **Responses to Q2 (1/2)**
>
> **Q2: Many missing connections with stochastic rounding work...unbiased estimation...higher variance...Moreover, Thm3 in https://arxiv.org/pdf/2502.20566 has simpler Adam analysis under quantization error of $q_V$. Better compare.**
>
> **A2:** We thank the reviewer for the suggestion. **We have revised main text to clarify this distinction and added the suggested citations in the revised PDF (Marked in Blue, Lines 49-57 in intro, 170-176 in related work, 398-402 in main results).** Stochastic Rounding (SR) is indeed widely used, but it corresponds to a **different quantization regime**: SR is (approximately) unbiased but increases variance, whereas our analysis focuses on the deterministic, biased floating-point rounding inherent to BF16/FP8 hardware (Assumption 3.1), which more accurately reflects the mantissa-limited behavior of modern training pipelines without relying on error-feedback. While SR is used in some 4-bit systems, recent FP4 training [1] typically applies SR only to gradients, and the subsequent low-precision GEMM operations can break end-to-end unbiasedness. This further motivates adopting a relative-error FP model to capture the effective quantization behavior in practice.
>
> **On the comparison to arXiv:2502.20566**
>
> **Regarding the setting:** For the reviewer's note on "Moreover thm3 [Ozkara et al., 2025] has some simpler Adam analysis under quantization error of $q_V$," we respectfully clarify a **key distinction in the setup**: [Ozkara et al., 2025]'s Theorem 3 (Nearest Rounding) assumes full-precision gradients and optimizer states (including the second moment $v_t$), applying quantization solely to the final model update step ($x_{t+1} = Q(x_t - \dots)$). Furthermore, a critical distinction in the theoretical setup is that **they assume $\beta_1 = 0$** for both their SR (Theorem 2) and NR (Theorem 3) analyses, effectively reducing Adam to RMSProp and avoiding the **complex recursive error accumulation** in the first moment ($m_t$). Additionally, they model quantization as an **additive perturbation** ($u_{quant} = u_{exact} + r_t$), whereas our framework employs a **multiplicative relative error model** ($|\delta| \le q|x|$), which aligns with the behavior of standard floating-point hardware. Our work solves the more challenging full Adam setting where $\beta_1 > 0$ and is the first to analyze this full state-quantized setting under a realistic FP model. While prior works primarily consider weight quantization, we explicitly model the quantization of gradients ($q_G$), weights ($q_W$), and optimizer states ($q_M, q_V$), thereby capturing the dominant sources of memory footprint and quantization error in practice.
>
> **Regarding the analysis:** The divergence in theoretical results stems from **the distinct handling of error models in the descent lemma**. **[Ozkara et al., 2025] analyze quantization as an additive perturbation $r_t$**. In their Lemma 2, they bound the expected squared norm (variance) of this perturbation using the update norm ($\mathbb{E}\|r_t\|_2^2 \le \Delta \sqrt{d}\alpha_t \|u_t\|_2$), which introduces a cumulative error term proportional to the sum of update norms $\sum \|u_t\|$ in the descent lemma. To bound this linear sum using the standard quadratic convergence term $\sum \|u_t\|^2$ (as established in [Défossez et al., 2022]), they apply Jensen’s inequality (scaling as $\sum \|u_t\| \le \sqrt{\sum \|u_t\|^2}$), which alters the bound structure by placing the standard Adam log-term under a square root (resulting in terms $T\sqrt{ \ln(1/\beta_2)}$ and $\sqrt{T \ln(1 + \frac{R^2}{\epsilon(1-\beta_2)})}$). In contrast, **our framework employs a multiplicative relative error model ($|\delta| \le q|x|$)** which aligns with standard floating-point hardware but introduces significant analytical challenges due to **recursive error propagation in optimizer states** ($q_M, q_V$) when $\beta_1 > 0$. To resolve this, we introduce auxiliary ideal states $m'_t, v'_t$ (Eq. A.7) and derive a specialized bound (Lemma A.2) to **control the non-linear deviation** $|\frac{1}{\sqrt{\epsilon+v_t}} - \frac{1}{\sqrt{\epsilon+v'_t}}|$. This allows us to rigorously contain the recursive error growth within the adaptive denominator and prove convergence without relying on the simplifications of additive noise models.
>
> > [1] "Pretraining Large Language Models with NVFP4", 2025

---

> ### Author Response · Authors · 2025-11-23
> **Responses to Q2 (2/2)**
>
> **Q2: Many missing connections with stochastic rounding work...unbiased estimation...higher variance...Moreover, Thm3 in https://arxiv.org/pdf/2502.20566 has simpler Adam analysis under quantization error of $q_V$. Better compare.**
>
> **A2:** ...
>
> **On the comparison to arXiv:2502.20566**
>
> ...
>
> **Regarding the result:** For the convergence guarantee,  Our work proves that Quantized Adam recovers the standard full-precision convergence rate $\widetilde{\mathcal{O}}(T^{-1/4})$ under appropriate parameter schedules (e.g., $\beta_2 \to 1$, $\eta \propto 1/\sqrt{T}$, and vanishing quantization error). In contrast, [Ozkara et al., 2025] identify non-vanishing error terms that prevent the algorithm from fully matching the standard rate. Specifically, the **structure of the error terms differs** fundamentally due to the proof techniques required for different error models. In [Ozkara et al., 2025], the additive assumption leads to square-root error terms ($\sqrt{T \ln(1 + \frac{R^2}{\epsilon(1-\beta_2)})}$ and $T\sqrt{\ln(1/\beta_2)}$) in Theorem 2 (SR). For Theorem 3 (NR), the deterministic bias introduces an additional non-vanishing term $\frac{\sqrt{1-\beta_2}d(R\Delta + L\Delta^2)}{\alpha}$ that is absent in the SR case. In contrast, our relative error framework results in more complex interaction terms. Beyond the optimizer state error $\frac{\tilde{Q}(T)}{T}$, our bound explicitly captures weight quantization effects, such as the term $\frac{4(1+q_G)d}{\sqrt{\epsilon(1-\beta_2)}} L q_W R^2 D$. Crucially, if we restrict our setting to weight-only quantization (setting $q_G, q_M, q_V \to 0$) to match their scope, our error bound retains terms involving $q_W$, most notably the term scaling with $q_W T$ (specifically $\frac{2q_W T \eta \dots}{\sqrt{1-\dots}}$), showing that error accumulates as the model updates over $T$ iterations. In contrast, Ozkara's error bound depends on $\Delta = \max_{i,t} \Delta_{x_{t,i}}$, defined as a global constant. In floating-point arithmetic, resolution scales with magnitude ($\Delta_x \approx \epsilon |x|$), meaning that $\Delta$ cannot be treated as a global constant without assuming that weights remain bounded throughout training. By formulating the bound in terms of a constant $\Delta$, **their analysis implicitly hides the dependency on the potentially growing weight magnitude** $\Vert W_t\Vert $**, whereas our framework explicitly exposes this physical reality through the relative error $q_W$ and initial weight bound $\Vert W_0\Vert _F \le D$**, providing a more rigorous characterization of stability under floating-point dynamics.

---

### Meta-Review · Area_Chair_FRPV · 2026-01-20

**Summary:**

This paper presents a theoretical framework analyzing the convergence of adaptive optimizers like Adam and Muon under floating-point quantization of gradients, weights, and optimizer states. It shows that both can maintain near full-precision convergence rates if mantissa precision scales logarithmically with iterations, with Muon proving more robust to quantization errors than Adam.

**Reviewer Concerns:**

The initial review scores were 6, 6, 6, 4, and 2. The positive reviewers (adJ6, d8Q8, kJLe) recognized a rigorous theoretical framework and practical relevance, while requesting further empirical validation on LLM training. The borderline reviewer (qhqY) raised novelty concerns, and the negative reviewer (GWqn) questioned the missing references and theoretical setup. The authors satisfactorily addressed all suggestions and responded to every question raised.

**Reviewer Scores:**

The initial review scores were 6, 6, 6, 4, and 2. The positive reviewers (adJ6, d8Q8, kJLe) recognized a rigorous theoretical framework and practical relevance, while requesting further empirical validation on LLM training. The borderline reviewer (qhqY) raised novelty concerns, and the negative reviewer (GWqn) questioned the missing references and theoretical setup. The authors satisfactorily addressed all suggestions and responded to every question raised.

I believe the scores would change upwards a bit.

---

### Decision · Program_Chairs · 2026-01-26

Accept (Poster)